# MODERNTCN:
# A MODERN PURE CONVOLUTION STRUCTURE FOR GENERAL TIME SERIES ANALYSIS

**Donghao Luo, Xue Wang**
Department of Precision Instrument, Tsinghua University, Beijing 100084, China
`ldh21@mails.tsinghua.edu.cn, wangxue@mail.tsinghua.edu.cn`

## ABSTRACT

Recently, Transformer-based and MLP-based models have emerged rapidly and won dominance in time series analysis. In contrast, convolution is losing steam in time series tasks nowadays for inferior performance. This paper studies the open question of how to better use convolution in time series analysis and makes efforts to bring convolution back to the arena of time series analysis. To this end, we modernize the traditional TCN and conduct time series related modifications to make it more suitable for time series tasks. As the outcome, we propose **ModernTCN** and successfully solve this open question through a seldom-explored way in time series community. As a pure convolution structure, ModernTCN still achieves the consistent state-of-the-art performance on five mainstream time series analysis tasks while maintaining the efficiency advantage of convolution-based models, therefore providing a better balance of efficiency and performance than state-of-the-art Transformer-based and MLP-based models. Our study further reveals that, compared with previous convolution-based models, our ModernTCN has much larger effective receptive fields (ERFs), therefore can better unleash the potential of convolution in time series analysis. Code is available at this repository: https://github.com/luodhhh/ModernTCN.

## 1 INTRODUCTION

Time series analysis is widely used in extensive applications, such as industrial forecasting (Zhou et al., 2021), missing value imputation (Friedman, 1962), action recognition (Ye & Keogh, 2009), and anomaly detection (Xu et al., 2021). Because of the immense practical value, the past few years have witnessed the rapid development in time series analysis(Wen et al., 2022; Lim & Zohren, 2021). Among them, the rise of Transformer-based methods and MLP-based models is especially compelling (Nie et al., 2023; Zhang & Yan, 2023; Zhou et al., 2022; Cirstea et al., 2022; Wu et al., 2021; Liu et al., 2021a; Li et al., 2019b; Kitaev et al., 2020; Vaswani et al., 2017) (Li et al., 2023b; Zhang et al., 2022; Zeng et al., 2022). **But around the same time, convolution-based models have received less attention for a long time.**

**It's non-trivial to use convolution in time series analysis for it provides a better balance of efficiency and performance.** Date back to the 2010s, TCN and its variants (Bai et al., 2018; Sen et al., 2019) are widely-used in many time series tasks. But things have changed in 2020s. Transformer-based models and MLP-based models have emerged rapidly and achieved impressive performance in recent years. Thanks to their global effective receptive fields (ERFs), they can better capture the long-term temporal (cross-time) dependency and thus outperform traditional TCNs by a significant margin. As a result, convolution-based models are losing steam nowadays due to their limited ERFs.

Some previous convolution-based models (Wang et al., 2023; Liu et al., 2022a) try to bring convolution back to the arena of time series analysis. But they mainly focus on designing extra sophisticated structures to work with the traditional convolution, ignoring the importance of updating the convolution itself. And they still cannot achieve comparable performance to the state-of-the-art Transformer-based and MLP-based models. The reason behind can be explained by Figure 1. Increasing the ERF is the key to bringing convolution back to time series analysis. But previous convolution-based models

Figure 1: The Effective Receptive Field (ERF) of ModernTCN and previous convolution-based methods. A more widely distributed light area indicates a larger ERF. Our ModernTCN can obtain a much larger ERF than previous convolution-based methods. Meanwhile, enlarging the kernel size is a more effective way to obtain large ERF than stacking more small kernels.

still have limited ERFs, which prevents their further performance improvements. **How to better use convolution in time series analysis is still a non-trivial and open question.**

As another area where convolution is widely used, computer vision (CV) took a very different path to explore the convolution. Unlike recent studies in time series community, lastest studies in CV focus on optimizing the convolution itself and propose modern convolution (Liu et al., 2022d; Ding et al., 2022; Liu et al., 2022b). Modern convolution is a new convolution paradigm inspired by Transformer. Concretely, modern convolution block incorporates some architectural designs in Transformer and therefore has a similar structure to Transformer block (Figure 2 (a) and (b)). Meanwhile, to catch up with the gloabal ERF in Transformer, modern convolution usually adopat a large kernel as it can effectively increase the ERF (Figure 1). Although the effectiveness of modern convolution has been demonstrated in CV, it has still received little attention from the time series community. Based on above findings, we intend to first modernize the convolution in time series analysis to see whether it can increase the ERF and bring performance improvement.

Besides, convolution is also a potentially efficient way to capture cross-variable dependency. Cross-variable dependency is another critical dependency in time series in addition to the cross-time one. It refers to the dependency among variables in multivariate time series. And early study(Lai et al., 2018b) has already tried to use convolution in variable dimension to capture the cross-variable dependency. Although its performance is not that competitive nowadays, it still demonstrates the feasibility of convolution in capturing cross-variable dependency. Therefore, it's reasonable to believe that convolution can become an efficient and effective way to capture cross-variable dependency after proper modifications and optimizations.

Based on above motivations, we take a seldom-explored way in time series community to successfully bring convolution-based models back to time series analysis. Concretely, we modernize the traditional TCN and conduct some time series related modifications to make it more suitable for time series tasks. As the outcome, we propose a modern pure convolution structure, namely **ModernTCN**, to efficiently utilize cross-time and cross-variable dependency for general time series analysis. We evaluate ModernTCN on five mainstream analysis tasks, including long-term and short-term forecasting, imputation, classification and anomaly detection. Surprisingly, as a pure convolution-based model, ModernTCN still achieves the consistent state-of-the-art performance on these tasks. Meanwhile, ModernTCN also maintains the efficiency advantage of convolution-based models, therefore providing a better balance of efficiency and performance. **Our contributions are as follows:**

- We dive into the question of how to better use convolution in time series and propose a novel solution. Experimental results show that our method can better unleash the potential of convolution in time series analysis than other existing convolution-based models.
- ModernTCN achieves the consistent state-of-the-art performance on multiple mainstream time series analysis tasks, demonstrating the excellent task-generalization ability.
- ModernTCN provides a better balance of efficiency and performance. It maintains the efficiency advantage of convolution-based models while competing favorably with or even better than state-of-the-art Transformer-based models in terms of performance.

## 2 RELATED WORK

### 2.1 CONVOLUTION IN TIME SERIES ANALYSIS

Convolution used to be popular in time series analysis in 2010s. For example, TCN and its variants (Bai et al., 2018; Sen et al., 2019; Franceschi et al., 2019) adopt causal convolution to model the

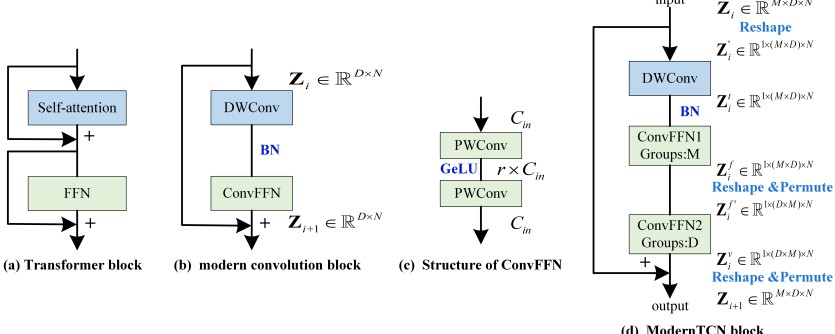

Figure 2: ModernTCN block design. $M$,$N$,$D$ are sizes of variable, temporal and feature dimensions. DWConv and PWConv are short for depth-wise and point-wise convolution (2017). $Groups$ is the group number in group convolution (2018). BN and GeLU (2015; 2016) are adpoted in our design.

temporal causality. But they suffer from the limited ERFs. With the rapid development of Transformer-based and MLP-based models, convolution has received less attention in recent years. Some studies try to bring convolution back to time series community. MICN (Wang et al., 2023) goes beyond causal convolution and proposes a multi-scale convolution structure to combine local features and global correlations in time series. SCINet (Liu et al., 2022a) removes the idea of causal convolution and introduces a recursive downsample-convolve-interact architecture to model time series with complex temporal dynamics. But they still have difficulty in modeling long-term dependency due to the limited ERFs. TimesNet (Wu et al., 2023) is special in the family of convolution-based models. Different from other models that mainly use 1D convolution, it transforms 1D time series into 2D-variations and uses 2D convolution backbones in CV to obtain informative representations.

## 2.2 MODERN CONVOLUTION IN COMPUTER VISION

Convolutional neural networks (ConvNets) (Krizhevsky et al., 2017; Simonyan & Zisserman, 2014; He et al., 2015; Xie et al., 2017; Huang et al., 2017) used to be the dominant backbone architectures in CV. But in 2020s, Vision Transformers (ViTs) (Dosovitskiy et al., 2020; Liu et al., 2021b) are proposed and outperform previous standard ConvNets. To catch up with the performance of ViTs, modern convolution in 2020s are introduced. Inspried by the architectural designs in Transformers, ConvNeXt(Liu et al., 2022d) re-design the convolution block to make it more similar to the Transformer block. To further catch up with the global ERF of Transformers, RepLKNet (Ding et al., 2022) scales the kernel size to 31×31 with the help of Structural Reparameter technique. Further more, SLaK (Liu et al., 2022b) enlarges the kernel size to 51×51 by decomposing a large kernel into two rectangular, parallel kernels and by using dynamic sparsity. Inspired by above studies, we modernize and modify 1D convolution in time series community to make it more suitable for time series analysis tasks.

## 3 MODERNTCN

In this section, we first provide a design roadmap for ModernTCN block to introduce how we modernize and optimize the traditional 1D convolution block in time series community. Then we introduce the overall structure of ModernTCN. And more related details are in Appendix G.

### 3.1 MODERNIZE THE 1D CONVOLUTION BLOCK

Following the idea of (Liu et al., 2022d), we firstly re-design the 1D convolution block as shown in Figure 2 (b). DWConv is responsible for learning the temporal information among tokens on a per-feature basis, which plays the same role as the self-attention module in Transformer. ConvFFN is similar to the FFN module in Transformer. It consists of two PWConvs and adopts an inverted bottleneck structure, where the hidden channel of the ConvFFN block is $r$ times wider than the input channel. This module is to learn the new feature representation of each token independently.

Above design leads to a separation of temporal and feature information mixing. Each of DWConv and ConvFFN only mixes information across one of the temporal or feature dimension, which is

differnet from the traditional convolution that jointly mixes information on both dimensions. This decoupling design can make the object tasks easier to learn and reduce the computational complexity.

Based on above design, we borrow from the success of CV and modernize the 1D convolution. But we find that simply modernizing the convolution in the same way as CV brings little to no performance improvement in time series tasks. In fact, above design does not take into account the characteristics of time series. In addition to feature dimension and temporal dimension, time series also has a variable dimension. But the backbone stacked by convolution blocks as designed in Figure 2 (b) cannot handle the variable dimension properly. Since cross-variable information is also critical in multivariate time series (Zhang & Yan, 2023; Li et al., 2023b), more time series related modifications are still needed to make the modern 1D convolution more suitable for time series analysis.

## 3.2 TIME SERIES RELATED MODIFICATIONS

**Maintaining the Variable Dimension** In CV, before the backbone, we embed 3 channel RGB features at each pixel into a $D$-dimensional vector to mix the information from RGB channels via the embedding layer. But the similar variable-mixing embedding (e.g., simply embed $M$ variables into a $D$-dimensional vector per time step) is not suitable for time series. Firstly, the difference among variables in time series is much greater than that among RGB channels in a picture (Cirstea et al., 2022). Just an embedding layer fails to learn the complex dependency across variables and even loses the independent characteristics of variables for not considering their different behaviors. Secondly, such embedding design leads to the discard of variable dimension, making it unable to further study the cross-variable dependency. To this issue, we propose patchify variable-independent embedding.

We denote $\mathbf{X}_{in} \in \mathbb{R}^{M \times L}$ as the $M$ variables input time series of length $L$ and will further divide it into $N$ patches of patch size $P$ after proper padding (Padding details are in Appendix B). The stride in the patching process is $S$, which also serves as the length of non overlapping region between two consecutive patches. Then the patches will be embedded into $D$-dimensional embedding vectors:

$$\mathbf{X}_{emb} = \text{Embedding}(\mathbf{X}_{in}) \tag{1}$$

$\mathbf{X}_{emb} \in \mathbb{R}^{M \times D \times N}$ is the input embedding. Different from previous studies (Nie et al., 2023; Zhang & Yan, 2023), we conduct this patchify embedding in an equivalent fully-convolution way for a simpler implementation. After unsqueezing the shape to $\mathbf{X}_{in} \in \mathbb{R}^{M \times 1 \times L}$, we feed the padded $\mathbf{X}_{in}$ into a 1D convolution stem layer with kernel size $P$ and stride $S$. And this stem layer maps 1 input channel into $D$ output channels. In above process, each of the $M$ univariate time series is embedded independently. Therefore, we can keep the variable dimension. And followings are modifications to make our structure able to capture information from the additional variable dimension.

**DWConv** DWConv is originally designed for learning the temporal information. Since it's more difficult to jointly learn the cross-time and cross-variable dependency only by DWConv, it's inappropriate to make DWConv also responsible for mixing information across variable dimension. Therefore, we modify the original DWConv from only feature independent to both feature and variable independent, making it learn the temporal dependency of each univariate time series independently. And we adopt large kernel in DWConv to increase ERFs and improve the temporal modeling ability.

**ConvFFN** Since DWConv is feature and variable independent, ConvFFN should mix the information across feature and variable dimensions as a complementary. A naive way is to jointly learn the dependency among features and variables by a single ConvFFN. But such method leads to higher computational complexity and worse performance. Therefore, we further decouple the single ConvFFN into ConvFFN1 and ConvFFN2 by replacing the PWConvs with grouped PWConvs and setting different group numbers. The ConvFFN1 is responsible for learning the new feature representations per variable and the ConvFFN2 is in charge of capturing the cross-variable dependency per feature.

After above modifications, we have the final **ModernTCN block** as shown in Figure 2 (d). And each of DWConv, ConvFFN1 and ConvFFN2 only mixes information across one of the temporal, feature or variable dimension, which maintains the idea of the decoupling design in modern convolution.

## 3.3 OVERALL STRUCTURE

After embedding, $\mathbf{X}_{emb}$ is fed into the backbone to capture both the cross-time and cross-variable dependency and learn the informative representation $\mathbf{Z} \in \mathbb{R}^{M \times D \times N}$:

$$\mathbf{Z} = \text{Backbone}(\mathbf{X}_{emb}) \tag{2}$$

Backbone($\cdot$) is the stacked ModernTCN blocks. Each ModernTCN block is organized in a residual way (He et al., 2015). The forward process in the $i$-th ModernTCN block is:

$$\mathbf{Z}_{i+1} = \mathrm{Block}(\mathbf{Z}_i) + \mathbf{Z}_i \tag{3}$$

Where $\mathbf{Z}_i \in \mathbb{R}^{M \times D \times N}$, $i \in \{1, ..., K\}$ is the $i$-th block's input,

$$\mathbf{Z}_i = \begin{cases} \mathbf{X}_{emb} & , i = 1 \\ \mathrm{Block}(\mathbf{Z}_{i-1}) + \mathbf{Z}_{i-1} & , i > 1 \end{cases} \tag{4}$$

Block($\cdot$) denotes the ModernTCN block. Then the final represnetation $\mathbf{Z} = \mathrm{Block}(\mathbf{Z}_K) + \mathbf{Z}_K$ will be further used for multiple time series analysis tasks. See Appendix B for pipelines of each task.

## 4 EXPERIMENTS

We evaluate ModernTCN on five mainstream analysis tasks, including long-term and short-term forecasting, imputation, classification and anomaly detection to verify the generality of ModernTCN.

**Baselines** Since we attempt to propose a foundation model for time series analysis, we extensively include the latest and advanced models in time series community as basic baselines, which includes the Transformer-based models: PatchTST (2023), Crossformer (2023) and FEDformer (2022); MLP-based models: MTS-Mixer (2023b), LightTS (2022), DLinear (2022), RLinear and RMLP (2023a); Convolution-based Model: TimesNet (2023), MICN (2023) and SCINet (2022a). We also include the state-of-the-art models in each specific task as additional baselines for a comprehensive comparison.

**Main Results** As shown in Figure 3, **ModernTCN achieves consistent state-of-the-art performance on five mainstream analysis tasks with higher efficiency**. Detailed discussions about experimental results are in Section 5.1. We provide the experiment details and results of each task in following subsections. In each table, the best results are in **bold** and the second best are underlined.

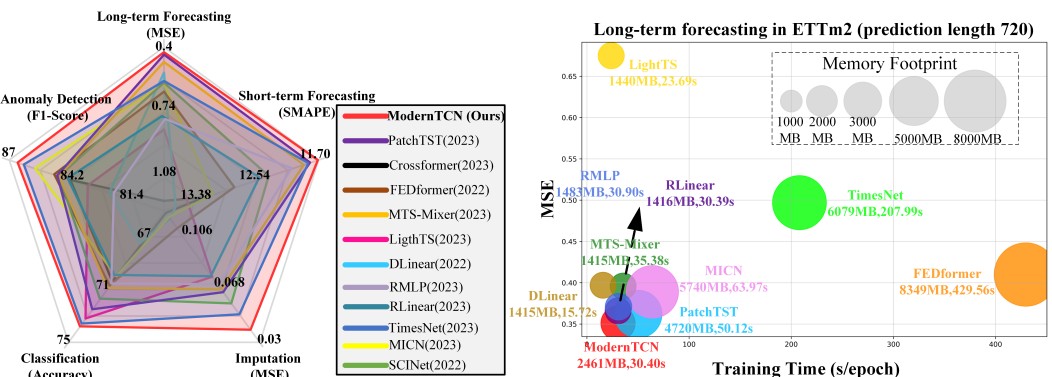

Figure 3: Model performance comparison (left) and efficiency comparison (right).

### 4.1 LONG-TERM FORECASTING

**Setups** We conducted long-term forecasting experiments on 9 popular real-world benchmarks, including Weather (Wetterstation), Traffic (PeMS), Electricity (UCI), Exchange (Lai et al., 2018a), ILI (CDC) and 4 ETT datasets (Zhou et al., 2021). Following (Nie et al., 2023; Zhang & Yan, 2023), we re-run all baselines with various input lengths and choose the best results to avoid under-estimating the baselines and provide a fairer comparison. We calculate the MSE and MAE of multivariate time series forecasting as metrics.

**Results** Table 1 shows the excellent performance of ModernTCN in long-term forecasting. Concretely, ModernTCN gains most of the best performance in above 9 cases, surpassing extensive state-of-the-art MLP-based and Transformer-based models. It competes favorably with the best Transformer-based model PatchTST in terms of performance while having faster speed and less memory usage (Figure 3 right), therefore providing a better balance of performance and efficiency. It's notable that ModernTCN surpasses existing convolution-based models by a large margin (27.4% reduction on MSE and 15.3% reduction on MAE), indicating that our design can better unleash the potential of convolution in time series forecasting.

Table 1: Long-term forecasting task. All the results are averaged from 4 different prediction lengths, that is $\{24, 36, 48, 60\}$ for ILI and $\{96, 192, 336, 720\}$ for the others. A lower MSE or MAE indicates a better performance. See Table 27 in Appendix for the full results with more baselines.

| Models | ModernTCN (Ours) | | PatchTST (2023) | | Crossformer (2023) | | FEDformer (2022) | | MTS-Mixer (2023b) | | RLinear (2023a) | | DLinear (2022) | | TimesNet (2023) | | MICN (2023) | | SCINet (2022a) | |
|---|---|---|---|---|---|---|---|---|---|---|---|---|---|---|---|---|---|---|---|---|
| Metric | MSE | MAE | MSE | MAE | MSE | MAE | MSE | MAE | MSE | MAE | MSE | MAE | MSE | MAE | MSE | MAE | MSE | MAE | MSE | MAE |
| ETTh1 | **0.404** | **0.420** | 0.413 | 0.431 | 0.441 | 0.465 | 0.428 | 0.454 | 0.430 | 0.436 | 0.408 | 0.421 | 0.423 | 0.437 | 0.458 | 0.450 | 0.433 | 0.462 | 0.460 | 0.462 |
| ETTh2 | 0.322 | 0.379 | 0.330 | 0.379 | 0.835 | 0.676 | 0.388 | 0.434 | 0.386 | 0.413 | **0.320** | **0.378** | 0.431 | 0.447 | 0.414 | 0.427 | 0.385 | 0.430 | 0.371 | 0.410 |
| ETTm1 | **0.351** | 0.381 | **0.351** | 0.381 | 0.431 | 0.443 | 0.382 | 0.422 | 0.370 | 0.395 | 0.358 | 0.376 | 0.357 | **0.379** | 0.400 | 0.406 | 0.383 | 0.406 | 0.387 | 0.411 |
| ETTm2 | **0.253** | **0.314** | 0.255 | 0.315 | 0.632 | 0.578 | 0.292 | 0.343 | 0.277 | 0.325 | 0.256 | **0.314** | 0.267 | 0.332 | 0.291 | 0.333 | 0.277 | 0.336 | 0.294 | 0.355 |
| Electricity | **0.156** | **0.253** | 0.159 | **0.253** | 0.293 | 0.351 | 0.207 | 0.321 | 0.173 | 0.272 | 0.169 | 0.261 | 0.177 | 0.274 | 0.192 | 0.295 | 0.182 | 0.292 | 0.195 | 0.281 |
| Weather | **0.224** | **0.264** | 0.226 | **0.264** | 0.230 | 0.290 | 0.310 | 0.357 | 0.235 | 0.272 | 0.247 | 0.279 | 0.240 | 0.300 | 0.259 | 0.287 | 0.242 | 0.298 | 0.287 | 0.317 |
| Traffic | 0.396 | 0.270 | **0.391** | **0.264** | 0.535 | 0.300 | 0.604 | 0.372 | 0.494 | 0.354 | 0.518 | 0.383 | 0.434 | 0.295 | 0.620 | 0.336 | 0.535 | 0.312 | 0.587 | 0.378 |
| Exchange | 0.302 | **0.366** | 0.387 | 0.419 | 0.701 | 0.633 | 0.478 | 0.478 | 0.373 | 0.407 | 0.345 | 0.394 | **0.297** | 0.378 | 0.416 | 0.443 | 0.315 | 0.404 | 0.435 | 0.445 |
| ILI | **1.440** | **0.786** | 1.443 | 0.798 | 3.361 | 1.235 | 2.597 | 1.070 | 1.555 | 0.819 | 4.269 | 1.490 | 2.169 | 1.041 | 2.139 | 0.931 | 2.567 | 1.055 | 2.252 | 1.021 |

Table 2: Short-term forecasting task. Results are weighted averaged from several datasets under different sample intervals. Lower metrics indicate better performance. See Table 28 for full results.

| Models | ModernTCN (Ours) | CARD (2023) | PatchTST (2023) | Crossformer (2023) | FEDformer (2022) | MTS-Mixer (2023b) | RLinear (2023a) | DLinear (2022) | TimesNet (2023) | MICN (2023) | SCINet (2022a) | N-HiTS (2023) | N-BEATS (2019) |
|---|---|---|---|---|---|---|---|---|---|---|---|---|---|
| SMAPE | **11.698** | 11.815 | 11.807 | 13.474 | 12.840 | 11.892 | 12.473 | 13.639 | 11.829 | 13.130 | 12.369 | 11.927 | 11.851 |
| MASE | **1.556** | 1.587 | 1.590 | 1.866 | 1.701 | 1.608 | 1.677 | 2.095 | 1.585 | 1.896 | 1.677 | 1.613 | 1.599 |
| OWA | **0.838** | 0.850 | 0.851 | 0.985 | 0.918 | 0.859 | 0.898 | 1.051 | 0.851 | 0.980 | 0.894 | 0.861 | 0.855 |

## 4.2 SHORT-TERM FORECASTING

**Setups** We adopt M4 dataset (Makridakis et al., 2018) as the short-term forecasting benchmark. Following (Wu et al., 2023), we fix the input length to be 2 times of prediction length and calculate Symmetric Mean Absolute Percentage Error (SMAPE), Mean Absolute Scaled Error (MASE) and Overall Weighted Average (OWA) as metrics. We include additional baselines like CARD (2023), N-BEATS (2019) and N-HiTS (2023) for this specific task. Since the M4 dataset only contains univariate time series, we remove the cross-variable component in ModernTCN and Crossformer.

**Results** The results are summarized in Table 2. Short-term forecasting in M4 dataset is a much more challenging task because the time series samples are collected from different sources and have quite different temporal properties. Our ModernTCN still achieves the consistent state-of-the-art in this difficult task, demonstrating its excellent temporal modeling ability.

## 4.3 IMPUTATION

**Setups** Imputation task aims to impute the missing values based on the partially observed time series. Due to unexpected accidents like equipment malfunctions or communication error, missing values in time series are very common. Since missing values may harm the performance of downstream analysis, imputation task is of high practical value. Following (Wu et al., 2023), we mainly focus on electricity and weather scenarios where the data-missing problem happens commonly. We select the datasets from these scenarios as benchmarks, including ETT (Zhou et al., 2021), Electricity (UCI) and Weather (Wetterstation). We randomly mask the time points in ratios of $\{12.5\%, 25\%, 37.5\%, 50\%\}$ to compare the model capacity under different proportions of missing data.

**Results** Table 3 shows the compelling performance of ModernTCN in imputation tasks. ModernTCN achieves 22.5% reduction on MSE and 12.9% reduction on MAE compared with previous state-of-the-art baseline TimesNet (2023). Due to the missing values, the remaining observed time series is irregular, making it more difficult to capture cross-time dependency. Our ModernTCN still

Table 3: Imputation task. We randomly mask $\{12.5\%, 25\%, 37.5\%, 50\%\}$ time points in length-96 time series. The results are averaged from 4 different mask ratios. A lower MSE or MAE indicates a better performance. See Table 29 in Appendix for full results with more baselines.

| Models | ModernTCN (Ours) | | PatchTST (2023) | | Crossformer (2023) | | FEDformer (2022) | | MTS-Mixer (2023b) | | RLinear (2023a) | | DLinear (2022) | | TimesNet (2023) | | MICN (2023) | | SCINet (2022a) | |
|---|---|---|---|---|---|---|---|---|---|---|---|---|---|---|---|---|---|---|---|---|---|
| Averaged | MSE | MAE | MSE | MAE | MSE | MAE | MSE | MAE | MSE | MAE | MSE | MAE | MSE | MAE | MSE | MAE | MSE | MAE | MSE | MAE |
| ETTm1 | **0.020** | **0.093** | 0.045 | 0.133 | 0.041 | 0.143 | 0.062 | 0.177 | 0.056 | 0.154 | 0.070 | 0.166 | 0.093 | 0.206 | 0.027 | 0.107 | 0.070 | 0.182 | 0.039 | 0.129 |
| ETTm2 | **0.019** | **0.082** | 0.028 | 0.098 | 0.046 | 0.149 | 0.101 | 0.215 | 0.032 | 0.107 | 0.032 | 0.108 | 0.096 | 0.208 | 0.022 | 0.088 | 0.144 | 0.249 | 0.027 | 0.102 |
| ETTh1 | **0.050** | **0.150** | 0.133 | 0.236 | 0.132 | 0.251 | 0.117 | 0.246 | 0.127 | 0.236 | 0.141 | 0.242 | 0.201 | 0.306 | 0.078 | 0.187 | 0.125 | 0.250 | 0.104 | 0.216 |
| ETTh2 | **0.042** | **0.131** | 0.066 | 0.164 | 0.122 | 0.240 | 0.163 | 0.279 | 0.069 | 0.168 | 0.066 | 0.165 | 0.142 | 0.259 | 0.049 | 0.146 | 0.205 | 0.307 | 0.064 | 0.165 |
| Electricity | **0.073** | **0.187** | 0.091 | 0.209 | 0.083 | 0.199 | 0.130 | 0.259 | 0.089 | 0.208 | 0.119 | 0.246 | 0.132 | 0.260 | 0.092 | 0.210 | 0.119 | 0.247 | 0.086 | 0.201 |
| Weather | **0.027** | **0.044** | 0.033 | 0.057 | 0.036 | 0.090 | 0.099 | 0.203 | 0.036 | 0.058 | 0.034 | 0.058 | 0.052 | 0.110 | 0.030 | 0.054 | 0.056 | 0.128 | 0.031 | 0.053 |

achieves the best performance in this challenging task, verifying the model capacity in capturing temporal dependency under extremely complicated situations.

It's also notable that cross-variable dependency plays a vital role in imputation task. Since in some time steps, only part of the variables are missing while others are still remaining, utilizing the cross-variable dependency between missing variables and remaining variables can help to effectively impute the missing values. Therefore, some variable-independent methods like PatchTST (2023) and DLinear (2022) fail in this task for not taking cross-variable dependency into consideration.

## 4.4 CLASSIFICATION AND ANOMALY DETECTION

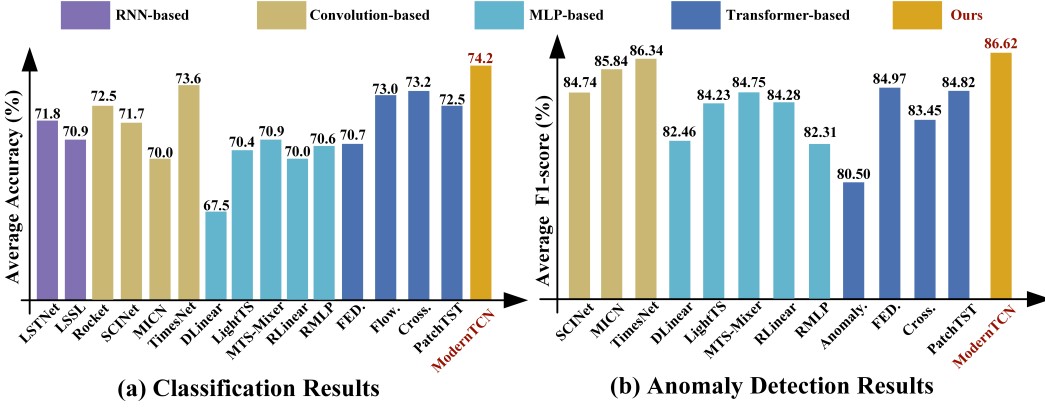

Figure 4: Results of classification and anomaly detection. The results are averaged from several datasets. Higher accuracy and F1 score indicate better performance. *. in the Transformer-based models indicates the name of *former. See Table 30 and 31 in Appendix for full results.

**Setups** For classification, we select 10 multivariate datasets from UEA Time Series Classification Archive (Bagnall et al., 2018) for benchmarking and pre-process the datasets following (Wu et al., 2023). We include some task-specific state-of-the-art methods like LSTNet (2018b), Rocket (2020) and Flowformer (2022) as additional baselines.

For anomaly detection, we compare models on five widely-used benchmarks: SMD (Su et al., 2019), SWaT (Mathur & Tippenhauer, 2016), PSM (Abdulaal et al., 2021), MSL and SMAP (Hundman et al., 2018). We include Anomaly transformer (2021) as additional baselines. Following it, we adopt the classical reconstruction task and choose the reconstruction error as the anomaly criterion.

**Results** **Time series classification** is a classic task in time series community and reflects the model capacity in high-level representation. As shown in Figure 4, ModernTCN achieves the best performance with an average accuracy of 74.2% . It's notable that some MLP-based models fail in classification tasks. This is because MLP-based models prefer to discard the feature dimension to

obtain a lightweight backbone, which leads to the insufficient representation capability and inferior classification performance. **Anomaly detection** results are shown in Figure 4. ModernTCN achieves competitive performance with previous state-of-the-art baseline TimesNet (2023). Meanwhile, compared with TimesNet, ModernTCN saves 55.1% average training time per epoch (3.19s vs 7.10s) in classification task and saves 57.3% average training time per epoch (132.65s vs 310.62s) in anomaly detection task, providing a better balance of efficiency and performance in both tasks.

## 5 MODEL ANALYSIS

### 5.1 COMPREHENSIVE COMPARISON OF PERFORMANCE AND EFFICIENCY

**Summary of Experimental Results**   ModernTCN achieves consistent state-of-the-art performance on five mainstream analysis tasks compared with other task-specific models or previous state-of-the-art baselines, demonstrating its excellent task-generality and highlighting the potential of convolution in time series analysis (Figure 3 left). ModernTCN also has more advantage in efficiency, therefore providing a better balance of efficiency and performance (Figure 3 right). It's worth noting that our method surpasses existing convolution-based models by a large margin, indicating that our design can provide a better solution to the problem of how to better use convolution in time series analysis.

**Compared with Transformer-based and MLP-based Models**   Unlike previous convolution-based models, ModernTCN competes favorably with or even better than state-of-the-art Transformer-based models in terms of performance. Meanwhile, as a pure convolution model, ModernTCN has higher efficiency than Transformer-based models. As shown in Figure 3 right, ModernTCN has faster training speed and less memory usage, which demonstrates the efficiency superiority of our model.

ModernTCN outperforms all MLP-based baselines in all five tasks thanks to the better representation capability in ModernTCN blocks. In contrast, MLP-based models prefer to adopt a lightweight backbone for a smaller memory usage. But such design in MLP-based models also leads to the insufficient representation capability and inferior performance. Although ModernTCN is sightly inferior in memory usage, it still has almost the same running time efficiency as some MLP-based baselines thanks to the fast floating point operation speed in convolution. **Considering both performance and efficiency, ModernTCN has more advantage in general time series analysis.**

**Compared with TimesNet (2023)**   In addition to ModernTCN, TimesNet also demonstrates excellent generality in five mainstream tasks. It's worth noting that both models are convolution-based models, which further reveals that convolution has a better comprehensive ability in time series analysis. **Meanwhile, both methods are inspired by CV and intend to make the time series analysis take advantage of the development of CV community.** But the two methods take different paths to accomplish this goal. TimesNet makes efforts to transform the 1D time series into 2D space, making the time series can be modeled by the 2D ConvNets in CV community. But the additional data transformation and aggregation modules also bring extra memory usage and slower training speed. Different from TimesNet, our ModernTCN maintains the 1D time series and turns to modernize and optimize the 1D convolution in time series community. Therefore, we design a modern pure convolution structure that without any additional modules. The fully-convolutional nature in our design brings higher efficiency and makes it extremely simple to implement, therefore leading to the both performance and efficiency superiority than TimesNet (Figure 3 left and right).

### 5.2 ANALYSIS OF EFFECTIVE RECEPTIVE FIELD (ERF)

Enlarging the ERF is the key to bring convolution back to time series analysis. In this section, we will discuss why ModernTCN can provide better performance than previous convolution-based models from the perspective of ERF. Firstly, rather than stacking more layers like other traditional TCNs (2018), ModernTCN increases the ERF by enlarging the kernel size. And in a pure convolution structure, enlarging the kernel size is a much more effective way to increases ERF. According to the theory of ERF in pure convolution-based models (Luo et al., 2016), ERF is proportion to $O(ks \times \sqrt{nl})$, where $ks$ and $nl$ refers to the kernel size and the number of layers respectively. ERF grows linearly with the kernel size while sub-linearly with the layer number. Therefore, by enlarging the kernel size, ModernTCN can easily obtain a larger ERF and further bring perfomance improvement.

Except for enlarging the kernel size and stacking more layers, some previous convolution-based methods in time series community (MICN (2023) and SCINet (2022a)) prefer to adopt some sophisticated structures to cooperate with the traditional convolution, intending to enlarge their ERFs. Since they are not pure convolution structures, it's hard to analyse their ERFs theoretically. Therefore, we visualize the ERFs of these methods for intuitive comparision. Following (Kim et al., 2023), we sample 50 length-336 input time series from the validation set in ETTh1 for the visualization. The idea behind is to visualize how many points in the input series can make contribution to the middle point of the final feature map. As shown in Figure 1, our method can obtain a much larger ERF than previous convolution-based methods. Therefore our method can better unleash the potential of convolution in time series and sucessfully bring performance improvements in multiple time series analysis tasks.

## 5.3 ABLATION STUDY

**Ablation of ModernTCN Block Design** To validate the effectiveness of our design in ModernTCN block, we conduct ablation study in long-term forecasting tasks. Results are shown on Table 4. *Discard Variable Dimension* cannot provide ideal performance, which confirms our arguement that simply modernizing the convolution in the same way as CV could not bring performance improvement for omitting the importance of variable dimension. To better handle the variable dimension, we decouple a single ConvFFN into ConvFFN1 and ConvFFN2 in our design. As shown in Table 4, the *undecoupled ConvFFN* provide the worst performance and the combination of our decoupled two ConvFFNs (*ConvFFN1+ConvFFN2*) achieve the best, which proves the necessity and effectiveness of our decouple modification to ConvFFN module. Please see Appendix H for more details.

Table 4: Ablation of ModernTCN block. We list the averaged MSE/MAE of different forecast lengths.

| Dataset | ETTh1 | | ETTh2 | | ETTm1 | | ETTm2 | | ECL | | Weather | | Exchange | | ILI | |
|---|---|---|---|---|---|---|---|---|---|---|---|---|---|---|---|---|
| Metric | MSE | MAE | MSE | MAE | MSE | MAE | MSE | MAE | MSE | MAE | MSE | MAE | MSE | MAE | MSE | MAE |
| **ConvFFN1 + ConvFFN2** | **0.404** | **0.420** | **0.322** | **0.379** | **0.351** | **0.381** | **0.253** | **0.314** | **0.156** | **0.253** | **0.224** | **0.264** | **0.302** | **0.366** | **1.440** | **0.786** |
| Undecoupled ConvFFN | 0.425 | 0.440 | 0.341 | 0.391 | 0.370 | 0.395 | 0.278 | 0.332 | 0.169 | 0.269 | 0.260 | 0.288 | 0.323 | 0.378 | 1.813 | 0.875 |
| only ConvFFN1 | 0.418 | 0.434 | 0.329 | 0.384 | 0.360 | 0.386 | 0.267 | 0.323 | 0.163 | 0.264 | 0.241 | 0.278 | 0.310 | 0.371 | 1.667 | 0.866 |
| only ConvFFN2 | 0.416 | 0.438 | 0.330 | 0.384 | 0.359 | 0.385 | 0.268 | 0.325 | 0.164 | 0.265 | 0.249 | 0.283 | 0.312 | 0.371 | 1.874 | 0.882 |
| ConvFFN1 + ConvFFN1 | 0.420 | 0.435 | 0.329 | 0.385 | 0.363 | 0.386 | 0.267 | 0.323 | 0.165 | 0.265 | 0.242 | 0.279 | 0.311 | 0.371 | 1.710 | 0.866 |
| ConvFFN2 + ConvFFN2 | 0.417 | 0.437 | 0.330 | 0.383 | 0.362 | 0.384 | 0.269 | 0.325 | 0.165 | 0.265 | 0.254 | 0.285 | 0.311 | 0.371 | 1.831 | 0.884 |
| Discard Variable Dimension | 0.590 | 0.560 | 0.382 | 0.430 | 0.508 | 0.494 | 0.319 | 0.367 | 0.441 | 0.486 | 0.300 | 0.331 | 0.361 | 0.412 | 1.932 | 0.936 |

**Ablation of Cross-variable Component** As an important time series related modification in our design, we design the ConvFFN2 as a cross-variable component to capture the cross-variable dependency. We conduct ablation studies in imputation tasks and anomaly detection tasks. As shown in Table 5, without the ConvFFN2 will cause severe performance degradation in these two tasks, which emphasizes the importance of cross-variable dependency in time series analysis.

Table 5: Ablation of Cross-variable component.

| | Dataset | ETTm1 | | ETTm2 | | ETTh1 | | ETTh2 | | ECL | | Weather | |
|---|---|---|---|---|---|---|---|---|---|---|---|---|---|
| | Metric | MSE | MAE | MSE | MAE | MSE | MAE | MSE | MAE | MSE | MAE | MSE | MAE |
| Imputation | **Ours** | **0.020** | **0.093** | **0.019** | **0.082** | **0.050** | **0.150** | **0.042** | **0.131** | **0.073** | **0.187** | **0.027** | **0.044** |
| | w/o Cross-variable | 0.038 | 0.121 | 0.028 | 0.097 | 0.089 | 0.195 | 0.059 | 0.150 | 0.096 | 0.203 | 0.030 | 0.048 |
| | Promotion | 47.4% | 23.1% | 32.1% | 15.5% | 43.8% | 23.1% | 28.8% | 12.7% | 24.0%% | 7.9% | 10.0% | 8.3% |

| | Dataset | SMD | MSL | SMAP | SWaT | PSM | |
|---|---|---|---|---|---|---|---|
| | Metric | F1-score | F1-score | F1-score | F1-score | F1-score | |
| Anomaly Detection | **Ours** | **85.81** | **84.92** | **71.26** | **93.86** | **97.23** | |
| | w/o Cross-variable | 81.33 | 72.13 | 64.93 | 83.46 | 96.08 | |
| | Promotion | 5.5% | 17.7% | 9.7% | 12.5% | 1.2% | |

## 6 CONCLUSION AND FUTURE WORK

In this paper, we take a seldom-explored way in time series community to solve the question of how to better use convolution in time series analysis. By modernizing and modifying the traditional TCN block with time series related modifications, we propose ModernTCN and successfully bring convolution back to the arena of time series analysis. Experimental results show the great task generality of ModernTCN. While performing on par with or better than state-of-the-art Transformer-based models in terms of performance, ModernTCN maintains the efficiency advantage of convolution-based models, therefore providing a better balance of performance and efficiency. Since convolution-based models have received less attention in time series analysis for a long time, we hope that the new results reported in this study will bring some fresh perspectives to time series community and prompt people to rethink the importance of convolution in time series analysis.

ACKNOWLEDGMENT

This work was supported by the Guangdong Key Area Research and Development Program (2019B010154001) and Hebei Innovation Plan (20540301D).

REPRODUCIBILITY STATEMENT

The model architecture is introduced in details with equations and figures in the main text. And all the implementation details are included in the Appendix, including dataset descriptions, metrics of each task, model configurations and experiment settings. Code is available at this repository: https://github.com/luodhhh/ModernTCN.

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

Table 6: Dataset descriptions of long-term forecasting and imputation.

| Dataset | Weather | Traffic | Exchange | Electricity | ILI | ETTh1 | ETTh2 | ETTm1 | ETTm2 |
|---|---|---|---|---|---|---|---|---|---|
| Dataset Size | 52696 | 17544 | 7207 | 26304 | 966 | 17420 | 17420 | 69680 | 69680 |
| Variable Number | 21 | 862 | 8 | 321 | 7 | 7 | 7 | 7 | 7 |
| Sampling Frequency | 10 mins | 1 hour | 1 day | 1 hour | 1 week | 1 hour | 1 hour | 15 mins | 15 mins |

## A  DATASETS

### A.1  LONG-TERM FORECASTING AND IMPUTATION DATASETS

We evaluate the long-term forecasting performance on 9 popular real-world datasets, including Weather, Traffic, Electricity, Exchange, ILI and 4 ETT datasets (ETTh1, ETTh2, ETTm1, ETTm2). And for imputation tasks, we choose Weather, Electricity and 4 ETT datasets (ETTh1, ETTh2, ETTm1, ETTm2) for benchmarking. These datasets have been extensively utilized for benchmarking and cover many aspects of life.

The dataset size (total timesteps), variable number and sampling frequency of each dataset are summarized in Table 6 . We follow standard protocol (Zhou et al., 2021) and split all datasets into training, validation and test set in chronological order by the ratio of 6:2:2 for the ETT dataset and 7:1:2 for the other datasets. And training, validation and test sets are zero-mean normalized with the mean and standard deviation of training set. Each of above datasets only contains one continuous long time series, and we obtain samples by sliding window.

More introduction of the datasets are as follow:

1) **Weather**[1] contains 21 meteorological indicators such as humidity and air temperature for 2020 whole year in Germany.

2) **Traffic**[2] contains the road occupancy rates measured by 862 different sensors on San Francisco Bay area freeways in 2 years. Data is collected from California Department of Transportation.

3) **Electricity**[3] contains hourly electricity consumption of 321 clients from 2012 to 2014.

4) **Exchange**[4] the daily exchange rates of eight different countries ranging from 1990 to 2016.

5) **ILI**(Influenza-Like Illness)[5] contains the weekly recorded influenza-like illness (ILI) patients data in the United States between 2002 and 2021. It contains 7 indicators like the numbers of ILI patients under different age ranges and the ratio of ILI patients to the total patients. Data is provided by Centers for Disease Control and Prevention of the United States.

6) **ETT**(Electricity Transformer Temperature)[6] contains the data collected from electricity transformers with 7 sensors, including load, oil temperature, etc. It contains two sub-dataset labeled with 1 and 2, corresponding to two different electric transformers from two separated counties in China. And each of them contains 2 different resolutions (15 minutes and 1 hour) denoted with m and h. Thus, in total we have 4 ETT datasets: ETTh1, ETTh2, ETTm1, ETTm2.

### A.2  SHORT-TERM FORECASTING DATASETS

M4 involves 100,000 different time series samples collected in different domains with different frequencies, covering a wide range of economic, industrial, financial and demographic areas.

It's notable that M4 dataset is different from the long-term forecasting datasets. Each of long-term forecasting dataset only contains one continuous long time series, and we obtain samples by sliding window. Therefore all samples are come from the same source time series and more likely to have similar temporal property. But the samples in M4 datasets are collected from different sources.

---

[1]https://www.bgc-jena.mpg.de/wetter/

[2]https://pems.dot.ca.gov/

[3]https://archive.ics.uci.edu/ml/datasets/ElectricityLoadDiagrams20112014

[4]https://github.com/laiguokun/multivariate-time-series-data

[5]https://gis.cdc.gov/grasp/fluview/fluportaldashboard.html

[6]https://github.com/zhouhaoyi/ETDataset

Therefore they may have quite different temporal property, making the forecasting tasks in M4 datasets more difficult. Table 7 summarizes details of statistics of short-term forecasting M4 datasets.

Table 7: Datasets and mapping details of M4 dataset (Makridakis et al., 2018).

| Dataset | Sample Numbers (train set,test set) | Variable Number | Prediction Length |
|---------|-------------------------------------|-----------------|-------------------|
| M4 Yearly | (23000, 23000) | 1 | 6 |
| M4 Quarterly | (24000, 24000) | 1 | 8 |
| M4 Monthly | (48000, 48000) | 1 | 18 |
| M4 Weekly | (359, 359) | 1 | 13 |
| M4 Daily | (4227, 4227) | 1 | 14 |
| M4 Hourly | (414, 414) | 1 | 48 |

### A.3 CLASSIFICATION DATASETS

UEA dataset involves many time series samples collected in different domains for classification, covering the recognition tasks based on face, gesture, action and audio as well as other practical tasks like industry monitoring, health monitoring and medical diagnosis based on heartbeat. Most of them have 10 classes. Table 8 summarizes details of statistics of classification UEA datasets.

Table 8: Datasets and mapping details of UEA dataset (Bagnall et al., 2018).

| Dataset | Sample Numbers (train set,test set) | Variable Number | Series Length |
|---------|-------------------------------------|-----------------|---------------|
| EthanolConcentration | (261, 263) | 3 | 1751 |
| FaceDetection | (5890, 3524) | 144 | 62 |
| Handwriting | (150, 850) | 3 | 152 |
| Heartbeat | (204, 205) | 61 | 405 |
| JapaneseVowels | (270, 370) | 12 | 29 |
| PEMS-SF | (267, 173) | 963 | 144 |
| SelfRegulationSCP1 | (268, 293) | 6 | 896 |
| SelfRegulationSCP2 | (200, 180) | 7 | 1152 |
| SpokenArabicDigits | (6599, 2199) | 13 | 93 |
| UWaveGestureLibrary | (120, 320) | 3 | 315 |

### A.4 ANOMALY DETECTION DATASETS

We adopt datasets from different domains like server machine, spacecraft and infrastructure for benchmarking. Each dataset is divided into training, validation and testing sets. Each dataset contains one continuous long time series, and we obtain samples from the continuous long time series with a fixed length sliding window. Table 9 summarizes details of statistics of the datasets.

Table 9: Datasets and mapping details of anomaly detection dataset.

| Dataset | Dataset sizes(train set,val set, test set) | Variable Number | Sliding Window Length |
|---------|--------------------------------------------|-----------------|-----------------------|
| SMD | (566724, 141681, 708420) | 38 | 100 |
| MSL | (44653, 11664, 73729) | 55 | 100 |
| SMAP | (108146, 27037, 427617) | 25 | 100 |
| SWaT | (396000, 99000, 449919) | 51 | 100 |
| PSM | (105984, 26497, 87841) | 25 | 100 |

## B PIPELINE

### B.1 Padding Details in Patchify Variable-independent Embedding

Before patching and embedding, we adopt a padding operation on the original time series $\mathbf{X}_{in}$ to keep $N = L//S$. Specifically, we repeat $\mathbf{X}_{in}$'s last value $(P - S)$ times and then pad them back to the end of $\mathbf{X}_{in}$.

Denoted $\mathbf{X}_{in} \in \mathbb{R}^{M \times L}$ as the $M$ variables input time series of length $L$, the overall process of Patchify Variable-independent Embedding is as follows:

1) Unsqueezing its shape to $\mathbf{X}_{in} \in \mathbb{R}^{M \times 1 \times L}$.

2) Adopting above padding operation on it.

3) Feeding the padded $\mathbf{X}_{in}$ to the 1D convolution stem layer for patching and embedding.

### B.2 Pipeline for Regression Tasks

The pipeline for forecasting, imputation and anomaly detection is shown as Figure 5.

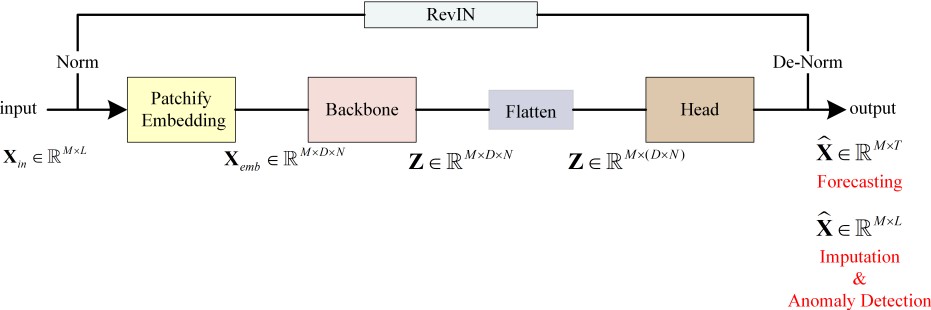

Figure 5: Pipeline for Regression Tasks.

After the backbone, we have $\mathbf{Z} \in \mathbb{R}^{M \times D \times N}$. Then the linear head with a flatten layer is used to obtain the final prediction:

$$\widehat{\mathbf{X}} = \text{Head}(\text{Flatten}(\mathbf{Z})) \tag{5}$$

Where $\widehat{\mathbf{X}} \in \mathbb{R}^{M \times T}$ is the prediction of length $T$ with $M$ variables. $\text{Flatten}(\cdot)$ denotes a flatten layer that changes the final representation's shape to $\mathbf{Z} \in \mathbb{R}^{M \times (D \times N)}$. $\text{Head}(\cdot)$ indicates the linear projection layer that maps the final representation to the final prediction.

**Stationary Technique** RevIN (Kim et al., 2021) is a special instance normalization for time series to mitigate the distribution shift between the training and testing data. In norm phase, we normalize the input time series per variable with zero mean and unit standard deviation before patching and embedding. Then in de-norm phase, we add the mean and deviation back to the final prediction per variable after the forward process.

**Low Rank Approximation for Traffic Datasets** To Traffic dataset that contains much more variables than others, directly applying our model to Traffic dataset leads to heavy memory usage. Since the variables in multivariate times series have dependency on each other, a possible way to solve this problem is to find a low rank approximation of these $M$ variables when $M$ is a very big number. For example, FEDformer (2022) uses a low rank approximated transformation in frequency domain for better memory efficiency. And Crossformer (2023) also uses a small fixed number of routers to aggregate messages from all variables to save memory usage.

In this paper, we design a bottleneck structure as a simple and direct method to achieve this goal. In details, before fed into the ConvFFN1 and ConvFFN2, the variable number will be projected to $M'$ by a projection layer, where $M'$ is much smaller than $M$. Then after the ConvFFN1 and ConvFFN2 process, another projection layer is used to project the variable number back to $M$.

Table 10: Results with different $M'$ in Traffic dataset.

| Models | w/o | | $M' = 256$ | | $M' = 64$ | | $M' = 16$ | |
|---|---|---|---|---|---|---|---|---|
| Metric | MSE | MAE | MSE | MAE | MSE | MAE | MSE | MAE |
| Traffic | 0.393 | 0.267 | 0.396 | 0.273 | 0.395 | 0.271 | 0.396 | 0.270 |
| Memory Usage (%) | 100% | | 82% | | 77% | | 72% | |

And we conduct experiments with different $M'$ to verify this solution. As shown in Table 10, our method can significantly reduce memory usage with only a little performance degradation. This result proves the fact that there is redundancy between the 862 variables in Traffic dataset. Therefore we can learn a low rank approximation of these 862 variables based on their dependency on each other. And such low rank approximation can help to reduce memory usage without too much performance degradation.

**Input**   The input is $M$ variables time series with input length $L$. In imputaiton tasks, the input series will further element-wise multipy with a mask matrix to represent the randomly missing values.

**Output**   In forecastiong tasks, the output is the prediction time series with prediction length $T$. In impuatation tasks, the output is the imputed input time series with input length $L$. In anomaly detection tasks, the output is the reconstructed input time series with input length $L$.

### B.3   PIPELINE FOR CLASSIFICATION TASKS

The pipeline for classification is shown as Figure 6.

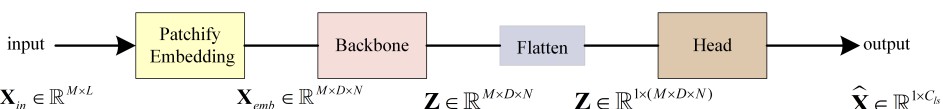

Figure 6: Pipeline for Classification Tasks.

There are two difference: (1) We remove the RevIN; (2) The flatten layer is different. In classification tasks, the flatten layer changes the final representation's shape to $\mathbf{Z} \in \mathbb{R}^{1 \times (M \times D \times N)}$. Then a projection layer with SoftMax activation is to map the final representation to the final classification result $\widehat{\mathbf{X}} \in \mathbb{R}^{1 \times C_{ls}}$, where $C_{ls}$ is the number of classes.

**Followings are implementation details and model parameters of each tasks.**

## C   EXPERIMENT DETAILS

### C.1   LONG-TERM FORECASTING

**Implementation Details**   Our method is trained with the L2 loss, using the ADAM (Kingma & Ba, 2014) optimizer with an initial learning rate of $10^{-4}$. The default training process is 100 epochs with proper early stopping. The mean square error (MSE) and mean absolute error (MAE) are used as metrics. All the experiments are repeated 5 times with different seeds and the means of the metrics are reported as the final results. All the deep learning networks are implemented in PyTorch(Paszke et al., 2019) and conducted on NVIDIA A100 40GB GPU.

All of the models are following the same experimental setup with prediction length $T \in \{24, 36, 48, 60\}$ for ILI dataset and $T \in \{96, 192, 336, 720\}$ for other datasets as (Nie et al., 2023). We collect some baseline results from (Nie et al., 2023) where all the baselines are re-run with various input length $L$ and the best results are chosen to avoid under-estimating the baselines. For other baselines, we follow the official implementation and run them with vary input length $L \in \{24, 36, 48, 60, 104, 144\}$ for ILI dataset and $L \in \{96, 192, 336, 512, 672, 720\}$ for other

datasets and choose the best results. All experiments are repeated five times. We calculate the MSE and MAE of multivariate time series forecasting as metrics.

**Model Parameter**  By default, ModernTCN contains 1 ModernTCN block with the channel number (dimension of hidden states) $D = 64$ and FFN ratio $r = 8$. The kernel size is set as $large\ size = 51$ and $small\ size = 5$. Patch size and stride are set as $P = 8, S = 4$ in the patchify embedding process. For bigger datasets (ETTm1 and ETTm2), we stack 3 ModernTCN blocks to improve the representation capability. For small datasets (ETTh1, ETTh2, Exchange and ILI), we recommend a small FFN ratio $r = 1$ to mitigate the possible overfitting and for better memory efficiency.

For baseline models, if the original papers conduct long-term forecasting experiments on the dataset we use, we follow the recommended model parameters in the original papers, including the number of layers, dimension of hidden states, etc. But we re-run them with vary input lengths as mentioned in Section 4.1 and choose the best results to obtain a strong baseline.

**Metric**  We adopt the mean square error (MSE) and mean absolute error (MAE) for long-term forecasting.

$$\text{MSE} = \frac{1}{T}\sum_{i=0}^{T}(\widehat{\mathbf{X}}_i - \mathbf{X}_i)^2$$

$$\text{MAE} = \frac{1}{T}\sum_{i=0}^{T}\left|\widehat{\mathbf{X}}_i - \mathbf{X}_i\right|$$

where $\widehat{\mathbf{X}}, \mathbf{X} \in \mathbb{R}^{T \times M}$ are the $M$ variables prediction results of length $T$ and corresponding ground truth. $\mathbf{X}_i$ means the $i$-th time step in the prediction result.

## C.2  Short-term Forecasting

**Implementation Details**  Our method is trained with the SMAPE loss, using the ADAM (Kingma & Ba, 2014) optimizer with an initial learning rate of $5 \times 10^{-4}$. The default training process is 100 epochs with proper early stopping. The symmetric mean absolute percentage error (SMAPE), mean absolute scaled error (MASE) and overall weighted average (OWA) are used as metrics. All the experiments are repeated 5 times with different seeds and the means of the metrics are reported as the final results.

Following (Wu et al., 2023), we fix the input length to be 2 times of prediction length for all models. Since the M4 dataset only contains univariate time series, we remove the cross-variable component in ModernTCN and Crossformer.

**Model Parameter**  By default, ModernTCN contains 2 ModernTCN blocks with the channel number (dimension of hidden states) $D = 2048$ and FFN ratio $r = 1$. The kernel size is set as $large\ size = 51$ and $small\ size = 5$. For datasets of less samples (M4 Weekly, M4 Daily and M4 Hourly), we use a smaller channel number $D = 1024$. Patch size and stride are set as $P = 8, S = 4$ in the patchify embedding process. For datasets with shorter input length, we reduce the patch size and stride (e.g., $P = 3, S = 3$ in M4 Yearly and $P = 2, S = 2$ in M4 Quarterly).

**Metric**  For the short-term forecasting, following (Oreshkin et al., 2019), we adopt the symmetric mean absolute percentage error (SMAPE), mean absolute scaled error (MASE) and overall weighted average (OWA) as the metrics, which can be calculated as follows:

$$\text{SMAPE} = \frac{200}{T}\sum_{i=1}^{T}\frac{|\mathbf{X}_i - \widehat{\mathbf{X}}_i|}{|\mathbf{X}_i| + |\widehat{\mathbf{X}}_i|}, \qquad \text{MAPE} = \frac{100}{T}\sum_{i=1}^{T}\frac{|\mathbf{X}_i - \widehat{\mathbf{X}}_i|}{|\mathbf{X}_i|},$$

$$\text{MASE} = \frac{1}{T}\sum_{i=1}^{T}\frac{|\mathbf{X}_i - \widehat{\mathbf{X}}_i|}{\frac{1}{T-p}\sum_{j=p+1}^{T}|\mathbf{X}_j - \mathbf{X}_{j-p}|}, \qquad \text{OWA} = \frac{1}{2}\left[\frac{\text{SMAPE}}{\text{SMAPE}_{\text{Naïve2}}} + \frac{\text{MASE}}{\text{MASE}_{\text{Naïve2}}}\right],$$

where $p$ is the periodicity of the data. $\widehat{\mathbf{X}}, \mathbf{X} \in \mathbb{R}^{T \times M}$ are the $M$ variables prediction results of length $T$ and corresponding ground truth. $\mathbf{X}_i$ means the $i$-th time step in the prediction result.

## C.3  IMPUTATION

**Implementation Details**   Our method is trained with the L2 loss, using the ADAM (Kingma & Ba, 2014) optimizer with an initial learning rate of $10^{-3}$. The default training process is 100 epochs with proper early stopping. The mean square error (MSE) and mean absolute error (MAE) are used as metrics. All the experiments are repeated 5 times with different seeds and the means of the metrics are reported as the final results.

We use a mask matrix $\mathbf{C} \in \mathbb{R}^{L \times M}$ to represent the missing values in input time series $\mathbf{X}_{in}$.

$$c_l^m = \begin{cases} 0 & \textit{if } x_l^m \textit{ is not observed} \\ 1 & \textit{otherwise} \end{cases}.$$

$\mathbf{X}_{in} \in \mathbb{R}^{L \times M}$ is the $M$ variables input time series of length $L$. And $L$ is set as 96 in imputation tasks. $x_l^m$ is the value at $l$-th timestep in the $m$-th univariate time series.

The input is the partially observed time series $\mathbf{C} \odot \mathbf{X}_{in}$ and the output is the imputed time series of the input. $\odot$ indicates the element-wise multiplication between the two tensors $\mathbf{X}_{in}$ and $\mathbf{C}$. And we only calculate MSE loss on masked tokens.

**Model Parameter**   By default, ModernTCN has 1 ModernTCN block with channel number $D = 128$ and FFN ratio $r = 1$. The kernel size is set as $large\ size = 71$ and $small\ size = 5$. Patch size and stride are set as $P = 1, S = 1$ to avoid mixing the masked and un-maskded tokens.

**Metric**   We adopt the mean square error (MSE) and mean absolute error (MAE) for imputation.

## C.4  CLASSIFICATION

**Implementation Details**   Our method is trained with the Cross Entropy Loss, using the ADAM (Kingma & Ba, 2014) optimizer with an initial learning rate of $10^{-3}$. The default training process is 30 epochs with proper early stopping. The classification accuracy is used as metrics. All the experiments are repeated 5 times with different seeds and the means of the metrics are reported as the final results.

**Model Parameter**   By default, ModernTCN has 2 ModernTCN blocks. The channel number $D$ is decided by $\min\{\max\{2^{\lceil \log M \rceil}, d_{\min}\}, d_{\max}\}$ ($d_{\min}$ is 32 and $d_{\max}$ is 512) following (Wu et al., 2023). The FFN ratio is $r = 1$. Patch size and stride are set as $P = 1, S = 1$ in the patchify embedding process.

**Metric**   For classification, we calculate the accuracy as metric.

## C.5  ANOMALY DETECTION

**Implementation Details**   We takes the classical reconstruction task and train it with the L2 loss. We use the ADAM (Kingma & Ba, 2014) optimizer with an initial learning rate of $3 \times 10^{-4}$. The default training process is 10 epochs with proper early stopping. We use the reconstruction error (MSE) as the anomaly criterion. The F1-Score is used as metric. All the experiments are repeated 5 times with different seeds and the means of the metrics are reported as the final results.

**Model Parameter**   By default, ModernTCN has 1 ModernTCN block. The channel number $D$ is decided by $\min\{\max\{2^{\lceil \log M \rceil}, d_{\min}\}, d_{\max}\}$ ($d_{\min}$ is 8 and $d_{\max}$ is 256) following (Wu et al., 2023). The FFN ratio is $r = 1$. The kernel size is set as $large\ size = 51$ and $small\ size = 5$. Patch size and stride are set as $P = 8, S = 4$ in the patchify embedding process.

Table 11: Impact of channel number. We conduct experiments with three different channel numbers ranging from $D = \{32, 64, 128\}$. A lower MSE or MAE indicates a better performance.

| Datasets | | ILI
Default $K=1, r=1$ | | | | ETTh1
Default $K=1, r=1$ | | | | Electricity
Default $K=1, r=8$ | | | |
|---|---|---|---|---|---|---|---|---|---|---|---|---|---|
| Prediction length | | 24 | 36 | 48 | 60 | 96 | 192 | 336 | 720 | 96 | 192 | 336 | 720 |
| $D=32$ | MSE | 1.772 | 1.598 | 1.725 | 1.976 | 0.377 | 0.412 | 0.395 | 0.453 | 0.132 | 0.145 | 0.159 | 0.193 |
| | MAE | 0.857 | 0.873 | 0.866 | 0.953 | 0.401 | 0.419 | 0.416 | 0.464 | 0.227 | 0.241 | 0.256 | 0.286 |
| $D=64$ | MSE | 1.347 | 1.250 | 1.388 | 1.774 | 0.368 | 0.405 | 0.391 | 0.450 | 0.129 | 0.143 | 0.161 | 0.191 |
| | MAE | 0.717 | 0.778 | 0.781 | 0.868 | 0.394 | 0.413 | 0.412 | 0.461 | 0.226 | 0.239 | 0.259 | 0.286 |
| $D=128$ | MSE | 2.010 | 1.751 | 1.378 | 1.806 | 0.364 | 0.402 | 0.387 | 0.449 | 0.135 | 0.147 | 0.168 | 0.196 |
| | MAE | 0.913 | 0.932 | 0.792 | 0.935 | 0.390 | 0.410 | 0.407 | 0.459 | 0.236 | 0.247 | 0.265 | 0.290 |

Table 12: Impact of FFN ratio. We conduct experiments with four different FFN ratios ranging from $r = \{1, 2, 4, 8\}$. A lower MSE or MAE indicates a better performance.

| Datasets | | ILI
Default $K=1, D=64$ | | | | Exchange
Default $K=1, D=64$ | | | | ETTh1
Default $K=1, D=64$ | | | | ETTm1
Default $K=3, D=64$ | | | |
|---|---|---|---|---|---|---|---|---|---|---|---|---|---|---|---|---|---|
| Prediction length | | 24 | 36 | 48 | 60 | 96 | 192 | 336 | 720 | 96 | 192 | 336 | 720 | 96 | 192 | 336 | 720 |
| $r=1$ | MSE | 1.347 | 1.250 | 1.388 | 1.774 | 0.080 | 0.166 | 0.307 | 0.656 | 0.368 | 0.405 | 0.391 | 0.450 | 0.294 | 0.335 | 0.369 | 0.419 |
| | MAE | 0.717 | 0.778 | 0.781 | 0.868 | 0.196 | 0.288 | 0.398 | 0.582 | 0.394 | 0.413 | 0.412 | 0.461 | 0.346 | 0.369 | 0.392 | 0.421 |
| $r=2$ | MSE | 2.083 | 1.480 | 1.940 | 1.758 | 0.080 | 0.167 | 0.306 | 0.657 | 0.368 | 0.407 | 0.392 | 0.450 | 0.291 | 0.332 | 0.365 | 0.417 |
| | MAE | 0.943 | 0.820 | 0.946 | 0.890 | 0.196 | 0.289 | 0.397 | 0.583 | 0.395 | 0.415 | 0.413 | 0.461 | 0.345 | 0.368 | 0.392 | 0.415 |
| $r=4$ | MSE | 1.877 | 1.589 | 1.401 | 2.042 | 0.080 | 0.167 | 0.308 | 0.659 | 0.367 | 0.411 | 0.395 | 0.453 | 0.292 | 0.333 | 0.365 | 0.416 |
| | MAE | 0.887 | 0.853 | 0.790 | 0.980 | 0.197 | 0.289 | 0.399 | 0.586 | 0.393 | 0.419 | 0.415 | 0.463 | 0.346 | 0.368 | 0.390 | 0.417 |
| $r=8$ | MSE | 2.038 | 1.657 | 1.577 | 1.937 | 0.080 | 0.167 | 0.309 | 0.660 | 0.369 | 0.409 | 0.401 | 0.459 | 0.292 | 0.332 | 0.365 | 0.416 |
| | MAE | 0.932 | 0.869 | 0.858 | 0.960 | 0.196 | 0.289 | 0.400 | 0.586 | 0.395 | 0.417 | 0.421 | 0.465 | 0.346 | 0.368 | 0.391 | 0.417 |

**Metric** For anomaly detection, we adopt the F1-score, which is the harmonic mean of precision and recall.

$$\text{F1-Score} = \frac{2 \times \text{Precision} \times \text{Recall}}{\text{Precision} + \text{Recall}}$$

# D MORE ABLATION STUDIES

We conduct more ablation studies in long-term forecasting tasks.

## D.1 RESULTS WITH DIFFERENT MODEL PARAMETERS

To see whether ModernTCN is sensitive to the choice of model parameters, we perform experiments with varying model parameters, including number of layers (number of ModernTCN blocks) ranging from $K = \{1, 2, 3, 4, 5\}$, channel number (dimension of hidden states) ranging from $D = \{32, 64, 128\}$ and FFN ratio ranging from $r = \{1, 2, 4, 8\}$. In general, except ILI dataset reveals high variance with different model parameter settings, other datasets are robust to the choice of model parameters. We conduct three experiments to figure out the impact of above three model parameters respectively. Detailed results are described in following paragraphs.

**Results with Different Channel Numbers** Table 11 shows the impact of different channel numbers $D$. Considering both the parameter efficiency and forecasting performance, we set the default channel number as $D = 64$. And the default channel number $D = 64$ works well for most of the datasets.

**Results with Different FFN Ratios** Table 12 shows the impact of different FFN Ratios $r$. Except for ILI dataset, our model is robust to the choice of the FFN ratio $r$ in other datasets. We recommend $r = 8$ for most of the datasets. And for small datasets like ETTh1, ETTh2, Exchange and ILI, we recommend a small FFN ratio like $r = 1$ to mitigate the possible overfitting and for better memory efficiency.

**Results with Different Numbers of Layers** Table 13 shows the impact of different numbers of layers (numbers of ModernTCN blocks) $K$. Considering both performance and efficiency, one ModernTCN block is enough for most of the datasets. But for bigger datasets like ETTm1 and ETTm2, we recommend to stack more ModernTCN blocks like $K = 3$ for better representation capability.

Table 13: Impact of layer number. We conduct experiments with five different layer numbers ranging from $K = \{1, 2, 3, 4, 5\}$. A lower MSE or MAE indicates a better performance.

| Datasets | | ETTh1 Default $D = 64, r = 1$ | | | | Electricity Default $D = 64, r = 8$ | | | | ETTm1 Default $D = 64, r = 8$ | | | |
|---|---|---|---|---|---|---|---|---|---|---|---|---|---|
| Prediction length | | 96 | 192 | 336 | 720 | 96 | 192 | 336 | 720 | 96 | 192 | 336 | 720 |
| $K = 1$ | MSE | 0.368 | 0.405 | 0.391 | 0.450 | 0.129 | 0.143 | 0.161 | 0.191 | 0.296 | 0.338 | 0.369 | 0.419 |
| | MAE | 0.394 | 0.413 | 0.412 | 0.461 | 0.226 | 0.239 | 0.259 | 0.286 | 0.346 | 0.370 | 0.392 | 0.420 |
| $K = 2$ | MSE | 0.367 | 0.405 | 0.393 | 0.450 | 0.129 | 0.143 | 0.161 | 0.191 | 0.298 | 0.337 | 0.367 | 0.417 |
| | MAE | 0.393 | 0.413 | 0.413 | 0.460 | 0.226 | 0.239 | 0.259 | 0.286 | 0.348 | 0.370 | 0.391 | 0.418 |
| $K = 3$ | MSE | 0.369 | 0.409 | 0.392 | 0.450 | 0.130 | 0.146 | 0.163 | 0.192 | 0.292 | 0.332 | 0.365 | 0.416 |
| | MAE | 0.394 | 0.417 | 0.412 | 0.460 | 0.227 | 0.243 | 0.260 | 0.286 | 0.346 | 0.368 | 0.391 | 0.417 |
| $K = 4$ | MSE | 0.368 | 0.406 | 0.394 | 0.451 | 0.130 | 0.146 | 0.163 | 0.192 | 0.298 | 0.334 | 0.366 | 0.416 |
| | MAE | 0.393 | 0.414 | 0.414 | 0.461 | 0.226 | 0.243 | 0.260 | 0.287 | 0.348 | 0.368 | 0.391 | 0.417 |
| $K = 5$ | MSE | 0.368 | 0.411 | 0.394 | 0.452 | 0.130 | 0.149 | 0.163 | 0.193 | 0.294 | 0.333 | 0.366 | 0.416 |
| | MAE | 0.394 | 0.418 | 0.414 | 0.461 | 0.227 | 0.247 | 0.261 | 0.288 | 0.346 | 0.368 | 0.393 | 0.417 |

### D.2 IMPACT OF INPUT LENGTH AND PATCHING SETTINGS

**Impact of Input Length.** Since a longer input length indicates more historical information an algorithm can utilize in time series forecasting, a model with strong ability to capture long-term temporal dependency should perform better when input length increases (Zeng et al., 2022; Wang et al., 2023; Nie et al., 2023). To validate our model, we conduct experiments with different input lengths under the same prediction length. As shown in Figure 7, in general, our model gains performance improvement with increasing input length, indicating our model can effectively extract useful information from longer history and capture long-term dependency. However, some Transformer-based models (Wu et al., 2021; Zhou et al., 2021; Vaswani et al., 2017) suffer from performance degradation with increasing input length owing to the repeated short-term patterns according to (Zhou et al., 2021).

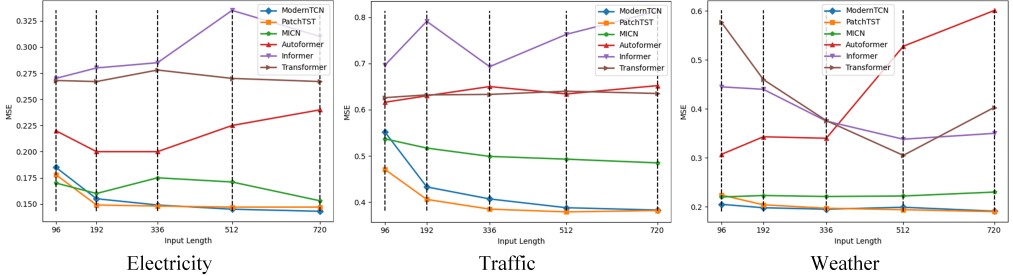

| Electricity | Traffic | Weather |

Figure 7: The MSE results with different input lengths and same prediction length(192 time steps).

**Impact of Patch size and Stride** To verify the impact of patch size $P$ and stride $S$, we perform experiments with two patching modes ($P = S$ and $P = 2 \times S$) and two different $S$. Results are shown on Table 14. In general, the performance doesn't vary significantly with different $P$ and $S$, indicating the robustness of our model against these two hyperparameters. The ideal $P$ and $S$ may vary from different datasets. We recommend $P = 8$ and $S = 4$ as general good choice for most of the datasets.

Table 14: Impact of patch size and stride. We compare four patching settings. A lower MSE or MAE indicates a better performance.

| Datasets | | ETTh1 Default $D = 64, r = 1, K = 1$ | | | | ETTm1 Default $D = 64, r = 8, K = 3$ | | | | Electricity Default $D = 64, r = 8, K = 1$ | | | |
|---|---|---|---|---|---|---|---|---|---|---|---|---|---|
| Prediction length | | 96 | 192 | 336 | 720 | 96 | 192 | 336 | 720 | 96 | 192 | 336 | 720 |
| $P = 4, S = 4$ | MSE | 0.369 | 0.409 | 0.391 | 0.449 | 0.295 | 0.336 | 0.369 | 0.417 | 0.130 | 0.145 | 0.160 | 0.191 |
| | MAE | 0.395 | 0.417 | 0.412 | 0.460 | 0.348 | 0.372 | 0.393 | 0.418 | 0.226 | 0.240 | 0.259 | 0.286 |
| $P = 8, S = 4$ | MSE | 0.368 | 0.405 | 0.391 | 0.450 | 0.292 | 0.332 | 0.365 | 0.416 | 0.129 | 0.143 | 0.161 | 0.191 |
| | MAE | 0.394 | 0.413 | 0.412 | 0.461 | 0.346 | 0.368 | 0.391 | 0.417 | 0.226 | 0.239 | 0.259 | 0.286 |
| $P = 8, S = 8$ | MSE | 0.377 | 0.414 | 0.400 | 0.455 | 0.297 | 0.335 | 0.372 | 0.420 | 0.132 | 0.147 | 0.168 | 0.195 |
| | MAE | 0.402 | 0.421 | 0.421 | 0.466 | 0.349 | 0.371 | 0.395 | 0.421 | 0.227 | 0.241 | 0.263 | 0.290 |
| $P = 16, S = 8$ | MSE | 0.378 | 0.413 | 0.398 | 0.456 | 0.299 | 0.336 | 0.372 | 0.419 | 0.133 | 0.146 | 0.168 | 0.196 |
| | MAE | 0.403 | 0.420 | 0.419 | 0.466 | 0.350 | 0.371 | 0.395 | 0.420 | 0.228 | 0.240 | 0.263 | 0.290 |

## E UNIVARIATE LONG-TERM FORECASTING RESULTS

Here we provide the univariate long-term forecasting results on 4 ETT datasets. There is a target feature *oil temperature* within those datasets, which is the univariate time series that we are trying to forecast. Since it's a univariate tasks, we mainly focus on capturing cross-time information and don't need to capture the cross-variable information. Thus, we remove the cross-variable component in ModernTCN. As shown in Table 15, thanks to the larger ERF and better temporal modeling ability in DWConv, our ModernTCN can achieve comparable performance with the state-of-the-art Transformer-based model PatchTST(2023) and MLP-based model DLinear(2022) in univariate forecasting tasks.

Table 15: Univariate long-term forecasting results on ETT datasets. Following PatchTST (2023) and DLinear (2022), input length is fixed as 336 and prediction lengths are $T \in \{96, 192, 336, 720\}$. The best results are in **bold**.

| Models | | ModernTCN (Ours) | | PatchTST (2023) | | DLinear (2022) | | FEDformer (2022) | | Autoformer (2021) | | Informer (2021) | | LogTrans (2019a) | |
|---|---|---|---|---|---|---|---|---|---|---|---|---|---|---|---|
| Metric | | MSE | MAE | MSE | MAE | MSE | MAE | MSE | MAE | MSE | MAE | MSE | MAE | MSE | MAE |
| ETTh1 | 96 | **0.055** | **0.179** | **0.055** | **0.179** | 0.056 | 0.180 | 0.079 | 0.215 | 0.071 | 0.206 | 0.193 | 0.377 | 0.283 | 0.468 |
| | 192 | **0.070** | 0.205 | 0.071 | 0.205 | 0.071 | **0.204** | 0.104 | 0.245 | 0.114 | 0.262 | 0.217 | 0.395 | 0.234 | 0.409 |
| | 336 | **0.074** | **0.214** | 0.076 | 0.220 | 0.098 | 0.244 | 0.119 | 0.270 | 0.107 | 0.258 | 0.202 | 0.381 | 0.386 | 0.546 |
| | 720 | **0.086** | **0.232** | 0.087 | 0.236 | 0.087 | 0.359 | 0.142 | 0.299 | 0.126 | 0.283 | 0.183 | 0.355 | 0.475 | 0.629 |
| ETTh2 | 96 | **0.124** | **0.274** | 0.129 | 0.282 | 0.131 | 0.279 | 0.128 | 0.271 | 0.153 | 0.306 | 0.213 | 0.373 | 0.217 | 0.379 |
| | 192 | **0.164** | **0.321** | 0.168 | 0.328 | 0.176 | 0.329 | 0.185 | 0.330 | 0.204 | 0.351 | 0.227 | 0.387 | 0.281 | 0.429 |
| | 336 | **0.171** | **0.336** | **0.171** | **0.336** | 0.209 | 0.367 | 0.231 | 0.378 | 0.246 | 0.389 | 0.242 | 0.401 | 0.293 | 0.437 |
| | 720 | 0.228 | 0.384 | **0.223** | **0.380** | 0.276 | 0.426 | 0.278 | 0.420 | 0.268 | 0.409 | 0.291 | 0.439 | 0.218 | 0.387 |
| ETTm1 | 96 | **0.026** | **0.121** | **0.026** | **0.121** | 0.028 | 0.123 | 0.033 | 0.140 | 0.056 | 0.183 | 0.109 | 0.277 | 0.049 | 0.171 |
| | 192 | 0.040 | 0.152 | **0.039** | **0.150** | 0.045 | 0.156 | 0.058 | 0.186 | 0.081 | 0.216 | 0.151 | 0.310 | 0.157 | 0.317 |
| | 336 | **0.053** | **0.173** | **0.053** | **0.173** | 0.061 | 0.182 | 0.084 | 0.231 | 0.076 | 0.218 | 0.427 | 0.591 | 0.289 | 0.459 |
| | 720 | **0.073** | **0.206** | **0.073** | **0.206** | 0.080 | 0.210 | 0.102 | 0.250 | 0.110 | 0.267 | 0.438 | 0.586 | 0.430 | 0.579 |
| ETTm2 | 96 | 0.065 | 0.183 | 0.065 | 0.186 | **0.063** | **0.183** | 0.067 | 0.198 | 0.065 | 0.189 | 0.088 | 0.225 | 0.075 | 0.208 |
| | 192 | 0.095 | 0.232 | 0.093 | 0.231 | **0.092** | **0.227** | 0.102 | 0.245 | 0.118 | 0.256 | 0.132 | 0.283 | 0.129 | 0.275 |
| | 336 | 0.119 | 0.261 | 0.120 | 0.265 | **0.119** | **0.261** | 0.130 | 0.279 | 0.154 | 0.305 | 0.180 | 0.336 | 0.154 | 0.302 |
| | 720 | 0.173 | 0.323 | **0.171** | **0.322** | 0.175 | 0.320 | 0.178 | 0.325 | 0.182 | 0.335 | 0.300 | 0.435 | 0.160 | 0.321 |
| 1st Count | | **21** | | 16 | | 7 | | 0 | | 0 | | 0 | | 0 | |

# F  Technique to Better Train a Large Kernel

## F.1  Introduction about Structural Re-parameterization

According to (Ding et al., 2022; Liu et al., 2022b), we can use the Structural Re-parameterization technique to better train a large kernel convolution. **In training phase**, additional Batch normalization (BN) layers (Ioffe & Szegedy, 2015) are used following the depth-wise convolution layers to form convolution-BN branches for a better training result. Then the $large\ size$ depth-wise convolution-BN branch is trained with a parallel $small\ size$ depth-wise convolution-BN branch to make up the optimization issue of large kernel. **After training**, each of these two branches is transformed into a single depth-wise convolution layer by fusing the BN parameters into the convolution kernels (Ding et al., 2021). Then the small kernel is zero-padded on both side to $large\ size$. After two kernels aligned to the same size, the two depth-wise convolution layers add up to merge into a single large kernel depth-wise convolution layer. Then the fused single large kernel depth-wise convolution can be used in inference phase. It's notable that the resultant large kernel model is totally equivalent to the model in training phase but no longer has small kernels. See Figure 8 for an example of Structural Re-parameterization after training.

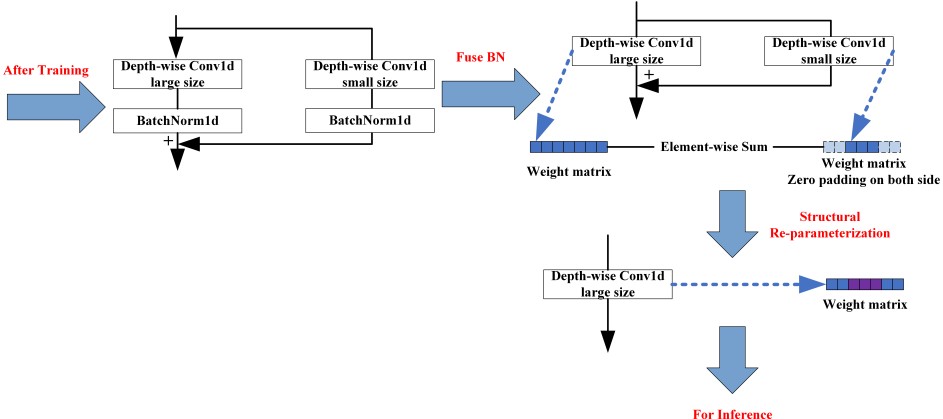

Figure 8: An example of Structural Re-parameterization.

Table 16: Impact of kernel size. We compare three different kernel sizes ranging from small to large. A lower MSE or MAE indicates a better performance. The best results are highlighted in **blod**.

| Datasets | | ILI | | | | ETTh1 | | | | Electricity | | | |
|---|---|---|---|---|---|---|---|---|---|---|---|---|---|
| Prediction length | | 24 | 36 | 48 | 60 | 96 | 192 | 336 | 720 | 96 | 192 | 336 | 720 |
| kernel size = 3 | MSE | 1.906 | 1.546 | 1.754 | 1.893 | 0.381 | 0.416 | 0.403 | 0.460 | 0.143 | 0.155 | 0.175 | 0.203 |
| | MAE | 0.862 | 0.841 | 0.891 | 0.900 | 0.405 | 0.423 | 0.419 | 0.470 | 0.237 | 0.249 | 0.270 | 0.299 |
| kernel size = 31 | MSE | 1.687 | 1.486 | 1.376 | 1.855 | **0.367** | **0.405** | **0.389** | **0.449** | 0.133 | 0.147 | **0.159** | 0.193 |
| | MAE | 0.848 | 0.855 | 0.797 | 0.929 | **0.393** | **0.413** | **0.410** | **0.460** | 0.228 | 0.243 | **0.257** | 0.288 |
| kernel size = 51 | MSE | **1.347** | **1.250** | **1.388** | **1.774** | 0.368 | **0.405** | 0.391 | 0.450 | **0.129** | **0.143** | 0.161 | **0.191** |
| | MAE | **0.717** | **0.778** | **0.781** | **0.868** | 0.394 | **0.413** | 0.412 | 0.461 | **0.226** | **0.239** | 0.259 | **0.286** |

Table 17: Impact of small kernel sizes. A lower MSE or MAE indicates a better performance.

| Datasets | | ETTh1 | | | | ETTm1 | | | | Electricity | | | |
|---|---|---|---|---|---|---|---|---|---|---|---|---|---|
| Prediction length | | 96 | 192 | 336 | 720 | 96 | 192 | 336 | 720 | 96 | 192 | 336 | 720 |
| $small\ size = 1$ | MSE | 0.369 | 0.406 | 0.390 | 0.450 | 0.296 | 0.346 | 0.373 | 0.441 | 0.130 | 0.145 | 0.160 | 0.191 |
| | MAE | 0.395 | 0.414 | 0.410 | 0.461 | 0.348 | 0.377 | 0.397 | 0.432 | 0.226 | 0.240 | 0.259 | 0.286 |
| $small\ size = 3$ | MSE | 0.369 | 0.407 | 0.391 | 0.450 | 0.295 | 0.336 | 0.369 | 0.423 | 0.129 | 0.143 | 0.161 | 0.191 |
| | MAE | 0.395 | 0.415 | 0.412 | 0.461 | 0.348 | 0.371 | 0.393 | 0.425 | 0.226 | 0.239 | 0.259 | 0.286 |
| $small\ size = 5$ | MSE | 0.368 | 0.405 | 0.391 | 0.450 | 0.292 | 0.332 | 0.365 | 0.416 | 0.129 | 0.143 | 0.161 | 0.191 |
| | MAE | 0.394 | 0.413 | 0.412 | 0.461 | 0.346 | 0.368 | 0.391 | 0.417 | 0.226 | 0.239 | 0.259 | 0.286 |
| $small\ size = 7$ | MSE | 0.368 | 0.405 | 0.390 | 0.449 | 0.295 | 0.341 | 0.373 | 0.419 | 0.133 | 0.146 | 0.168 | 0.196 |
| | MAE | 0.394 | 0.413 | 0.411 | 0.460 | 0.347 | 0.376 | 0.397 | 0.421 | 0.228 | 0.240 | 0.263 | 0.290 |
| $small\ size = 9$ | MSE | 0.368 | 0.406 | 0.391 | 0.450 | 0.296 | 0.351 | 0.375 | 0.428 | 0.133 | 0.146 | 0.168 | 0.196 |
| | MAE | 0.394 | 0.414 | 0.412 | 0.462 | 0.348 | 0.380 | 0.398 | 0.424 | 0.228 | 0.240 | 0.263 | 0.290 |

## F.2 ABLATION OF STRUCTURAL RE-PARAMETERIZATION

### F.2.1 IMPACT OF LARGE KERNEL SIZE

According to (Ding et al., 2022), a large kernel size is the key to obtain a large ERF in 2D convolution. To verify whether this finding still works on 1D convolution and to figure out the impact of kernel size, we perform experiments with 3 different kernel sizes ranging from small to large on 3 datasets. Results on Table 16 show that increasing the kernel size leads to performance improvement. The experiment results indicate that directly enlarging the kernel size in 1D convolution layer and training it with Structural Re-parameterization technique can effectively increase ERF and help convolution layer to better capture temporal dependency.

### F.2.2 RESULT WITH DIFFERENT SMALL KERNELS

According to (Ding et al., 2022; Liu et al., 2022b), adding a parallel Structural Re-parameterization branch with a small kernel can help to train the large kernel convolution layer. To further figure out the impact of different small kernel sizes, we perform experiments with five different small kernel sizes ranging from 1 to 9. As shown in Table 17, the performance is robust to the choice of small kernel sizes as long as they're much smaller than the large kenrel. And $small\ size = 5$ is a general good choice.

### F.2.3 RESULT WITH MORE PARALLEL BRANCHES

In Structural Re-parameterization technique, we usually add one additional Structural Re-parameterization branch paralleled to the large kernel convolution branch to make up its optimization issue. Here we perform experiments with more branches to see the impact of parallel branches number. As shown in Table 18, the performance is robust to the choice of parallel branches numbers. Considering both performance and efficiency, we only add one Structural Re-parameterization branch in our main experiments.

Table 18: Impact of more parallel branches. A lower MSE or MAE indicates a better performance.

| Datasets | | ETTh1 | | | | ETTm1 | | | | Electricity | | | |
|---|---|---|---|---|---|---|---|---|---|---|---|---|---|
| Prediction length | | 96 | 192 | 336 | 720 | 96 | 192 | 336 | 720 | 96 | 192 | 336 | 720 |
| $51-5$ | MSE | 0.368 | 0.405 | 0.391 | 0.450 | 0.292 | 0.332 | 0.365 | 0.416 | 0.129 | 0.143 | 0.161 | 0.191 |
| | MAE | 0.394 | 0.413 | 0.412 | 0.461 | 0.346 | 0.368 | 0.391 | 0.417 | 0.226 | 0.239 | 0.259 | 0.286 |
| $51-5-1$ | MSE | 0.368 | 0.406 | 0.396 | 0.451 | 0.292 | 0.338 | 0.374 | 0.425 | 0.130 | 0.144 | 0.162 | 0.190 |
| | MAE | 0.394 | 0.414 | 0.417 | 0.462 | 0.345 | 0.372 | 0.397 | 0.421 | 0.226 | 0.240 | 0.261 | 0.286 |
| $51-5-3$ | MSE | 0.368 | 0.406 | 0.393 | 0.451 | 0.293 | 0.340 | 0.374 | 0.423 | 0.131 | 0.144 | 0.160 | 0.192 |
| | MAE | 0.394 | 0.414 | 0.413 | 0.463 | 0.346 | 0.373 | 0.399 | 0.421 | 0.227 | 0.239 | 0.259 | 0.288 |
| $51-5-7$ | MSE | 0.368 | 0.406 | 0.393 | 0.452 | 0.291 | 0.343 | 0.375 | 0.420 | 0.131 | 0.144 | 0.161 | 0.191 |
| | MAE | 0.394 | 0.414 | 0.413 | 0.462 | 0.345 | 0.374 | 0.398 | 0.422 | 0.226 | 0.240 | 0.260 | 0.287 |
| $51-5-7-3-1$ | MSE | 0.373 | 0.406 | 0.395 | 0.450 | 0.291 | 0.340 | 0.371 | 0.424 | 0.131 | 0.144 | 0.163 | 0.191 |
| | MAE | 0.399 | 0.414 | 0.415 | 0.461 | 0.344 | 0.371 | 0.394 | 0.422 | 0.226 | 0.240 | 0.262 | 0.286 |

# G  DETAILS OF MODERNTCN BLOCKS

## G.1  BACKGROUND

**Group Convolution**    Group convolution (Krizhevsky et al., 2017; Xie et al., 2017; Zhang et al., 2018) divides the convolution channels into separate groups. Only the channels in the same group can interact with each other while the channels in different groups are independent.

**Depthwise Separable Convolution**    Depthwise convolution (Howard et al., 2017) can be seen as a special group convolution in which the number of groups is equal to the number of channels. Therefore, all channels in depthwise convolution are independent. The depthwise convolution only mixes information among tokens across the temporal dimension.

Since depthwise convolution layer is totally channel-independent. It can not combine channels to create new representations. Therefore, the pointwise convolution layers, which are in charge of mixing channel information to create new representations, should be used following the depthwise convolution as a complementary. The kernel size of pointwise convolution is 1. Therefore it is applied to each token independently and only mixes information among channels.

**Modern Convolution**    Modern convolution (Liu et al., 2022d; Ding et al., 2022; Liu et al., 2022b) is a new convolution paradigm inspired by Transformer. The basic components in a modern convolution block are depthwise and pointwise convolution layers. Inspired by the architectural designs in Transformer, the depthwise and pointwise convolution layers are organized into a similar structure to Transformer block, which is shown in Figure 2 (a) and (b).

As shown in Figure 2 (b), the DWConv is a depthwise convolution layer which is responsible for learning the temporal information among tokens on a per-channel basis. It plays the same role as the self-attention module in Transformer. Meanwhile, a large kernel is adopted in DWConv to catch up with the gloabl effective receptive field in Transformer.

ConvFFN consists of two point-wise convolution layers (PWConvs) and adopts an inverted bottleneck structure, where the hidden channel of the ConvFFN block is $r$ times wider than the input channel (Figure 2 (c)). ConvFFN plays the same role as the FFN module in Transformer blocks, which is to learn the new representation of each token independently.

## G.2  DETAILS OF MODERNTCN BLOCK DESIGNS

The input to the $i$-th ModernTCN block is $\mathbf{Z}_i \in \mathbb{R}^{M \times D \times N}$, where $M$, $D$ and $N$ are the size of variable dimension, feature dimension and temporal dimension respectively. We merge the feature dimension and variable dimension before feeding the embedded time series into the ModernTCN block. Therefore the convolution channel number is $M \times D$.

As shown in Figure 2 (d), in the ModernTCN block, DWConv is a depthwise convolution layer which maps $M \times D$ input channels to $M \times D$ output channels. The group number in DWConv is set as $M \times D$ to make each channel independent. Therefore, DWConv is both variable and feature independent. It only mixes information across temporal dimension. And we set a large kernel size for DWConv to enlarge its effective receptive field and improve its temporal modeling ability.

The ConvFFN1 and ConvFFN2 in ModernTCN block are the decoupled version of ConvFFN based on the idea of group convolution. We replace the two pointwise convolution layers in ConvFFN with two group pointwise convolution layers with group number as $M$ to obtain ConvFFN1. Similarly, we replace the two pointwise convolution layers in ConvFFN with two group pointwise convolution layers with group number as $D$ to obtain ConvFFN2.

In details, the input channel number in ConvFFN1 is $M \times D$. By setting group number as $M$, the $M \times D$ input channels will be divided into $M$ groups. And only the $D$ features in the same group can interact with each other. Since each group represents a variable, it means that only the features of the same variable can interact with each other to create new feature representation while the features of different variables are independent. Therefore ConvFFN1 can learn the new feature representation for each variable independently.

The same goes for ConvFFN2. After permute operation, the input channel number in ConvFFN2 is $D \times M$. By setting group number as $D$, the $D \times M$ input channels will be divide into $D$ groups. And only the $M$ variables in the same group can interact with each other. Therefore ConvFFN2 can capturing the cross-variable dependency per feature.

### G.3 Parameters of Convolution Layers in ModernTCN Block

We provide the parameters of convolution layers in ModernTCN block in Table 19.

Table 19: Parameters of convolution layers in ModernTCN block.

| Module | | input channels | output channels | kernel size | stride | group number |
|---|---|---|---|---|---|---|
| DWConv | | $M \times D$ | $M \times D$ | $large\ size$ | 1 | $M \times D$ |
| ConvFFN1 | First PWConv | $M \times D$ | $r \times M \times D$ | 1 | 1 | $M$ |
| | Second PWConv | $r \times M \times D$ | $M \times D$ | 1 | 1 | $M$ |
| ConvFFN2 | First PWConv | $D \times M$ | $r \times D \times M$ | 1 | 1 | $D$ |
| | Second PWConv | $r \times D \times M$ | $D \times M$ | 1 | 1 | $D$ |

## H Details of Ablation Study

In Section 5.3, we provide an ablation study on our model designs. In this appendix, we provide more details about this ablation study.

### H.1 Detail of Different Embedding Settings

In Section 3.2, we design the patchify variable-independent embedding to maintain the variable dimension, replacing the common patchify variable-mixing embedding which will lead to the discard of variable dimension in the embedded series. The difference of the two embeddings is shown on Figure 9 and introduced as follows:

- In variable-independent embedding, we treat each univariate time series in the multi-variate time series as an individual sample with 1 feature and embed them independently.

- In variable-mixing embedding, we treat the whole multi-variate time series as a single sample with M features.

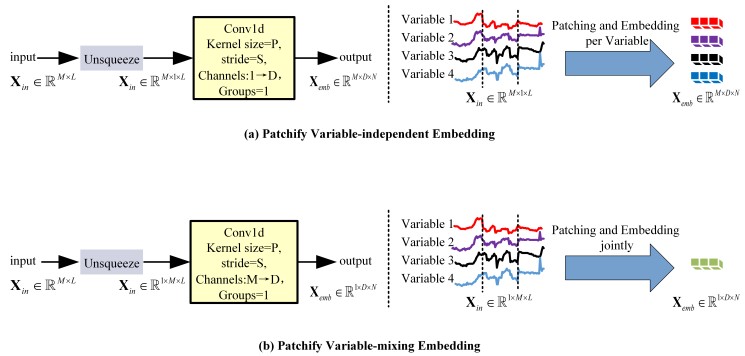

(a) Patchify Variable-independent Embedding

(b) Patchify Variable-mixing Embedding

Figure 9: Different embedding methods in ablation study.

**The essential difference behind these two embedding methods is the opinion of whether we should maintain the variable dimension when analysing multivariate time series.** Since applying the variable-mixing embedding to the multivariate time series will discard its variable dimension, we denote it as the setting *Discard Variable Dimension*. To verifies our opinion to maintain the variable dimension, we use the setting *Discard Variable Dimension* for comparision. In this setting, we apply

variable-mixing embedding to the multivariate input time series to discard its variable dimension. Then the embedded series is fed into a backbone stacked by convolution blocks like Figure 10 (a) to learn representation. As shown in Table 4, the setting *Discard Variable Dimension* leads to significant performance degradation, which verifies the effectiveness of our variable-independent embedding design and highlights the importance to maintain the variable dimension.

## H.2 Detail of Different Block Design Settings

Then we conduct experiments to study the impact of different block designs. All block design settings we used are shown in Figure 10. Since variable-mixing embedding will lead to severe performance degradation, we conduct these experiments with our variable-independent embedding.

Given that variable-independent embedding will maintain the variable dimension, we need to make our structure able to capture information from the additional variable dimension. A direct and naive way is to jointly learn the dependency among features and variables only by a single ConvFFN, which is denoted as *Undecoupled ConvFFN*. But such setting leads to higher computational complexity and worse performance, which is shown in Table 4. In contrast, the combination of our decoupled two ConvFFNs (denoted as *ConvFFN1+ConvFFN2*) achieve the best performance, which proves the necessity and effectiveness of our decouple modification to ConvFFN module.

We further study how much ConvFFN1 and ConvFFN2 contribute to the forecasting respectively. *Only ConvFFN1* can learn each variable's deep feature representation independently but doesn't take cross-variable dependency into consideration. *Only ConvFFN2* can only capture the cross-variable dependency per feature. But to each variable, it omits to learn their deep feature representations. As a result, the performance decreases on both settings.

And we also include settings like *ConvFFN1+ConvFFN1* and *ConvFFN2+ConvFFN2* to eliminate the impact of the number of ConvFFNs. The results show that the combination of ConvFFN1 and ConvFFN2 is the key to performance improvement, not the the number of ConvFFNs.

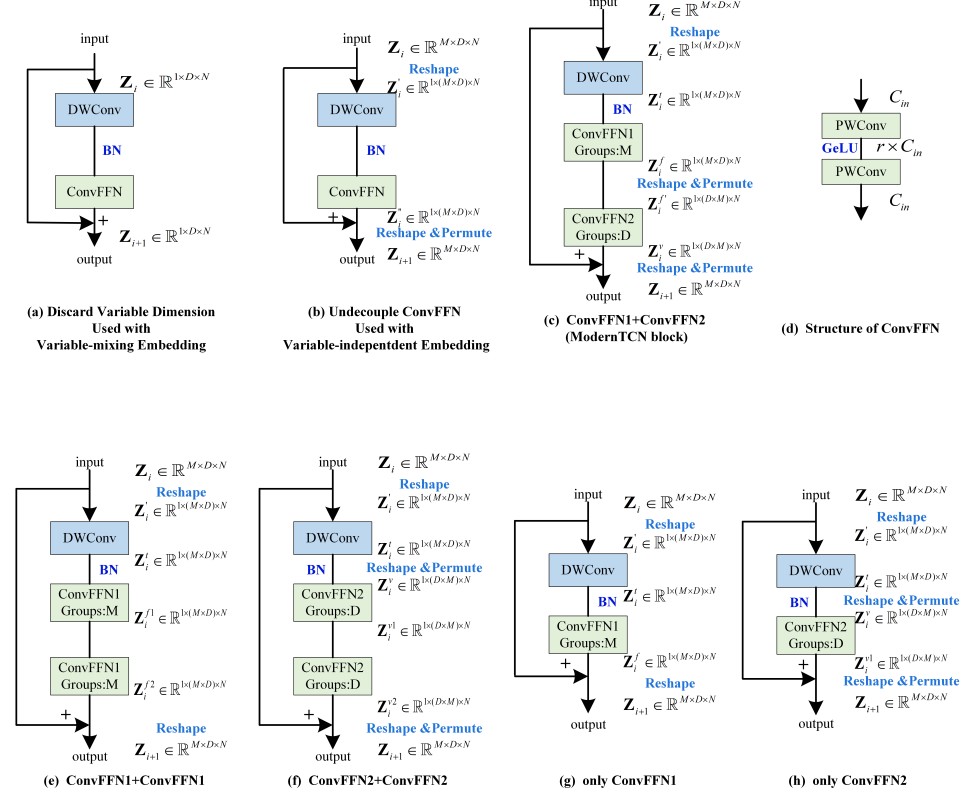

Figure 10: Different block designs in ablation study.

# I   MORE COMPARISON WITH TIMESNET IN TERMS OF ERF

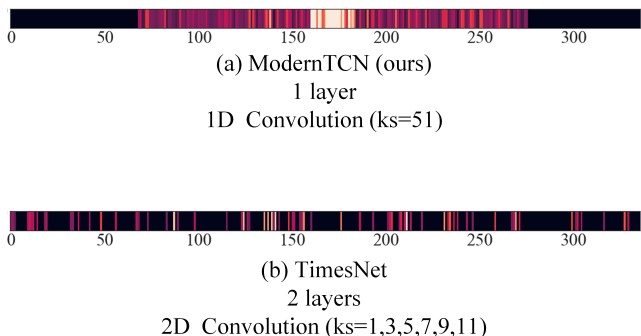

(a) ModernTCN (ours)
1 layer
1D  Convolution (ks=51)

(b) TimesNet
2 layers
2D  Convolution (ks=1,3,5,7,9,11)

Figure 11: The Effective Receptive Field (ERF) of ModernTCN and TimesNet (2023).  A more widely distributed light area indicates a larger ERF.

ModernTCN and TimesNet successfully enlarge the ERFs. But the ERFs of these two models have quite different property (Figure 11).

The ERFs of ModernTCN and other 1D convolution-based models are concentrated at the middle point and continuously expand to both ends, which means that the final representation of the middle point is highly related to its adjacent points. Such phenomenon demonstrates the locality, which is a common property of convolution.

The ERF of ModernTCN can expand to a wider range.  Therefore ModernTCN has larger ERF than other 1D convolution-based models.  But at the same time, the ERF of ModernTCN is still concentrated at the middle point.  So ModernTCN is able to capture long-term dependency while focusing on the local context.

However, the ERF of TimesNet is discrete and not concentrated at the middle point, which doesn't reflect the locality of convolution. This is because the additional 2D data transformation in TimesNet also influences the pattern of ERF. In 2D data transformation, the time series is divided into several segments and further rearranged in 2D space based on periods. In this process, a time point may be separated from its adjacent points, thereby losing continuity and locality. As a result, the ERF and performance of TimesNet are also highly (or even mainly) related to its special 2D data transformation, but not just depend on convolution.

In summary, influenced by its special 2D data transformation, TimesNet' ERF is of quite different property from other 1D convolution-based models' and doesn't reflect the locality of convolution. Although both ModernTCN and TimesNet can successfully enlarge the ERFs, ModernTCN can also maintain the locality when enlarging the ERF, therefore providing better performance than TimesNet.

# J  APPLY CROSS-VARIABLE COMPONENT TO VARIABLE-INDEPENDENT MODELS

We apply our cross-variable component (ConvFFNs) to some variable-independent models like DLinear (2022) and PatchTST (2023). We make some necessary modifications to our cross-variable component to adapt it to these two models, which is summarized as follows and shown on Figure 12.

- For DLinear which only has two individual Linear layers (one for trend part and the other for seasonal part), we add an additional ConvFFN module before the Linear layer. This additional ConvFFN module can mix information across variable dimension to incorporate the cross-variable information.
- For PatchTST, we replace the original FFN module in PatchTST block with our ConvFFNs but with some difference in residual connections to align with the block design in Transformer.

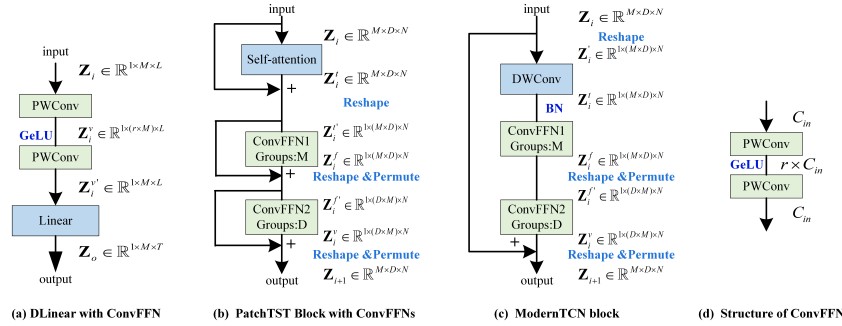

Figure 12: Applying cross-variable component to DLinear (2022) and PatchTST (2023).

We conduct experiments in imputation tasks, where cross-variable dependency plays an important role. We report the performance promotion of each model in Table 20. In imputation tasks, equipped with our cross-variable component achieves averaged 28.7% promotion on PatchTST and 19.7% promotion on DLinear. The result validates that our cross-variable component can be used on top of other variable-independent models, helping them incorporate the cross-variable information and improving their performance.

Table 20: Performance promotion by applying our cross-variable component to PatchTST and DLinear in imputation tasks. We report the averaged MSE/MAE of all four mask ratios and the relative MSE reduction ratios (Promotion) by our cross-variable component.

| Dataset | ETTm1 | | ETTm2 | | ETTh1 | | ETTh2 | | Electricity | | Weather | |
|---|---|---|---|---|---|---|---|---|---|---|---|---|
| Model | MSE | MAE | MSE | MAE | MSE | MAE | MSE | MAE | MSE | MAE | MSE | MAE |
| PatchTST | 0.045 | 0.133 | 0.028 | 0.098 | 0.133 | 0.236 | 0.066 | 0.164 | 0.091 | 0.209 | 0.033 | 0.057 |
| **+ Ours** | **0.026** | **0.107** | **0.020** | **0.087** | **0.083** | **0.192** | **0.048** | **0.145** | **0.069** | **0.177** | **0.029** | **0.050** |
| Promotion | 42.2% | | 28.6% | | 37.6% | | 27.3% | | 24.2% | | 12.1% | |
| DLinear | 0.093 | 0.206 | 0.096 | 0.208 | 0.201 | 0.306 | 0.142 | 0.259 | 0.132 | 0.260 | 0.052 | 0.110 |
| **+ Ours** | **0.065** | **0.175** | **0.089** | **0.199** | **0.115** | **0.236** | **0.136** | **0.251** | **0.120** | **0.253** | **0.039** | **0.091** |
| Promotion | 30.1% | | 7.3% | | 42.8% | | 4.2% | | 9.0% | | 25.0% | |

# K SHORT-TERM FORECASTING RESULTS ON MULTIVARIATE TIME SERIES DATASETS

We conduct short-term forecasting on 9 multivariate time series datasets, including Weather (Wetterstation), Traffic (PeMS), Electricity (UCI), Exchange (Lai et al., 2018a), ILI (CDC) and four ETT datasets (Zhou et al., 2021).

The experiment details are as follows:

- We choose the prediction lengths as $T \in \{6, 12, 18\}$, which meets the prediction length in M4 datasets. And following M4 short-term forecasting tasks, we set input length $L$ to be 2 times of prediction length $T$.
- We choose models that perform well in multivariate datasets (DLinear, RLinear, RMLP and PatchTST) or perform well in short-term forecasting tasks (PatchTST, CARD and TimesNet) as strong baselines.
- All models follow their official configurations on above datasets, we only change the input lengths and prediction lengths.
- We calculate MSE and MAE of the multivariate prediction results as metric.

The results are shown in Table 21. ModernTCN can outperform above competitive baselines in most cases, indicating that ModernTCN can better use limited input information to provide better forecasting results.

Table 21: Short-term forecasting on multivariate time series datasets. We set prediction lengths as $T \in \{6, 12, 18\}$ and set input length $L$ to be 2 times of prediction length $T$. The best results are in **bold**.

| Models | | ModernTCN (Ours) | | PatchTST (2023) | | CARD (2023) | | TimesNet (2023) | | DLinear (2022) | | RLinear (2023a) | | RMLP (2023a) | |
|---|---|---|---|---|---|---|---|---|---|---|---|---|---|---|---|
| Metric | | MSE | MAE | MSE | MAE | MSE | MAE | MSE | MAE | MSE | MAE | MSE | MAE | MSE | MAE |
| ETTh1 | 6 | **0.505** | **0.423** | 0.776 | 0.545 | 0.754 | 0.530 | 0.785 | 0.558 | 0.941 | 0.649 | 1.184 | 0.676 | 0.682 | 0.504 |
| | 12 | **0.316** | **0.357** | 0.438 | 0.431 | 0.383 | 0.397 | 0.412 | 0.424 | 0.559 | 0.504 | 0.701 | 0.541 | 0.406 | 0.414 |
| | 18 | **0.317** | **0.359** | 0.418 | 0.418 | 0.374 | 0.392 | 0.434 | 0.439 | 0.535 | 0.485 | 0.677 | 0.535 | 0.409 | 0.418 |
| ETTm1 | 6 | **0.131** | **0.218** | 0.181 | 0.250 | 0.161 | 0.235 | 0.138 | 0.225 | 0.272 | 0.328 | 0.281 | 0.309 | 0.162 | 0.234 |
| | 12 | **0.216** | **0.279** | 0.311 | 0.331 | 0.294 | 0.321 | 0.231 | 0.289 | 0.402 | 0.396 | 0.555 | 0.437 | 0.301 | 0.317 |
| | 18 | **0.311** | **0.329** | 0.409 | 0.388 | 0.389 | 0.366 | 0.323 | 0.340 | 0.511 | 0.445 | 0.628 | 0.471 | 0.410 | 0.375 |
| ETTh2 | 6 | **0.132** | **0.227** | 0.159 | 0.264 | 0.157 | 0.260 | 0.162 | 0.267 | 0.199 | 0.327 | 0.213 | 0.310 | 0.150 | 0.252 |
| | 12 | **0.144** | **0.240** | 0.175 | 0.273 | 0.170 | 0.268 | 0.169 | 0.266 | 0.219 | 0.337 | 0.205 | 0.295 | 0.174 | 0.273 |
| | 18 | **0.159** | **0.253** | 0.181 | 0.273 | 0.178 | 0.270 | 0.200 | 0.294 | 0.215 | 0.319 | 0.247 | 0.325 | 0.192 | 0.284 |
| ETTm2 | 6 | **0.060** | **0.138** | 0.066 | 0.155 | 0.065 | 0.151 | 0.062 | 0.144 | 0.083 | 0.200 | 0.083 | 0.176 | 0.065 | 0.150 |
| | 12 | **0.079** | **0.165** | 0.089 | 0.186 | 0.089 | 0.181 | 0.080 | 0.170 | 0.109 | 0.228 | 0.118 | 0.219 | 0.092 | 0.186 |
| | 18 | **0.093** | **0.182** | 0.110 | 0.211 | 0.112 | 0.208 | 0.098 | 0.191 | 0.125 | 0.242 | 0.130 | 0.232 | 0.114 | 0.210 |
| ECL | 6 | **0.199** | **0.306** | 0.661 | 0.597 | 0.333 | 0.391 | 0.205 | 0.307 | 1.021 | 0.836 | 1.386 | 0.931 | 0.651 | 0.571 |
| | 12 | **0.117** | **0.221** | 0.232 | 0.340 | 0.152 | 0.239 | 0.130 | 0.234 | 0.394 | 0.491 | 0.507 | 0.531 | 0.212 | 0.296 |
| | 18 | **0.117** | **0.219** | 0.237 | 0.348 | 0.158 | 0.243 | 0.133 | 0.240 | 0.341 | 0.440 | 0.492 | 0.523 | 0.216 | 0.300 |
| Weather | 6 | **0.058** | **0.069** | 0.065 | 0.083 | 0.063 | 0.078 | 0.070 | 0.088 | 0.081 | 0.145 | 0.109 | 0.135 | 0.065 | 0.083 |
| | 12 | **0.071** | **0.086** | 0.085 | 0.108 | 0.084 | 0.101 | 0.086 | 0.114 | 0.096 | 0.161 | 0.124 | 0.152 | 0.091 | 0.111 |
| | 18 | **0.083** | **0.105** | 0.099 | 0.126 | 0.098 | 0.119 | 0.105 | 0.141 | 0.116 | 0.188 | 0.130 | 0.100 | 0.106 | 0.130 |
| ILI | 6 | 1.419 | 0.694 | 3.402 | 1.213 | 2.793 | 1.079 | **1.101** | **0.597** | 3.407 | 1.255 | 3.378 | 1.170 | 2.485 | 1.024 |
| | 12 | **2.420** | **0.939** | 3.958 | 1.427 | 4.591 | 1.534 | 2.760 | 0.927 | 5.597 | 1.753 | 6.106 | 1.784 | 4.271 | 1.461 |
| | 18 | **2.394** | **0.983** | 4.905 | 1.594 | 3.774 | 1.402 | 3.516 | 1.014 | 4.875 | 1.666 | 5.953 | 1.815 | 4.327 | 1.497 |
| Traffic | 6 | 0.937 | **0.414** | 1.023 | 0.586 | **0.790** | 0.442 | 1.064 | 0.402 | 1.785 | 0.946 | 2.392 | 1.095 | 1.414 | 0.736 |
| | 12 | 0.461 | 0.301 | 0.618 | 0.416 | **0.453** | **0.289** | 0.566 | 0.297 | 0.964 | 0.608 | 1.475 | 0.809 | 0.662 | 0.414 |
| | 18 | **0.453** | **0.300** | 0.653 | 0.429 | 0.468 | 0.305 | 0.587 | 0.305 | 0.933 | 0.574 | 1.321 | 0.731 | 0.691 | 0.430 |
| Exchange | 6 | **0.008** | **0.052** | 0.010 | 0.068 | 0.010 | 0.065 | 0.009 | 0.063 | 0.048 | 0.175 | 0.014 | 0.078 | 0.009 | 0.061 |
| | 12 | **0.013** | **0.072** | 0.017 | 0.089 | 0.017 | 0.087 | 0.018 | 0.089 | 0.038 | 0.148 | 0.023 | 0.106 | 0.017 | 0.089 |
| | 18 | **0.019** | **0.089** | 0.023 | 0.104 | 0.023 | 0.103 | 0.025 | 0.107 | 0.048 | 0.171 | 0.032 | 0.127 | 0.024 | 0.108 |
| 1$^{st}$ Count | | **49** | | 0 | | 3 | | 2 | | 0 | | 0 | | 0 | |

## L ABLATION STUDY ABOUT RevIN ON REGRESSION TASKS

### L.1 ABLATION STUDY ABOUT RevIN ON LONG-TERM FORECASTING

We conduct ablation study on Weather, ETTh1 and ETTm1 datasets for long-term forecasting tasks.

The results in Table 22 show that:

- Although there is a slight degradation in performance without RevIN, our ModernTCN still achieves competitive performance. ModernTCN can still achieve significantly better performance than some baselines and compete favorably with PatchTST in the same case of no RevIN. The results indicate that our designs in ModernTCN also make great contribution to the performance improvement.

- We also find that RevIN doesn't provide consistent improvement under some settings (e.g., prediciton length 192 and 720 in Weather dataset), which means our ModernTCN can directly predict from the non-stationary time series to some degree. This finding further confirms that our designs in ModernTCN can bring great performance improvement on their own. And such phenomenon mainly happens in Weather dataset for its degree of stationarity is not very high according to (Liu et al., 2022c).

Table 22: Ablation Study about RevIN on long-term forecasting tasks. We report MSE and MAE under four different prediction lengths as metrics. +Rev means with RevIN. -Rev means without RevIN.

| Models | | ModernTCN (+Rev) | | ModernTCN (-Rev) | | PatchTST (+Rev) | | PatchTST (-Rev) | | FEDformer | | Autoformer | | Informer | |
|---|---|---|---|---|---|---|---|---|---|---|---|---|---|---|---|
| Metric | | MSE | MAE | MSE | MAE | MSE | MAE | MSE | MAE | MSE | MAE | MSE | MAE | MSE | MAE |
| Weather | 96 | 0.149 | 0.200 | 0.155 | 0.233 | 0.149 | 0.198 | 0.161 | 0.219 | 0.238 | 0.314 | 0.249 | 0.329 | 0.354 | 0.405 |
| | 192 | 0.196 | 0.245 | 0.196 | 0.249 | 0.194 | 0.241 | 0.201 | 0.254 | 0.275 | 0.329 | 0.325 | 0.370 | 0.419 | 0.434 |
| | 336 | 0.238 | 0.277 | 0.247 | 0.323 | 0.245 | 0.282 | 0.253 | 0.298 | 0.339 | 0.377 | 0.351 | 0.391 | 0.583 | 0.543 |
| | 720 | 0.314 | 0.334 | 0.312 | 0.366 | 0.314 | 0.334 | 0.323 | 0.357 | 0.389 | 0.409 | 0.415 | 0.426 | 0.916 | 0.705 |
| ETTh1 | 96 | 0.368 | 0.394 | 0.381 | 0.406 | 0.370 | 0.399 | 0.385 | 0.410 | 0.376 | 0.415 | 0.435 | 0.446 | 0.941 | 0.769 |
| | 192 | 0.405 | 0.413 | 0.418 | 0.426 | 0.413 | 0.421 | 0.417 | 0.432 | 0.423 | 0.446 | 0.456 | 0.457 | 1.007 | 0.786 |
| | 336 | 0.391 | 0.412 | 0.413 | 0.434 | 0.422 | 0.436 | 0.439 | 0.449 | 0.444 | 0.462 | 0.486 | 0.487 | 1.038 | 0.784 |
| | 720 | 0.450 | 0.461 | 0.487 | 0.500 | 0.447 | 0.466 | 0.478 | 0.494 | 0.469 | 0.492 | 0.515 | 0.517 | 1.144 | 0.857 |
| ETTm1 | 96 | 0.292 | 0.346 | 0.313 | 0.362 | 0.290 | 0.342 | 0.308 | 0.358 | 0.326 | 0.390 | 0.510 | 0.492 | 0.626 | 0.560 |
| | 192 | 0.332 | 0.368 | 0.336 | 0.375 | 0.332 | 0.369 | 0.356 | 0.390 | 0.365 | 0.415 | 0.514 | 0.495 | 0.725 | 0.619 |
| | 336 | 0.365 | 0.391 | 0.367 | 0.393 | 0.366 | 0.392 | 0.389 | 0.411 | 0.392 | 0.425 | 0.510 | 0.492 | 1.005 | 0.741 |
| | 720 | 0.416 | 0.417 | 0.427 | 0.426 | 0.416 | 0.420 | 0.430 | 0.439 | 0.446 | 0.458 | 0.527 | 0.493 | 1.133 | 0.845 |

### L.2 THE IMPACT OF RevIN ON DIFFERENT TASKS AND MODELS

To study the impact of RevIN on different tasks and models, we further conduct experiments on four regression tasks with 6 models: ModernTCN, PatchTST (2023), TimesNet(2023), SCINet(2022a), RMLP and RLinear(2023a). Details are as follows:

- We conduct ablation study about RevIN on long-term forecasting with more models. And the results are provided in Table 23.

- We conduct ablation study about RevIN on imputation tasks in Weather, ETTh1 and ETTm1 datasets. And we report the MSE and MAE under four different mask ratios as metrics. Results are shown in Table 24.

- We conduct ablation study about RevIN on short-term forecasting tasks in M4 datasets and roport the weighted averaged results from six M4 sub-datasets as metric. Results are shown in Table 25.

- We conduct ablation study about RevIN on anomaly detection tasks in SMD, MSL and PSM datasets and roport the F1-score as metric. Results are shown in Table 26.

**RevIN's impact on different tasks** In general, removing the RevIN will lead to performance degradation in all four tasks because it is more difficult to predict from the non-stationary time series (Liu et al., 2022c; Kim et al., 2021). Given that stationary techniques like RevIN or decomposition (e.g., MICN, DLinear, FEDformer) can transform the input into stationary series that is easier to analyze with deep learning methods, it is generally necessary to adopt stationary techniques in regression tasks.

We also find that short-term forecasting is more sensitive to RevIN. The samples in M4 datasets are collected from different sources and have quite different temporal property. Therefore, there is a greater need of RevIN to mitigate the distribution shift.

**RevIN's impact on different models** Although removing RevIN will cause performance degradation on all 6 models in our experiments, our ModernTCN is one of the less influenced models, indicating that our ModernTCN is robust to the usage of RevIN.

The extent of RevIN's influence on the model is related to the model's mechanisms. For example, an important step in TimesNet is calculate the periods of time series based on FFT. Since FFT mainly works well on stationary signals, there is a greater need for the time series to be stationary. As a result, TimesNet's performance is highly related to RevIN.

Table 23: Ablation about RevIN on Long-term forecasting for other models. We report MSE and MAE under four different prediction lengths as metrics. +Rev means with RevIN. -Rev means without RevIN.

| Models | RMLP (+Rev) | | RMLP (-Rev) | | RLinear (+Rev) | | RLinear (-Rev) | | TimesNet (+Rev) | | TimesNet (-Rev) | | SCINet (+Rev) | | SCINet (-Rev) | |
|---|---|---|---|---|---|---|---|---|---|---|---|---|---|---|---|---|
| Metric | MSE | MAE | MSE | MAE | MSE | MAE | MSE | MAE | MSE | MAE | MSE | MAE | MSE | MAE | MSE | MAE |
| ETTm1 96 | 0.298 | 0.345 | 0.310 | 0.359 | 0.301 | 0.342 | 0.334 | 0.374 | 0.338 | 0.375 | 0.622 | 0.596 | 0.325 | 0.372 | 0.399 | 0.432 |
| ETTm1 192 | 0.344 | 0.375 | 0.350 | 0.382 | 0.335 | 0.363 | 0.365 | 0.392 | 0.371 | 0.387 | 0.622 | 0.583 | 0.354 | 0.386 | 0.411 | 0.436 |
| ETTm1 336 | 0.390 | 0.410 | 0.387 | 0.405 | 0.370 | 0.383 | 0.397 | 0.410 | 0.410 | 0.411 | 0.703 | 0.644 | 0.394 | 0.415 | 0.459 | 0.465 |
| ETTm1 720 | 0.445 | 0.441 | 0.448 | 0.442 | 0.425 | 0.414 | 0.456 | 0.448 | 0.478 | 0.450 | 0.901 | 0.736 | 0.476 | 0.469 | 0.530 | 0.511 |
| ETTh1 96 | 0.390 | 0.410 | 0.471 | 0.468 | 0.366 | 0.391 | 0.532 | 0.497 | 0.384 | 0.402 | 1.261 | 0.883 | 0.375 | 0.406 | 0.525 | 0.513 |
| ETTh1 192 | 0.430 | 0.432 | 0.511 | 0.492 | 0.404 | 0.412 | 0.559 | 0.513 | 0.557 | 0.436 | 1.132 | 0.835 | 0.416 | 0.421 | 0.570 | 0.542 |
| ETTh1 336 | 0.441 | 0.441 | 0.540 | 0.515 | 0.420 | 0.423 | 0.569 | 0.526 | 0.491 | 0.469 | 1.166 | 0.841 | 0.504 | 0.495 | 0.580 | 0.548 |
| ETTh1 720 | 0.506 | 0.495 | 0.603 | 0.575 | 0.442 | 0.456 | 0.602 | 0.568 | 0.521 | 0.500 | 1.269 | 0.926 | 0.544 | 0.527 | 0.589 | 0.564 |
| Weather 96 | 0.149 | 0.202 | 0.179 | 0.239 | 0.175 | 0.225 | 0.226 | 0.283 | 0.172 | 0.220 | 0.222 | 0.316 | 0.161 | 0.226 | 0.180 | 0.260 |
| Weather 192 | 0.194 | 0.242 | 0.220 | 0.278 | 0.218 | 0.260 | 0.270 | 0.319 | 0.219 | 0.261 | 0.249 | 0.326 | 0.220 | 0.283 | 0.229 | 0.299 |
| Weather 336 | 0.243 | 0.282 | 0.264 | 0.313 | 0.265 | 0.294 | 0.332 | 0.369 | 0.280 | 0.306 | 0.360 | 0.413 | 0.275 | 0.328 | 0.280 | 0.333 |
| Weather 720 | 0.316 | 0.333 | 0.327 | 0.364 | 0.329 | 0.339 | 0.188 | 0.250 | 0.365 | 0.359 | 0.414 | 0.434 | 0.311 | 0.356 | 0.337 | 0.370 |

Table 24: Ablation about RevIN on imputation tasks. We report MSE and MAE under four different mask ratios as metrics. +Rev means with RevIN. -Rev means without RevIN.

| Models | ModernTCN (+Rev) | | ModernTCN (-Rev) | | PatchTST (+Rev) | | PatchTST (-Rev) | | RMLP (+Rev) | | RMLP (-Rev) | | RLinear (+Rev) | | RLinear (-Rev) | | TimesNet (+Rev) | | TimesNet (-Rev) | | SCINet (+Rev) | | SCINet (-Rev) | |
|---|---|---|---|---|---|---|---|---|---|---|---|---|---|---|---|---|---|---|---|---|---|---|---|---|
| Metric | MSE | MAE | MSE | MAE | MSE | MAE | MSE | MAE | MSE | MAE | MSE | MAE | MSE | MAE | MSE | MAE | MSE | MAE | MSE | MAE | MSE | MAE | MSE | MAE |
| ETTm1 0.125 | 0.015 | 0.082 | 0.017 | 0.090 | 0.041 | 0.128 | 0.042 | 0.141 | 0.049 | 0.139 | 0.056 | 0.162 | 0.047 | 0.137 | 0.056 | 0.161 | 0.019 | 0.092 | 0.022 | 0.103 | 0.031 | 0.116 | 0.290 | 0.370 |
| ETTm1 0.25 | 0.018 | 0.088 | 0.020 | 0.095 | 0.043 | 0.130 | 0.046 | 0.143 | 0.057 | 0.154 | 0.072 | 0.186 | 0.061 | 0.157 | 0.077 | 0.190 | 0.023 | 0.101 | 0.029 | 0.118 | 0.036 | 0.124 | 0.042 | 0.144 |
| ETTm1 0.375 | 0.021 | 0.095 | 0.025 | 0.107 | 0.044 | 0.133 | 0.054 | 0.155 | 0.067 | 0.168 | 0.089 | 0.206 | 0.077 | 0.175 | 0.099 | 0.217 | 0.029 | 0.111 | 0.034 | 0.129 | 0.041 | 0.134 | 0.050 | 0.158 |
| ETTm1 0.5 | 0.026 | 0.105 | 0.031 | 0.118 | 0.050 | 0.142 | 0.062 | 0.164 | 0.079 | 0.183 | 0.108 | 0.226 | 0.096 | 0.195 | 0.128 | 0.246 | 0.036 | 0.124 | 0.044 | 0.147 | 0.049 | 0.143 | 0.060 | 0.173 |
| ETTh1 0.125 | 0.035 | 0.128 | 0.035 | 0.126 | 0.094 | 0.199 | 0.115 | 0.233 | 0.096 | 0.205 | 0.120 | 0.239 | 0.098 | 0.206 | 0.115 | 0.236 | 0.057 | 0.159 | 0.089 | 0.217 | 0.089 | 0.202 | 0.123 | 0.251 |
| ETTh1 0.25 | 0.042 | 0.140 | 0.049 | 0.154 | 0.119 | 0.225 | 0.138 | 0.256 | 0.120 | 0.228 | 0.151 | 0.270 | 0.123 | 0.229 | 0.149 | 0.268 | 0.069 | 0.178 | 0.112 | 0.243 | 0.099 | 0.211 | 0.132 | 0.258 |
| ETTh1 0.375 | 0.054 | 0.157 | 0.060 | 0.168 | 0.145 | 0.248 | 0.166 | 0.281 | 0.145 | 0.250 | 0.186 | 0.299 | 0.153 | 0.253 | 0.189 | 0.300 | 0.084 | 0.196 | 0.154 | 0.288 | 0.107 | 0.218 | 0.145 | 0.272 |
| ETTh1 0.5 | 0.067 | 0.174 | 0.079 | 0.196 | 0.173 | 0.271 | 0.200 | 0.306 | 0.176 | 0.274 | 0.223 | 0.328 | 0.188 | 0.278 | 0.232 | 0.331 | 0.102 | 0.215 | 0.181 | 0.305 | 0.120 | 0.231 | 0.159 | 0.287 |
| Weather 0.125 | 0.023 | 0.038 | 0.026 | 0.059 | 0.029 | 0.049 | 0.041 | 0.106 | 0.030 | 0.051 | 0.039 | 0.086 | 0.029 | 0.048 | 0.039 | 0.091 | 0.025 | 0.045 | 0.032 | 0.078 | 0.028 | 0.047 | 0.032 | 0.074 |
| Weather 0.25 | 0.025 | 0.041 | 0.032 | 0.077 | 0.031 | 0.053 | 0.045 | 0.108 | 0.033 | 0.057 | 0.046 | 0.103 | 0.032 | 0.055 | 0.047 | 0.109 | 0.029 | 0.052 | 0.036 | 0.085 | 0.029 | 0.050 | 0.032 | 0.077 |
| Weather 0.375 | 0.027 | 0.046 | 0.030 | 0.069 | 0.034 | 0.058 | 0.042 | 0.098 | 0.036 | 0.062 | 0.052 | 0.115 | 0.036 | 0.062 | 0.057 | 0.123 | 0.031 | 0.057 | 0.041 | 0.094 | 0.031 | 0.055 | 0.036 | 0.083 |
| Weather 0.5 | 0.031 | 0.051 | 0.037 | 0.090 | 0.039 | 0.066 | 0.043 | 0.092 | 0.040 | 0.068 | 0.060 | 0.126 | 0.040 | 0.067 | 0.066 | 0.137 | 0.034 | 0.062 | 0.046 | 0.102 | 0.034 | 0.059 | 0.040 | 0.090 |

Table 25: Ablation study about RevIN on short-term forecasting. Results are weighted averaged from several datasets under different sample intervals. +Rev means with RevIN. -Rev means without RevIN. Since removing RevIn in PatchTST will make the training loss become NaN, making it hard to train on short-term forecasting, we don't report the result of PatchTST without RevIN.

| Models | | **ModernTCN** (**Ours**) | PatchTST (2023) | RMLP (2023a) | RLinear (2023a) | TimesNet (2023) | SCINet (2022a) |
|---|---|---|---|---|---|---|---|
| +Rev | SMAPE | 11.698 | 11.807 | 12.072 | 12.473 | 11.829 | 12.369 |
| | MASE | 1.556 | 1.590 | 1.624 | 1.677 | 1.585 | 1.677 |
| | OWA | 0.838 | 0.851 | 0.870 | 0.898 | 0.851 | 0.894 |
| -Rev | SMAPE | 12.241 | - | 14.984 | 12.777 | 90.132 | 12.950 |
| | MASE | 1.796 | - | 2.716 | 1.789 | 20.17 | 1.955 |
| | OWA | 0.921 | - | 1.262 | 0.939 | 8.597 | 0.989 |

Table 26: Ablation study about RevIN on anomaly detection. We report F1-score as metric. +Rev means with RevIN. -Rev means without RevIN.

| Models | | **ModernTCN** (**Ours**) | PatchTST (2023) | RMLP (2023a) | RLinear (2023a) | TimesNet (2023) | SCINet (2022a) |
|---|---|---|---|---|---|---|---|
| +Rev | SMD | 85.81 | 84.44 | 82.46 | 83.70 | 85.81 | 84.24 |
| | MSL | 84.92 | 85.14 | 74.48 | 81.15 | 85.15 | 83.38 |
| | PSM | 97.23 | 96.37 | 92.76 | 96.33 | 97.47 | 96.00 |
| -Rev | SMD | 81.42 | 71.17 | 73.63 | 81.32 | 71.20 | 83.75 |
| | MSL | 82.07 | 81.70 | 78.05 | 81.86 | 82.20 | 81.28 |
| | PSM | 96.36 | 95.93 | 93.36 | 96.00 | 90.66 | 95.93 |

# M    FULL RESULTS

Due to the space limitation of the main text, we place the full results of all experiments in the following: long-term forecasting in Table 27, short-term forecasting in Table 28, imputation in Table 29, classification in Table 30 and anomaly detection in Table 31.

And some showcases are provided in Appendix N.

## M.1    LONG-TERM

## M.2    SHORT-TERM

## M.3    IMPUTATION

## M.4    CLASSIFICATION

## M.5    ANOMALY DETECTION

Table 27: Full results for the **long-term forecasting** task. We compare extensive competitive models under four prediction lengths. The input sequence length is searched to the best for a fairer comparison. *Avg* is averaged from all four prediction lengths.

| Models | | ModernTCN (Ours) | | PatchTST (2023) | | Crossformer (2023) | | FEDformer (2022) | | MTS-mixer (2023b) | | LightTS (2022) | | DLinear (2022) | | TimesNet (2023) | | MICN (2023) | | SCINet (2022a) | | RLinear (2023a) | | RMLP (2023a) | |
|---|---|---|---|---|---|---|---|---|---|---|---|---|---|---|---|---|---|---|---|---|---|---|---|---|---|
| Metric | | MSE | MAE | MSE | MAE | MSE | MAE | MSE | MAE | MSE | MAE | MSE | MAE | MSE | MAE | MSE | MAE | MSE | MAE | MSE | MAE | MSE | MAE | MSE | MAE |
| ETTm1 | 96 | 0.292 | 0.346 | 0.290 | 0.342 | 0.316 | 0.373 | 0.326 | 0.390 | 0.314 | 0.358 | 0.374 | 0.400 | 0.299 | 0.343 | 0.338 | 0.375 | 0.314 | 0.360 | 0.325 | 0.372 | 0.301 | 0.342 | 0.298 | 0.345 |
| | 192 | 0.332 | 0.368 | 0.332 | 0.369 | 0.377 | 0.411 | 0.365 | 0.415 | 0.354 | 0.386 | 0.400 | 0.407 | 0.335 | 0.365 | 0.371 | 0.387 | 0.359 | 0.387 | 0.354 | 0.386 | 0.335 | 0.363 | 0.344 | 0.375 |
| | 336 | 0.365 | 0.391 | 0.366 | 0.392 | 0.431 | 0.442 | 0.392 | 0.425 | 0.384 | 0.405 | 0.438 | 0.438 | 0.369 | 0.386 | 0.410 | 0.411 | 0.398 | 0.413 | 0.394 | 0.415 | 0.370 | 0.383 | 0.390 | 0.410 |
| | 720 | 0.416 | 0.417 | 0.416 | 0.420 | 0.600 | 0.547 | 0.446 | 0.458 | 0.427 | 0.432 | 0.527 | 0.502 | 0.425 | 0.421 | 0.478 | 0.450 | 0.459 | 0.464 | 0.476 | 0.469 | 0.425 | 0.414 | 0.445 | 0.441 |
| | Avg | 0.351 | 0.381 | 0.351 | 0.381 | 0.431 | 0.443 | 0.382 | 0.422 | 0.370 | 0.395 | 0.435 | 0.437 | 0.357 | 0.379 | 0.400 | 0.406 | 0.383 | 0.406 | 0.387 | 0.411 | 0.358 | 0.376 | 0.369 | 0.393 |
| ETTm2 | 96 | 0.166 | 0.256 | 0.165 | 0.255 | 0.421 | 0.461 | 0.180 | 0.271 | 0.177 | 0.259 | 0.209 | 0.308 | 0.167 | 0.260 | 0.187 | 0.267 | 0.178 | 0.273 | 0.186 | 0.281 | 0.164 | 0.253 | 0.174 | 0.259 |
| | 192 | 0.222 | 0.293 | 0.220 | 0.292 | 0.503 | 0.519 | 0.252 | 0.318 | 0.241 | 0.303 | 0.311 | 0.382 | 0.224 | 0.303 | 0.249 | 0.309 | 0.245 | 0.316 | 0.277 | 0.356 | 0.219 | 0.290 | 0.236 | 0.303 |
| | 336 | 0.272 | 0.324 | 0.274 | 0.329 | 0.611 | 0.580 | 0.324 | 0.364 | 0.297 | 0.338 | 0.442 | 0.466 | 0.281 | 0.342 | 0.321 | 0.351 | 0.295 | 0.350 | 0.311 | 0.369 | 0.273 | 0.326 | 0.291 | 0.338 |
| | 720 | 0.351 | 0.381 | 0.362 | 0.385 | 0.996 | 0.750 | 0.410 | 0.420 | 0.396 | 0.398 | 0.675 | 0.587 | 0.397 | 0.421 | 0.497 | 0.403 | 0.389 | 0.406 | 0.403 | 0.412 | 0.366 | 0.385 | 0.371 | 0.391 |
| | Avg | 0.253 | 0.314 | 0.255 | 0.315 | 0.632 | 0.578 | 0.292 | 0.343 | 0.277 | 0.325 | 0.409 | 0.436 | 0.267 | 0.332 | 0.291 | 0.333 | 0.277 | 0.336 | 0.294 | 0.355 | 0.256 | 0.314 | 0.268 | 0.322 |
| ETTh1 | 96 | 0.368 | 0.394 | 0.370 | 0.399 | 0.386 | 0.429 | 0.376 | 0.415 | 0.372 | 0.395 | 0.424 | 0.432 | 0.375 | 0.399 | 0.384 | 0.402 | 0.396 | 0.427 | 0.375 | 0.406 | 0.366 | 0.391 | 0.390 | 0.410 |
| | 192 | 0.405 | 0.413 | 0.413 | 0.421 | 0.419 | 0.444 | 0.423 | 0.446 | 0.416 | 0.426 | 0.475 | 0.462 | 0.405 | 0.416 | 0.557 | 0.436 | 0.430 | 0.453 | 0.416 | 0.421 | 0.404 | 0.412 | 0.430 | 0.432 |
| | 336 | 0.391 | 0.412 | 0.422 | 0.436 | 0.440 | 0.461 | 0.444 | 0.462 | 0.455 | 0.449 | 0.518 | 0.521 | 0.439 | 0.443 | 0.491 | 0.469 | 0.433 | 0.458 | 0.504 | 0.495 | 0.420 | 0.423 | 0.441 | 0.441 |
| | 720 | 0.450 | 0.461 | 0.447 | 0.466 | 0.519 | 0.524 | 0.469 | 0.492 | 0.475 | 0.472 | 0.547 | 0.533 | 0.472 | 0.490 | 0.521 | 0.500 | 0.474 | 0.508 | 0.544 | 0.527 | 0.442 | 0.456 | 0.506 | 0.495 |
| | Avg | 0.404 | 0.420 | 0.413 | 0.431 | 0.441 | 0.465 | 0.428 | 0.454 | 0.430 | 0.436 | 0.491 | 0.479 | 0.423 | 0.437 | 0.458 | 0.450 | 0.433 | 0.462 | 0.460 | 0.462 | 0.408 | 0.421 | 0.442 | 0.445 |
| ETTh2 | 96 | 0.263 | 0.332 | 0.274 | 0.336 | 0.628 | 0.563 | 0.332 | 0.374 | 0.307 | 0.354 | 0.397 | 0.437 | 0.289 | 0.353 | 0.340 | 0.374 | 0.289 | 0.357 | 0.295 | 0.361 | 0.262 | 0.331 | 0.288 | 0.352 |
| | 192 | 0.320 | 0.374 | 0.339 | 0.379 | 0.703 | 0.624 | 0.407 | 0.446 | 0.374 | 0.399 | 0.520 | 0.504 | 0.383 | 0.418 | 0.402 | 0.414 | 0.409 | 0.438 | 0.349 | 0.383 | 0.320 | 0.374 | 0.343 | 0.387 |
| | 336 | 0.313 | 0.376 | 0.329 | 0.380 | 0.827 | 0.675 | 0.400 | 0.447 | 0.398 | 0.432 | 0.626 | 0.559 | 0.448 | 0.465 | 0.452 | 0.452 | 0.417 | 0.452 | 0.365 | 0.409 | 0.325 | 0.386 | 0.353 | 0.402 |
| | 720 | 0.392 | 0.433 | 0.379 | 0.422 | 1.181 | 0.840 | 0.412 | 0.469 | 0.463 | 0.465 | 0.863 | 0.672 | 0.605 | 0.551 | 0.462 | 0.468 | 0.426 | 0.473 | 0.475 | 0.488 | 0.372 | 0.421 | 0.410 | 0.440 |
| | Avg | 0.322 | 0.379 | 0.330 | 0.379 | 0.835 | 0.676 | 0.388 | 0.434 | 0.386 | 0.413 | 0.602 | 0.543 | 0.431 | 0.447 | 0.414 | 0.427 | 0.385 | 0.430 | 0.371 | 0.410 | 0.320 | 0.378 | 0.349 | 0.395 |
| Electricity | 96 | 0.129 | 0.226 | 0.129 | 0.222 | 0.187 | 0.283 | 0.186 | 0.302 | 0.141 | 0.243 | 0.207 | 0.307 | 0.153 | 0.237 | 0.168 | 0.272 | 0.159 | 0.267 | 0.171 | 0.256 | 0.140 | 0.235 | 0.129 | 0.224 |
| | 192 | 0.143 | 0.239 | 0.147 | 0.240 | 0.258 | 0.330 | 0.197 | 0.311 | 0.163 | 0.261 | 0.213 | 0.316 | 0.152 | 0.249 | 0.184 | 0.289 | 0.168 | 0.279 | 0.177 | 0.265 | 0.154 | 0.248 | 0.147 | 0.240 |
| | 336 | 0.161 | 0.259 | 0.163 | 0.259 | 0.323 | 0.369 | 0.213 | 0.328 | 0.176 | 0.277 | 0.230 | 0.333 | 0.169 | 0.267 | 0.198 | 0.300 | 0.196 | 0.308 | 0.197 | 0.285 | 0.171 | 0.264 | 0.164 | 0.257 |
| | 720 | 0.191 | 0.286 | 0.197 | 0.290 | 0.404 | 0.423 | 0.233 | 0.344 | 0.212 | 0.308 | 0.265 | 0.360 | 0.233 | 0.344 | 0.220 | 0.320 | 0.203 | 0.312 | 0.234 | 0.318 | 0.209 | 0.297 | 0.203 | 0.291 |
| | Avg | 0.156 | 0.253 | 0.159 | 0.253 | 0.293 | 0.351 | 0.207 | 0.321 | 0.173 | 0.272 | 0.229 | 0.329 | 0.177 | 0.274 | 0.192 | 0.295 | 0.182 | 0.292 | 0.195 | 0.281 | 0.169 | 0.261 | 0.161 | 0.253 |
| Weather | 96 | 0.149 | 0.200 | 0.149 | 0.198 | 0.153 | 0.217 | 0.238 | 0.314 | 0.156 | 0.206 | 0.182 | 0.242 | 0.152 | 0.237 | 0.172 | 0.220 | 0.161 | 0.226 | 0.178 | 0.233 | 0.175 | 0.225 | 0.149 | 0.202 |
| | 192 | 0.196 | 0.245 | 0.194 | 0.241 | 0.197 | 0.269 | 0.275 | 0.329 | 0.199 | 0.248 | 0.227 | 0.287 | 0.220 | 0.282 | 0.219 | 0.261 | 0.220 | 0.283 | 0.235 | 0.277 | 0.218 | 0.260 | 0.194 | 0.242 |
| | 336 | 0.238 | 0.277 | 0.245 | 0.282 | 0.252 | 0.311 | 0.339 | 0.377 | 0.249 | 0.291 | 0.282 | 0.334 | 0.265 | 0.319 | 0.280 | 0.306 | 0.275 | 0.328 | 0.337 | 0.345 | 0.265 | 0.294 | 0.243 | 0.282 |
| | 720 | 0.314 | 0.334 | 0.314 | 0.334 | 0.318 | 0.363 | 0.389 | 0.409 | 0.336 | 0.343 | 0.352 | 0.386 | 0.323 | 0.362 | 0.365 | 0.359 | 0.311 | 0.356 | 0.396 | 0.413 | 0.329 | 0.339 | 0.316 | 0.333 |
| | Avg | 0.224 | 0.264 | 0.226 | 0.264 | 0.230 | 0.290 | 0.310 | 0.357 | 0.235 | 0.272 | 0.261 | 0.312 | 0.240 | 0.300 | 0.259 | 0.287 | 0.242 | 0.298 | 0.287 | 0.317 | 0.247 | 0.279 | 0.225 | 0.265 |
| Traffic | 96 | 0.368 | 0.253 | 0.360 | 0.249 | 0.512 | 0.290 | 0.576 | 0.359 | 0.462 | 0.332 | 0.615 | 0.391 | 0.410 | 0.282 | 0.593 | 0.321 | 0.508 | 0.301 | 0.613 | 0.395 | 0.496 | 0.375 | 0.430 | 0.327 |
| | 192 | 0.379 | 0.261 | 0.379 | 0.256 | 0.523 | 0.297 | 0.610 | 0.380 | 0.488 | 0.354 | 0.601 | 0.382 | 0.423 | 0.287 | 0.617 | 0.336 | 0.536 | 0.315 | 0.559 | 0.363 | 0.503 | 0.377 | 0.451 | 0.340 |
| | 336 | 0.397 | 0.270 | 0.392 | 0.264 | 0.530 | 0.300 | 0.608 | 0.375 | 0.498 | 0.360 | 0.613 | 0.386 | 0.436 | 0.296 | 0.629 | 0.336 | 0.525 | 0.310 | 0.555 | 0.358 | 0.517 | 0.382 | 0.470 | 0.351 |
| | 720 | 0.440 | 0.296 | 0.432 | 0.286 | 0.573 | 0.313 | 0.621 | 0.375 | 0.529 | 0.370 | 0.658 | 0.407 | 0.466 | 0.315 | 0.640 | 0.350 | 0.571 | 0.323 | 0.620 | 0.394 | 0.555 | 0.398 | 0.513 | 0.372 |
| | Avg | 0.396 | 0.270 | 0.391 | 0.264 | 0.535 | 0.300 | 0.604 | 0.372 | 0.494 | 0.354 | 0.622 | 0.392 | 0.434 | 0.295 | 0.620 | 0.336 | 0.535 | 0.312 | 0.587 | 0.378 | 0.518 | 0.383 | 0.466 | 0.348 |
| Exchange | 96 | 0.080 | 0.196 | 0.093 | 0.214 | 0.186 | 0.346 | 0.139 | 0.276 | 0.083 | 0.201 | 0.116 | 0.262 | 0.081 | 0.203 | 0.107 | 0.234 | 0.102 | 0.235 | 0.116 | 0.254 | 0.083 | 0.201 | 0.083 | 0.201 |
| | 192 | 0.166 | 0.288 | 0.192 | 0.312 | 0.467 | 0.522 | 0.256 | 0.369 | 0.174 | 0.296 | 0.215 | 0.359 | 0.157 | 0.293 | 0.226 | 0.344 | 0.172 | 0.316 | 0.218 | 0.345 | 0.170 | 0.293 | 0.170 | 0.292 |
| | 336 | 0.307 | 0.398 | 0.350 | 0.432 | 0.783 | 0.721 | 0.426 | 0.464 | 0.336 | 0.417 | 0.377 | 0.466 | 0.305 | 0.414 | 0.367 | 0.448 | 0.272 | 0.407 | 0.294 | 0.413 | 0.309 | 0.401 | 0.309 | 0.401 |
| | 720 | 0.656 | 0.582 | 0.911 | 0.716 | 1.367 | 0.943 | 1.090 | 0.800 | 0.900 | 0.715 | 0.831 | 0.699 | 0.643 | 0.601 | 0.964 | 0.746 | 0.714 | 0.658 | 1.110 | 0.767 | 0.817 | 0.680 | 0.816 | 0.680 |
| | Avg | 0.302 | 0.366 | 0.387 | 0.419 | 0.701 | 0.633 | 0.478 | 0.478 | 0.373 | 0.407 | 0.385 | 0.447 | 0.297 | 0.378 | 0.416 | 0.443 | 0.315 | 0.404 | 0.435 | 0.445 | 0.345 | 0.394 | 0.345 | 0.394 |
| ILI | 24 | 1.347 | 0.717 | 1.319 | 0.754 | 3.040 | 1.186 | 2.624 | 1.095 | 1.472 | 0.798 | 8.313 | 2.144 | 2.215 | 1.081 | 2.317 | 0.934 | 2.684 | 1.112 | 2.150 | 1.005 | 4.337 | 1.507 | 4.445 | 1.536 |
| | 36 | 1.250 | 0.778 | 1.430 | 0.834 | 3.356 | 1.230 | 2.516 | 1.021 | 1.435 | 0.745 | 6.631 | 1.902 | 1.963 | 0.963 | 1.972 | 0.920 | 2.507 | 1.013 | 2.103 | 0.983 | 4.205 | 1.481 | 4.409 | 1.519 |
| | 48 | 1.388 | 0.781 | 1.553 | 0.815 | 3.441 | 1.223 | 2.505 | 1.041 | 1.474 | 0.822 | 7.299 | 1.982 | 2.130 | 1.024 | 2.238 | 0.940 | 2.423 | 1.012 | 2.432 | 1.061 | 4.257 | 1.484 | 4.388 | 1.507 |
| | 60 | 1.774 | 0.868 | 1.470 | 0.788 | 3.608 | 1.302 | 2.742 | 1.122 | 1.839 | 0.912 | 7.283 | 1.985 | 2.368 | 1.096 | 2.027 | 0.928 | 2.653 | 1.085 | 2.325 | 1.035 | 4.278 | 1.487 | 4.306 | 1.502 |
| | Avg | 1.440 | 0.786 | 1.443 | 0.798 | 3.361 | 1.235 | 2.597 | 1.070 | 1.555 | 0.819 | 7.382 | 2.003 | 2.169 | 1.041 | 2.139 | 0.931 | 2.567 | 1.055 | 2.252 | 1.021 | 4.269 | 1.490 | 4.387 | 1.516 |
| 1st Count | | 43 | | 23 | | 0 | | 0 | | 0 | | 0 | | 2 | | 0 | | 2 | | 0 | | 10 | | 1 | |

Table 28: Full results for the **short-term forecasting** task in the M4 dataset. ∗. in the Transformers indicates the name of ∗former.

| Models | | **ModernTCN (Ours)** | CARD (2023) | PatchTST (2023) | Crossformer (2023) | FEDformer (2022) | RMLP (2023a) | RLinear (2023a) | MTS-mixer (2023b) | LightTS (2022) | DLinear (2022) | TimesNet (2023) | MICN (2023) | SCINet (2022a) | N-HiTS (2023) | N-BEATS (2019) |
|---|---|---|---|---|---|---|---|---|---|---|---|---|---|---|---|---|
| Yearly | SMAPE | **13.226** | 13.302 | 13.258 | 13.392 | 13.728 | 13.418 | 13.994 | 13.548 | 14.247 | 16.965 | 13.387 | 14.935 | 13.717 | 13.418 | 13.436 |
| | MASE | **2.957** | 3.016 | 2.985 | 3.001 | 3.048 | 3.006 | 3.015 | 3.091 | 3.109 | 4.283 | 2.996 | 3.523 | 3.076 | 3.045 | 3.043 |
| | OWA | **0.777** | 0.786 | 0.781 | 0.787 | 0.803 | 0.789 | 0.807 | 0.803 | 0.827 | 1.058 | 0.786 | 0.900 | 0.807 | 0.793 | 0.794 |
| Quarterly | SMAPE | **9.971** | 10.031 | 10.179 | 16.317 | 10.792 | 10.382 | 10.702 | 10.128 | 11.364 | 12.145 | 10.100 | 11.452 | 10.845 | 10.202 | 10.124 |
| | MASE | **1.167** | 1.176 | 0.803 | 2.197 | 1.283 | 1.234 | 1.299 | 1.196 | 1.328 | 1.520 | 1.182 | 1.389 | 1.295 | 1.194 | 1.169 |
| | OWA | **0.878** | 0.884 | 0.803 | 1.542 | 0.958 | 0.921 | 0.959 | 0.896 | 1.000 | 1.106 | 0.890 | 1.026 | 0.965 | 0.899 | 0.886 |
| Monthly | SMAPE | **12.556** | 12.670 | 12.641 | 12.924 | 14.260 | 12.998 | 13.363 | 12.717 | 14.014 | 13.514 | 12.670 | 13.773 | 13.208 | 12.791 | 12.667 |
| | MASE | **0.917** | 0.933 | 0.930 | 0.966 | 1.102 | 0.976 | 1.014 | 0.931 | 1.053 | 1.037 | 0.933 | 1.076 | 0.999 | 0.969 | 0.937 |
| | OWA | **0.866** | 0.878 | 0.876 | 0.902 | 1.012 | 0.909 | 0.940 | 0.879 | 0.981 | 0.956 | 0.878 | 0.983 | 0.928 | 0.899 | 0.880 |
| Others | SMAPE | **4.715** | 5.330 | 4.946 | 5.493 | 4.954 | 5.098 | 5.437 | 4.817 | 15.880 | 6.709 | 4.891 | 6.716 | 5.423 | 5.061 | 4.925 |
| | MASE | 3.107 | 3.261 | **2.985** | 3.690 | 3.264 | 3.364 | 3.706 | 3.255 | 11.434 | 4.953 | 3.302 | 4.717 | 3.583 | 3.216 | 3.391 |
| | OWA | **0.986** | 1.075 | 1.044 | 1.160 | 1.036 | 1.067 | 1.157 | 1.02 | 3.474 | 1.487 | 1.035 | 1.451 | 1.136 | 1.040 | 1.053 |
| Weighted Average | SMAPE | **11.698** | 11.815 | 11.807 | 13.474 | 12.840 | 12.072 | 12.473 | 11.892 | 13.252 | 13.639 | 11.829 | 13.130 | 12.369 | 11.927 | 11.851 |
| | MASE | **1.556** | 1.587 | 1.590 | 1.866 | 1.701 | 1.624 | 1.677 | 1.608 | 2.111 | 2.095 | 1.585 | 1.896 | 1.677 | 1.613 | 1.599 |
| | OWA | **0.838** | 0.850 | 0.851 | 0.985 | 0.918 | 0.870 | 0.898 | 0.859 | 1.051 | 1.051 | 0.851 | 0.980 | 0.894 | 0.861 | 0.855 |

∗ The original paper of N-BEATS (Oreshkin et al., 2019) adopts a special ensemble method to promote the performance. For fair comparisons, we remove the ensemble and only compare the pure forecasting models.

∗ CARD is re-implemented by us based on the pseudo-code in the original paper (Xue et al., 2023). In original paper, CARD is trained with cosine learning rate decay and linear warm-up. For fair comparisons, we remove this additional training scheme.

Table 29: Full results for the **imputation** task. We randomly mask 12.5%, 25%, 37.5% and 50% time points to compare the model performance under different missing degrees.

| Models | ModernTCN (Ours) | | PatchTST (2023) | | Crossformer (2023) | | FEDformer (2022) | | MTS-mixer (2023b) | | LightTS (2022) | | DLinear (2022) | | TimesNet (2023) | | MICN (2023) | | SCINet (2022a) | | RLinear (2023a) | | RMLP (2023a) | |
|---|---|---|---|---|---|---|---|---|---|---|---|---|---|---|---|---|---|---|---|---|---|---|---|---|
| Mask Ratio | MSE | MAE | MSE | MAE | MSE | MAE | MSE | MAE | MSE | MAE | MSE | MAE | MSE | MAE | MSE | MAE | MSE | MAE | MSE | MAE | MSE | MAE | MSE | MAE |
| **ETTm1** 12.5% | 0.015 | 0.082 | 0.041 | 0.128 | 0.037 | 0.137 | 0.035 | 0.135 | 0.043 | 0.134 | 0.075 | 0.180 | 0.058 | 0.162 | 0.019 | 0.092 | 0.039 | 0.137 | 0.031 | 0.116 | 0.047 | 0.137 | 0.049 | 0.139 |
| 25% | 0.018 | 0.088 | 0.043 | 0.130 | 0.038 | 0.141 | 0.052 | 0.166 | 0.051 | 0.147 | 0.093 | 0.206 | 0.080 | 0.193 | 0.023 | 0.101 | 0.059 | 0.170 | 0.036 | 0.124 | 0.061 | 0.157 | 0.057 | 0.154 |
| 37.5% | 0.021 | 0.095 | 0.044 | 0.133 | 0.041 | 0.142 | 0.069 | 0.191 | 0.060 | 0.160 | 0.113 | 0.231 | 0.103 | 0.219 | 0.029 | 0.111 | 0.080 | 0.199 | 0.041 | 0.134 | 0.077 | 0.175 | 0.067 | 0.168 |
| 50% | 0.026 | 0.105 | 0.050 | 0.142 | 0.047 | 0.152 | 0.089 | 0.218 | 0.070 | 0.174 | 0.134 | 0.255 | 0.132 | 0.248 | 0.036 | 0.124 | 0.103 | 0.221 | 0.049 | 0.143 | 0.096 | 0.195 | 0.079 | 0.183 |
| Avg | 0.020 | 0.093 | 0.045 | 0.133 | 0.041 | 0.143 | 0.062 | 0.177 | 0.056 | 0.154 | 0.104 | 0.218 | 0.093 | 0.206 | 0.027 | 0.107 | 0.070 | 0.182 | 0.039 | 0.129 | 0.070 | 0.166 | 0.063 | 0.161 |
| **ETTm2** 12.5% | 0.017 | 0.076 | 0.025 | 0.092 | 0.044 | 0.148 | 0.056 | 0.159 | 0.026 | 0.096 | 0.034 | 0.127 | 0.062 | 0.166 | 0.018 | 0.080 | 0.060 | 0.165 | 0.023 | 0.093 | 0.026 | 0.093 | 0.026 | 0.096 |
| 25% | 0.018 | 0.080 | 0.027 | 0.095 | 0.047 | 0.151 | 0.080 | 0.195 | 0.030 | 0.103 | 0.042 | 0.143 | 0.085 | 0.196 | 0.020 | 0.085 | 0.100 | 0.216 | 0.026 | 0.100 | 0.030 | 0.103 | 0.030 | 0.106 |
| 37.5% | 0.020 | 0.084 | 0.029 | 0.099 | 0.044 | 0.145 | 0.110 | 0.231 | 0.033 | 0.110 | 0.051 | 0.159 | 0.106 | 0.222 | 0.023 | 0.091 | 0.163 | 0.273 | 0.028 | 0.105 | 0.034 | 0.113 | 0.034 | 0.113 |
| 50% | 0.022 | 0.090 | 0.032 | 0.106 | 0.047 | 0.150 | 0.156 | 0.276 | 0.037 | 0.118 | 0.059 | 0.174 | 0.131 | 0.247 | 0.026 | 0.098 | 0.254 | 0.342 | 0.031 | 0.111 | 0.039 | 0.123 | 0.039 | 0.121 |
| Avg | 0.019 | 0.082 | 0.028 | 0.098 | 0.046 | 0.149 | 0.101 | 0.215 | 0.032 | 0.107 | 0.046 | 0.151 | 0.096 | 0.208 | 0.022 | 0.088 | 0.144 | 0.249 | 0.027 | 0.102 | 0.032 | 0.108 | 0.032 | 0.109 |
| **ETTh1** 12.5% | 0.035 | 0.128 | 0.094 | 0.199 | 0.099 | 0.218 | 0.070 | 0.190 | 0.097 | 0.209 | 0.240 | 0.345 | 0.151 | 0.267 | 0.057 | 0.159 | 0.072 | 0.192 | 0.089 | 0.202 | 0.098 | 0.206 | 0.096 | 0.205 |
| 25% | 0.042 | 0.140 | 0.119 | 0.225 | 0.125 | 0.243 | 0.106 | 0.236 | 0.115 | 0.226 | 0.265 | 0.364 | 0.180 | 0.292 | 0.069 | 0.178 | 0.105 | 0.232 | 0.099 | 0.211 | 0.123 | 0.229 | 0.120 | 0.228 |
| 37.5% | 0.054 | 0.157 | 0.145 | 0.248 | 0.146 | 0.263 | 0.124 | 0.258 | 0.135 | 0.244 | 0.296 | 0.382 | 0.215 | 0.318 | 0.084 | 0.196 | 0.139 | 0.267 | 0.107 | 0.218 | 0.153 | 0.253 | 0.145 | 0.250 |
| 50% | 0.067 | 0.174 | 0.173 | 0.271 | 0.158 | 0.281 | 0.165 | 0.299 | 0.160 | 0.263 | 0.334 | 0.404 | 0.257 | 0.347 | 0.102 | 0.215 | 0.185 | 0.310 | 0.120 | 0.231 | 0.188 | 0.278 | 0.176 | 0.274 |
| Avg | 0.050 | 0.150 | 0.133 | 0.236 | 0.132 | 0.251 | 0.117 | 0.246 | 0.127 | 0.236 | 0.284 | 0.373 | 0.201 | 0.306 | 0.078 | 0.187 | 0.125 | 0.250 | 0.104 | 0.216 | 0.141 | 0.242 | 0.134 | 0.239 |
| **ETTh2** 12.5% | 0.037 | 0.121 | 0.057 | 0.150 | 0.103 | 0.220 | 0.095 | 0.212 | 0.061 | 0.157 | 0.101 | 0.231 | 0.100 | 0.216 | 0.040 | 0.130 | 0.106 | 0.223 | 0.061 | 0.161 | 0.057 | 0.152 | 0.058 | 0.153 |
| 25% | 0.040 | 0.127 | 0.062 | 0.158 | 0.110 | 0.229 | 0.137 | 0.258 | 0.065 | 0.163 | 0.115 | 0.246 | 0.127 | 0.247 | 0.046 | 0.141 | 0.151 | 0.271 | 0.062 | 0.162 | 0.062 | 0.160 | 0.064 | 0.161 |
| 37.5% | 0.043 | 0.134 | 0.068 | 0.168 | 0.129 | 0.246 | 0.187 | 0.304 | 0.070 | 0.171 | 0.126 | 0.257 | 0.158 | 0.276 | 0.052 | 0.151 | 0.229 | 0.332 | 0.065 | 0.166 | 0.068 | 0.168 | 0.070 | 0.171 |
| 50% | 0.048 | 0.143 | 0.076 | 0.179 | 0.148 | 0.265 | 0.232 | 0.341 | 0.078 | 0.181 | 0.136 | 0.268 | 0.183 | 0.299 | 0.060 | 0.162 | 0.334 | 0.403 | 0.069 | 0.172 | 0.076 | 0.179 | 0.078 | 0.181 |
| Avg | 0.042 | 0.131 | 0.066 | 0.164 | 0.122 | 0.240 | 0.163 | 0.279 | 0.069 | 0.168 | 0.119 | 0.250 | 0.142 | 0.259 | 0.049 | 0.146 | 0.205 | 0.307 | 0.064 | 0.165 | 0.066 | 0.165 | 0.068 | 0.166 |
| **Electricity** 12.5% | 0.059 | 0.171 | 0.073 | 0.188 | 0.068 | 0.181 | 0.107 | 0.237 | 0.069 | 0.182 | 0.102 | 0.229 | 0.092 | 0.214 | 0.085 | 0.202 | 0.090 | 0.216 | 0.073 | 0.185 | 0.079 | 0.199 | 0.073 | 0.188 |
| 25% | 0.071 | 0.188 | 0.082 | 0.200 | 0.079 | 0.198 | 0.120 | 0.251 | 0.083 | 0.202 | 0.121 | 0.252 | 0.118 | 0.247 | 0.089 | 0.206 | 0.108 | 0.236 | 0.081 | 0.198 | 0.105 | 0.233 | 0.090 | 0.211 |
| 37.5% | 0.077 | 0.190 | 0.097 | 0.217 | 0.087 | 0.203 | 0.136 | 0.266 | 0.097 | 0.218 | 0.141 | 0.273 | 0.144 | 0.276 | 0.094 | 0.213 | 0.128 | 0.257 | 0.090 | 0.207 | 0.131 | 0.262 | 0.107 | 0.231 |
| 50% | 0.085 | 0.200 | 0.110 | 0.232 | 0.096 | 0.212 | 0.158 | 0.284 | 0.108 | 0.231 | 0.160 | 0.293 | 0.175 | 0.305 | 0.100 | 0.221 | 0.151 | 0.278 | 0.099 | 0.214 | 0.160 | 0.291 | 0.125 | 0.252 |
| Avg | 0.073 | 0.187 | 0.091 | 0.209 | 0.083 | 0.199 | 0.130 | 0.259 | 0.089 | 0.208 | 0.131 | 0.262 | 0.132 | 0.260 | 0.092 | 0.210 | 0.119 | 0.247 | 0.086 | 0.201 | 0.119 | 0.246 | 0.099 | 0.221 |
| **Weather** 12.5% | 0.023 | 0.038 | 0.029 | 0.049 | 0.036 | 0.092 | 0.041 | 0.107 | 0.033 | 0.052 | 0.047 | 0.101 | 0.039 | 0.084 | 0.025 | 0.045 | 0.036 | 0.088 | 0.028 | 0.047 | 0.029 | 0.048 | 0.030 | 0.051 |
| 25% | 0.025 | 0.041 | 0.031 | 0.053 | 0.035 | 0.088 | 0.064 | 0.163 | 0.034 | 0.056 | 0.052 | 0.111 | 0.048 | 0.103 | 0.029 | 0.052 | 0.047 | 0.115 | 0.029 | 0.050 | 0.032 | 0.055 | 0.033 | 0.057 |
| 37.5% | 0.027 | 0.046 | 0.034 | 0.058 | 0.035 | 0.088 | 0.107 | 0.229 | 0.037 | 0.060 | 0.058 | 0.121 | 0.057 | 0.117 | 0.031 | 0.057 | 0.062 | 0.141 | 0.031 | 0.055 | 0.036 | 0.062 | 0.036 | 0.062 |
| 50% | 0.031 | 0.051 | 0.039 | 0.066 | 0.038 | 0.092 | 0.183 | 0.312 | 0.041 | 0.066 | 0.065 | 0.133 | 0.066 | 0.134 | 0.034 | 0.062 | 0.080 | 0.168 | 0.034 | 0.059 | 0.040 | 0.067 | 0.040 | 0.068 |
| Avg | 0.027 | 0.044 | 0.033 | 0.057 | 0.036 | 0.090 | 0.099 | 0.203 | 0.036 | 0.058 | 0.055 | 0.117 | 0.052 | 0.110 | 0.030 | 0.054 | 0.056 | 0.128 | 0.031 | 0.053 | 0.034 | 0.058 | 0.035 | 0.060 |
| 1st Count | 48 | | 0 | | 0 | | 0 | | 0 | | 0 | | 0 | | 0 | | 0 | | 0 | | 0 | | 0 | | 0 | |

Table 30: Full results for the **classification** task. ∗. in the Transformer-based models indicates the name of ∗former. We report the classification accuracy (%) as the result. The standard deviation is within 0.1%.

| Datasets / Models | RNN-based | | Convolution-based | | | | MLP-based | | | | | Transformer-based | | | | ModernTCN |
|---|---|---|---|---|---|---|---|---|---|---|---|---|---|---|---|---|
| | LSTNet (2018b) | LSSL (2022) | Rocket (2020) | SCINet (2022a) | MICN (2023) | TimesNet (2023) | DLinear (2022) | LightTS (2022) | MTS-Mixer (2023b) | RLinear (2023a) | RMLP (2023a) | FED. (2022) | Flow. (2022) | Cross. (2023) | PatchTST (2023) | (Ours) |
| EthanolConcentration | 39.9 | 31.1 | 45.2 | 34.4 | 35.3 | 35.7 | 36.2 | 29.7 | 33.8 | 28.9 | 31.3 | 31.2 | 33.8 | 38.0 | 32.8 | 36.3 |
| FaceDetection | 65.7 | 66.7 | 64.7 | 68.9 | 65.2 | 68.6 | 68.0 | 67.5 | 70.2 | 65.6 | 67.3 | 66.0 | 67.6 | 68.7 | 68.3 | 70.8 |
| Handwriting | 25.8 | 24.6 | 58.8 | 23.6 | 25.5 | 32.1 | 27.0 | 26.1 | 26.0 | 28.1 | 30.0 | 28.0 | 33.8 | 28.8 | 29.6 | 30.6 |
| Heartbeat | 77.1 | 72.7 | 75.6 | 77.5 | 74.7 | 78.0 | 75.1 | 75.1 | 77.1 | 72.6 | 72.7 | 73.7 | 77.6 | 77.6 | 74.9 | 77.2 |
| JapaneseVowels | 98.1 | 98.4 | 96.2 | 96.0 | 94.6 | 98.4 | 96.2 | 96.2 | 94.3 | 95.9 | 95.9 | 98.4 | 98.9 | 99.1 | 97.5 | 98.8 |
| PEMS-SF | 86.7 | 86.1 | 75.1 | 83.8 | 85.5 | 89.6 | 75.1 | 88.4 | 80.9 | 82.7 | 83.9 | 80.9 | 86.0 | 85.9 | 89.3 | 89.1 |
| SelfRegulationSCP1 | 84.0 | 90.8 | 90.8 | 92.5 | 86.0 | 91.8 | 87.3 | 89.8 | 91.7 | 91.1 | 92.1 | 88.7 | 92.5 | 92.1 | 90.7 | 93.4 |
| SelfRegulationSCP2 | 52.8 | 52.2 | 53.3 | 57.2 | 53.6 | 57.2 | 50.5 | 51.1 | 55.0 | 56.1 | 51.0 | 54.4 | 56.1 | 58.3 | 57.8 | 60.3 |
| SpokenArabicDigits | 100.0 | 100.0 | 71.2 | 98.1 | 97.1 | 99.0 | 81.4 | 100.0 | 97.4 | 96.5 | 97.6 | 100.0 | 98.8 | 97.9 | 98.3 | 98.7 |
| UWaveGestureLibrary | 87.8 | 85.9 | 94.4 | 85.1 | 82.8 | 85.3 | 82.1 | 80.3 | 82.3 | 82.5 | 83.8 | 85.3 | 86.6 | 85.3 | 85.8 | 86.7 |
| Average Accuracy | 71.8 | 70.9 | 72.5 | 71.7 | 70.0 | 73.6 | 67.5 | 70.4 | 70.9 | 70.0 | 70.6 | 70.7 | 73.0 | 73.2 | 72.5 | **74.2** |

Table 31: Full results for the **anomaly detection** task. The P, R and F1 represent the precision, recall and F1-score (%) respectively. F1-score is the harmonic mean of precision and recall. A higher value of P, R and F1 indicates a better performance.

| Datasets | | SMD | | | MSL | | | SMAP | | | SWaT | | | PSM | | | Avg F1 |
|---|---|---|---|---|---|---|---|---|---|---|---|---|---|---|---|---|---|---|
| Metrics | | P | R | F1 | P | R | F1 | P | R | F1 | P | R | F1 | P | R | F1 | (%) |
| SCINet | (2022a) | 85.97 | 82.57 | 84.24 | 84.16 | 82.61 | 83.38 | 93.12 | 54.81 | 69.00 | 87.53 | 94.95 | 91.09 | 97.93 | 94.15 | 96.00 | 84.74 |
| MICN | (2023) | 88.45 | 83.47 | 85.89 | 83.02 | 83.67 | 83.34 | 90.65 | 61.42 | 73.23 | 91.87 | 95.08 | 93.45 | 98.40 | 88.69 | 93.29 | 85.84 |
| TimesNet | (2023) | 88.66 | 83.14 | 85.81 | 83.92 | 86.42 | 85.15 | 92.52 | 58.29 | 71.52 | 86.76 | 97.32 | 91.74 | 98.19 | 96.76 | 97.47 | 86.34 |
| DLinear | (2022) | 83.62 | 71.52 | 77.10 | 84.34 | 85.42 | 84.88 | 92.32 | 55.41 | 69.26 | 80.91 | 95.30 | 87.52 | 98.28 | 89.26 | 93.55 | 82.46 |
| LightTS | (2022) | 87.10 | 78.42 | 82.53 | 82.40 | 75.78 | 78.95 | 92.58 | 55.27 | 69.21 | 91.98 | 94.72 | 93.33 | 98.37 | 95.97 | 97.15 | 84.23 |
| MTS-Mixer | (2023b) | 88.60 | 82.92 | 85.67 | 85.35 | 84.13 | 84.74 | 92.13 | 58.01 | 71.19 | 84.49 | 93.81 | 88.91 | 98.41 | 88.63 | 93.26 | 84.75 |
| RLinear | (2023a) | 87.79 | 79.98 | 83.70 | 89.26 | 74.39 | 81.15 | 89.94 | 54.01 | 67.49 | 92.27 | 93.18 | 92.73 | 98.47 | 94.28 | 96.33 | 84.28 |
| RMLP | (2023a) | 87.35 | 78.10 | 82.46 | 86.67 | 65.30 | 74.48 | 90.62 | 52.22 | 66.26 | 92.32 | 93.20 | 92.76 | 98.01 | 93.25 | 95.57 | 82.31 |
| Reformer | (2020) | 82.58 | 69.24 | 75.32 | 85.51 | 83.31 | 84.40 | 90.91 | 57.44 | 70.40 | 72.50 | 96.53 | 82.80 | 59.93 | 95.38 | 73.61 | 77.31 |
| Informer | (2021) | 86.60 | 77.23 | 81.65 | 81.77 | 86.48 | 84.06 | 90.11 | 57.13 | 69.92 | 70.29 | 96.75 | 81.43 | 64.27 | 96.33 | 77.10 | 78.83 |
| Anomaly* | (2021) | 88.91 | 82.23 | 85.49 | 79.61 | 87.37 | 83.31 | 91.85 | 58.11 | 71.18 | 72.51 | 97.32 | 83.10 | 68.35 | 94.72 | 79.40 | 80.50 |
| Pyraformer | (2021a) | 85.61 | 80.61 | 83.04 | 83.81 | 85.93 | 84.86 | 92.54 | 57.71 | 71.09 | 87.92 | 96.00 | 91.78 | 71.67 | 96.02 | 82.08 | 82.57 |
| Autoformer | (2021) | 88.06 | 82.35 | 85.11 | 77.27 | 80.92 | 79.05 | 90.40 | 58.62 | 71.12 | 89.85 | 95.81 | 92.74 | 99.08 | 88.15 | 93.29 | 84.26 |
| Stationary | (2022c) | 88.33 | 81.21 | 84.62 | 68.55 | 89.14 | 77.50 | 89.37 | 59.02 | 71.09 | 68.03 | 96.75 | 79.88 | 97.82 | 96.76 | 97.29 | 82.08 |
| FEDformer | (2022) | 87.95 | 82.39 | 85.08 | 77.14 | 80.07 | 78.57 | 90.47 | 58.10 | 70.76 | 90.17 | 96.42 | 93.19 | 97.31 | 97.16 | 97.23 | 84.97 |
| Crossformer | (2023) | 83.06 | 76.61 | 79.70 | 84.68 | 83.71 | 84.19 | 92.04 | 55.37 | 69.14 | 88.49 | 93.48 | 90.92 | 97.16 | 89.73 | 93.30 | 83.45 |
| PatchTST | (2023) | 87.42 | 81.65 | 84.44 | 84.07 | 86.23 | 85.14 | 92.43 | 57.51 | 70.91 | 80.70 | 94.93 | 87.24 | 98.87 | 93.99 | 96.37 | 84.82 |
| ModernTCN | **(ours)** | 87.86 | 83.85 | 85.81 | 83.94 | 85.93 | 84.92 | 93.17 | 57.69 | 71.26 | 91.83 | 95.98 | 93.86 | 98.09 | 96.38 | 97.23 | **86.62** |

∗ The original paper of Anomaly Transformer (Xu et al., 2021) adopts the temporal association and reconstruction error as a joint anomaly criterion. For fair comparisons, we only use reconstruction error here.

# N  SHOWCASES

We provide showcases to the regression tasks, including the imputation (Figure 13), long-term forecasting (Figure 14 and 15) and short-term forecasting (Figure 16), as an intuitive comparison among different models.

## N.1  IMPUTATION

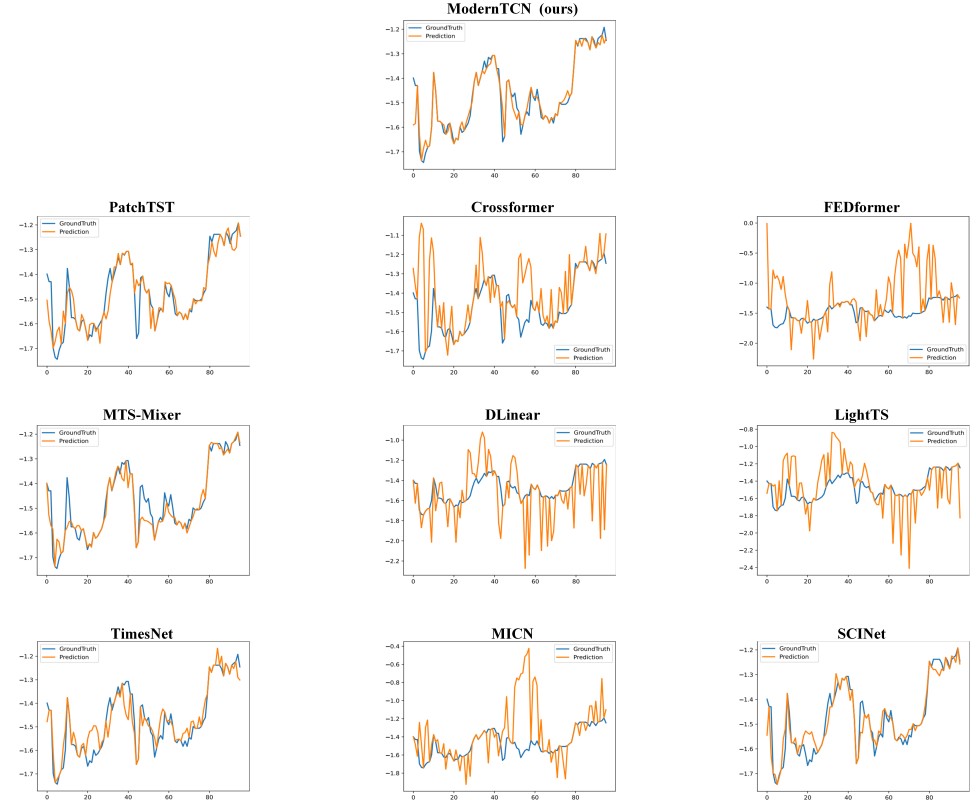

Figure 13: Visualization of ETTh1 imputation results given by models under the 50% mask ratio setting. The blue lines stand for the ground truth and the orange lines stand for predicted values.

## N.2 LONG-TERM FORECASTING

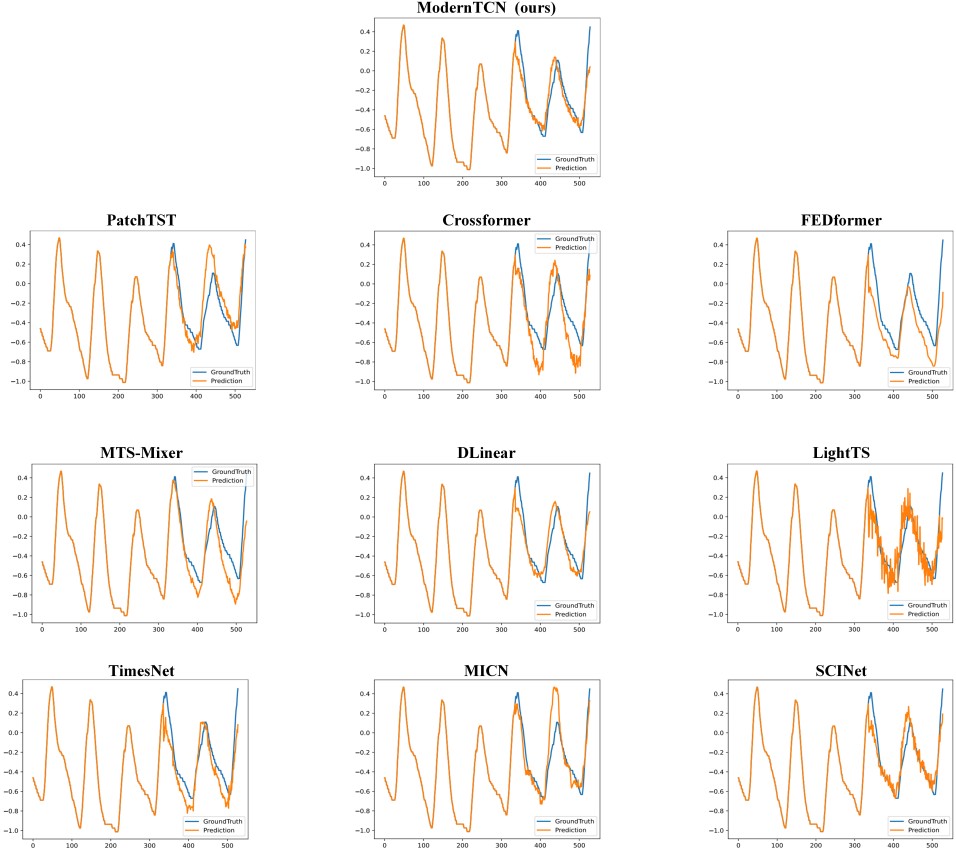

Figure 14: Visualization of ETTm2 predictions by different models under the input-336-predict-192 setting. The blue lines stand for the ground truth and the orange lines stand for predicted values.

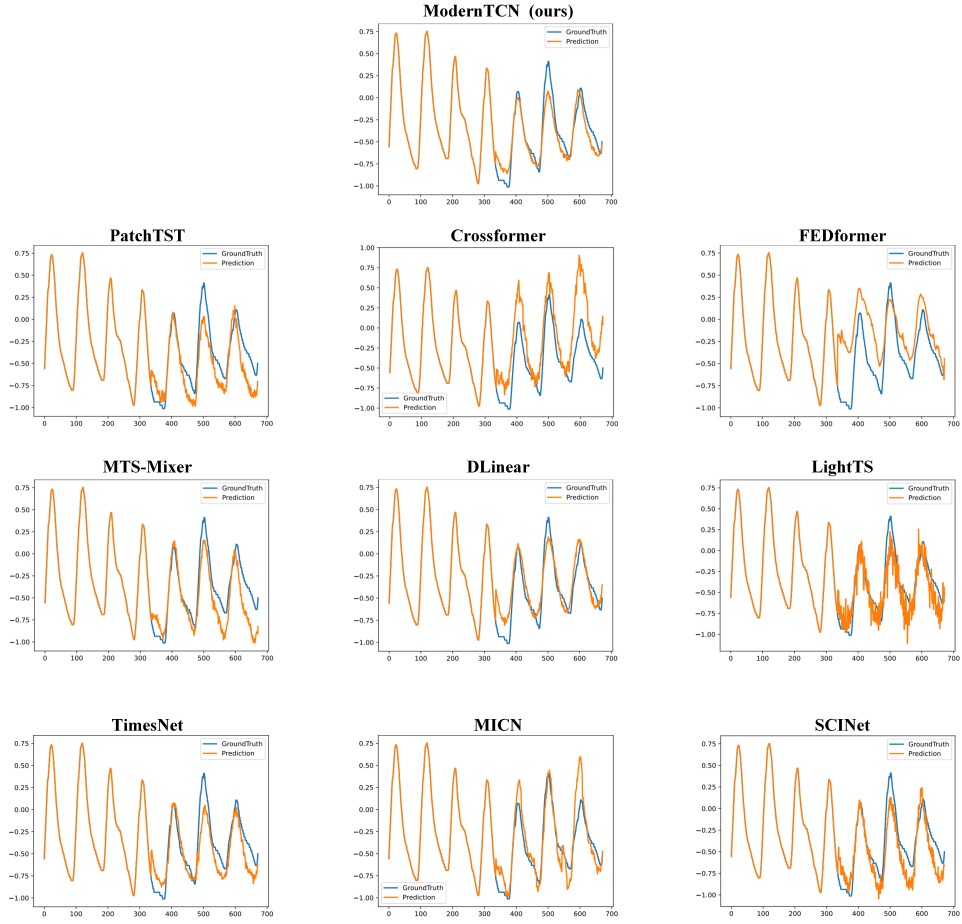

Figure 15: Visualization of ETTm2 predictions by different models under the input-336-predict-336 setting. The blue lines stand for the ground truth and the orange lines stand for predicted values.

### N.3 Short-term Forecasting

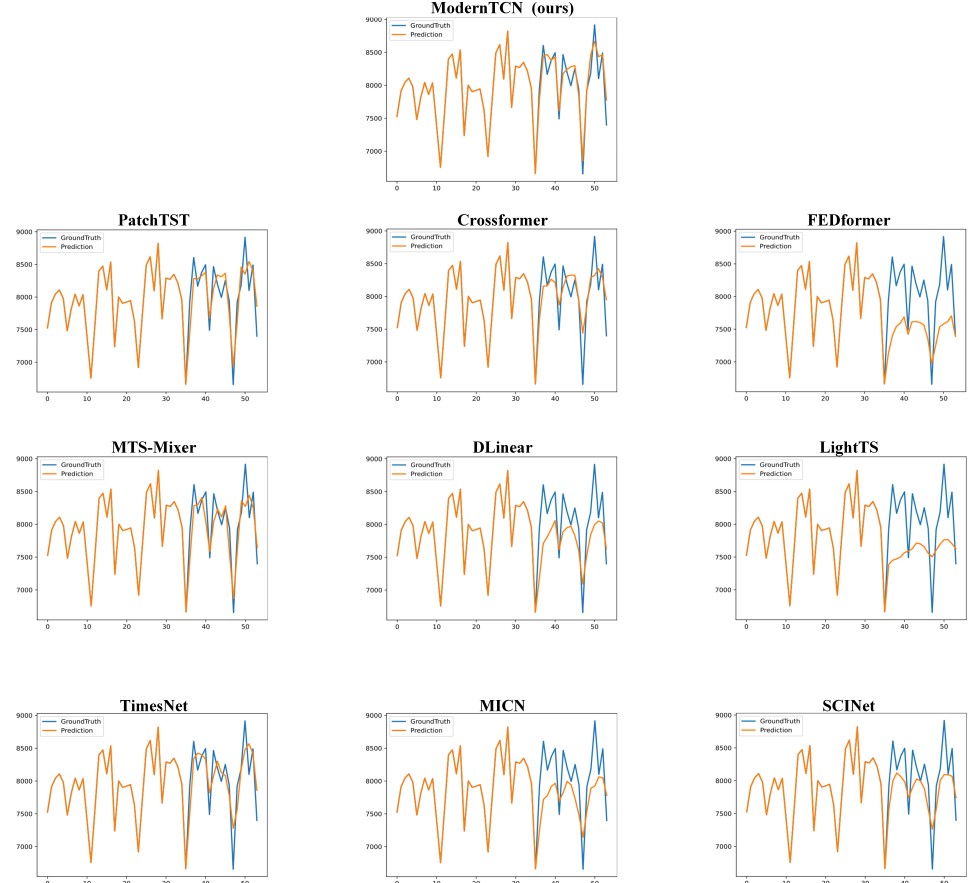

Figure 16: Visualization of M4 predictions by different models. The blue lines stand for the ground truth and the orange lines stand for predicted values.

