# OpenReview forum: "ModernTCN: A Modern Pure Convolution Structure for General Time Series Analysis"
_ICLR.cc/2024/Conference — ICLR 2024 spotlight_

### Official Review · Reviewer_dxG8 · 2023-10-30

**Soundness:** 4 excellent
**Presentation:** 4 excellent
**Contribution:** 3 good
**Rating:** 8
**Confidence:** 3

**Summary:**

-	The paper investigates a modern convolution architecture for time series data. The paper analyzes a pure convolutional architecture for time series analysis with the idea of modernizing the convolution structure to resemble that of a transformer. As existing temporal convolutional neural networks (TCNs) still lack the capability to capture long-term dependencies due to the limited effective receptive fields of the convolutional architecture, the paper proposes ModernTCN, a pure convolutional feed-forward block. It achieves this by optimizing depth-wise convolution with a large kernel size and two types of grouped point-wise convolutions, separately grouped on variable and feature dimensions. The three components capture cross-time, cross-variable, and cross-feature dependencies, respectively. The results demonstrate that the proposed method outperforms transformer-based models and other task-specific baselines in long-term and short-term prediction, classification, imputation, and anomaly detection, providing enhanced efficiency.

**Strengths:**

-	The paper investigates a recent idea of modernization in computer vision applied to time series analysis and identifies consistent performance and efficiency with sufficient comparison experiments and presentational supports.

**Weaknesses:**

-	Although the primary claim for performance improvement centers on enlarging effective receptive fields, the paper lacks theoretical analysis as the authors acknowledge.

-	Some description details are not sufficiently clear. For example, in Table 4, it would be good to clarify which module the authors used as the ‘undecoupled ConvFFN’.

**Questions:**

-	In the experiments on regression tasks, it is concerned whether the RevIN (2021) normalization (as detailed in Appendix B.2) has a significant impact on the performance improvement. Therefore, an additional ablation study about RevIN is requested to enhance clarity. This request is based on the observation that RevIN normalization alone led to improvements in both qualitative and quantitative performance in some of the baseline models.

-	While the paper asserts the claim of enlarging the effective receptive field (ERF) as the key to using convolution in time series analysis in section 5.2, it would be beneficial to include an additional visualization of the ERF of TimesNet (2022) in Figure 1 to further substantiate the claim. TimesNet is another convolution-based model that utilizes shorter kernel sizes and more layers within a block, and it exhibits lower performance than ModernTCN.

-	Simple errata: Enlaring -> Enlarging (in section 5.2.)

---

> ### Author Response · Authors · 2023-11-18
> **Response to Reviewer dxG8 (Part 1/2)**
>
> We would like to sincerely thank Reviewer dxG8 for providing the insightful suggestions and recognizing the value of our work.
>
> > **Weaknesses1 :** The paper lacks theoretical analysis on effective receptive fields (ERFs)
>
> + Thanks for pointing out this issue. In current manuscript, we can only provide the theoretical analysis on ERF of our ModernTCN, but not on ERFs of other baselines.  As mentioned in $\underline{\text{section 5.2}}$, the classic theory of ERFs [1] is designed for pure convolution networks. But most of the convolution-based models in time series community are not pure convolution structures. They only use convolution as part of their designs. Therefore, the classic theory of ERFs may not suitable for those baselines.
>
> + Under such condition, we adopt the visualization of ERFs as an alternative to theoretical analysis for it can provide intuitive comparision of both pure and non-pure convolution-based models.
>
> + It will be our future work to further study the ERFs of the non-pure convolution-based models and find better theory to explain the ERFs of those models.
>
> > **Weaknesses2 :** Some description details are not sufficiently clear. For example, in Table 4, it would be good to clarify which module the authors used as the ‘undecoupled ConvFFN’.
>
> + Thanks for your valuable suggestion. We provide more details about our ablation study in $\underline{\text{Appendix H in revised paper}}$. And in $\underline{\text{Appendix H}}$, we introduce all modules we used in our ablation study with $\underline{\text{Figure 10}}$. Please refer to it for more details.
>
> > **Q1 :** Ablation study about RevIN on regression tasks
>
> + Thanks for your valuable suggestion. We conduct ablation study about RevIN on four regression tasks with 6 models to study the impact of RevIN. Results are provided in $\underline{\text{Appendix L in revised paper}}$. The 6 models are ModernTCN, PatchTST, TimesNet, SCINet and two new baselines: RLinear and RMLP, which all include RevIN as part of their structure.
>
> + Some conclusions are summarized briefly as follows:
>     + Although there is a slight degradation in performance without RevIN, our ModernTCN still achieves competitive performance in four regression tasks, indicating that our designs in ModernTCN also make great contribution to the performance improvement.
>     + We also find that **RevIN doesn't provide consistent improvement under some forecasting settings**, which means our ModernTCN can directly predict from the non-stationary time series to some degree. This finding further confirms that our designs in ModernTCN can bring great performance improvement on their own.
>     + RevIN's impact on different models: Although removing RevIN will cause performance degradation on all 6 models in our experiments, **our ModernTCN is one of the less influenced models**, indicating that our ModernTCN is robust to the usage of RevIN. And we suppose that the extent of RevIN's influence on the model is related to the model's mechanisms.
>     + RevIN's impact on different tasks: We find that short-term forecasting tasks are more sensitive to RevIN.

---

> ### Author Response · Authors · 2023-11-18
> **Response to Reviewer dxG8 (Part 2/2)**
>
> > **Q2 a:** it would be beneficial to include an additional visualization of the ERF of TimesNet in Figure 1.
>
> + Thanks for your valuable suggestion. As we introduced in $\underline{\text{Section 2.1}}$ and $\underline{\text{Section 5.1 'Compared with TimesNet'}}$, TimesNet is special in the family of convolution-based models.
> It transforms 1D time series into 2D-variations and applies 2D convolution on it, which is quite different from other convolution-based models that mainly use 1D convolution.
> Since TimesNet takes a very different path to use convolution in time series analysis, we don't compare the ERF of TimesNet with other 1D convolution-based models in the first version of our manuscript.
>
> + Following your valuable suggestion, we visualize the ERF of TimesNet and we find that the property of TimesNet' ERF is quite different from other 1D convolution-based models'.
> Therefore, we put the visualization result of TimesNet in $\underline{\text{Appendix I in revised paper}}$, along with a detailed comparision of the ERFs between TimesNet and ModernTCN.
> One of the conclusions is that the additional data transformation in TimesNet can also influence the ERF, which leads to the special property of TimesNet’s ERF. Please refer to $\underline{\text{Appendix I}}$ for more details.
>
> > **Q2 b:** Compare two kinds of convolution setting: ‘smaller kernels + more layers’ or ‘larger kernel + less layers’
>
> + Since the ERF of TimesNet is also influenced by the 2D data transformation, comparing the ERFs between TimesNet and ModernTCN cannot directly tell which kind of convolution setting can provide larger ERF.
>
> + Instead, we provide $\underline{\text{Figure 1 (a) and (b)}}$ for this comparision. These two visualization results are all based on the same ModernTCN structure but with the two different convolution settings, therefore can providing a fairer comparision.
> As shown in $\underline{\text{Figure 1 (a) and (b)}}$, ‘larger kernel + less layers’ can provide larger ERF than ‘smaller kernels + more layers’, which is consistent with the theoretical analysis in $\underline{\text{section 5.2}}$.
>
> > **Q3 :** Simple errata: Enlaring -> Enlarging (in section 5.2.)
>
> * Thanks for your reminder. We have fixed it.
>
> > Reference
>
> [1] Wenjie Luo, et al. "Understanding the effective receptive field in deep convolutional neural networks."

---

> > ### Comment · Reviewer_dxG8 · 2023-11-22
> >
> > Dear Authors,
> >
> > Thanks for your kind answer.
> > The response addressed most of the concerns and I still agree that the paper deserves to be accepted.
> >
> > Sincerely,
> >
> > Reviewer dxG8

---

> > > ### Author Response · Authors · 2023-11-23
> > > **Thanks for the Response of Reviewer dxG8**
> > >
> > > Dear Reviewer dxG8,
> > >
> > > Thanks again for providing the insightful review and valuable suggestions, which enable us to make an effective response and help us a lot to improve our paper.
> > >
> > > And we'd also thank you for recognizing and recommending our paper!
> > >
> > > Sincerely,
> > >
> > > Authors

---

### Official Review · Reviewer_exNa · 2023-10-31

**Soundness:** 3 good
**Presentation:** 3 good
**Contribution:** 3 good
**Rating:** 8
**Confidence:** 4

**Summary:**

This paper uses a pure convolution structure (ModernTCN) to perform the time series analysis. Specifically, ModernTCN uses a 1D convolution stem layer, large kernel DWConv and ConvFFN to patchify embedding, capture the temporal dependency of each univariate time series independently and mix the information across feature and variable dimensions respectively. Five mainstream analysis tasks, including long-term and short-term forecasting, imputation, classification and anomaly detection are used to evaluate the effectiveness of ModernTCN.

**Strengths:**

This paper uses the pure convolutional neural network (CNN) for the time series analysis, which is seldomly explored. In addition, five mainstream analysis tasks for the time series data are considered for the evaluation.

**Weaknesses:**

1. The details of the blocks (DWConv and CovnFFN) should be introduced clearly (Some readers may not have related background Knowledge).

2. It is not clear how to select the best results from different input lengths, based on the validation loss or the test results directly?

3. The performance improvement is marginal. In addition, there is another MLP-based work beating PatchTST in same cases, which should be compared with.

Li, Zhe, et al. "Revisiting Long-term Time Series Forecasting: An Investigation on Linear Mapping." arXiv preprint arXiv:2305.10721 (2023).

4. Although it is said that "we also include the state-of-the-art models in each specific task as additional baselines for a comprehensive comparison", some SoTA models are missed in the paper, e.g.,

Short-term forecasting: Wang Xue, et al. "Make Transformer Great Again for Time Series Forecasting: Channel Aligned Robust Dual
Transformer." arXiv preprint arXiv:2305.12095 (2023).

Anomaly Detection: Yang, Yiyuan, et al. "DCdetector: Dual Attention Contrastive Representation Learning for Time Series Anomaly Detection." KDD 2023.

5. How about the short-term forecasting results on the multivariate time series datasets (e.g., the datasets used for long-term forecasting in the paper)?

6. The code is not available for reproducibility.

**Questions:**

Please refer to the Weaknesses.

---

> ### Author Response · Authors · 2023-11-18
> **Response to Reviewer exNa (Part 1/2)**
>
> We would like to sincerely thank Reviewer exNa for providing the thorough and insightful comments.
>
> > **Q1 :** The details of the blocks (DWConv and CovnFFN) should be introduced clearly (Some readers may not have related background knowledge).
>
> + Thanks for your valuable suggestion. We provide more background knowledge and details of ModernTCN block in $\underline{\text{Appendix G in revised paper}}$.
>
> > **Q2 :** How to select the best results from different input lengths, based on the validation loss or the test results directly?
>
> + We select the best results from different input lengths based on the test results directly, which is the same as PatchTST and Crossformer.
>
> + For each model, we provide a set of input lengths and obtain a set of test results. Then we select the best one to represent the model performance.
>
> > **Q3 a:** The performance improvement is marginal.
>
> We would like to illustrate our performance improvement from following points:
> 1. ModernTCN achieves the consistent state-of-the-art performance on five mainstream time series analysis tasks:
>     + ModernTCN surpasses lots of advanced time series models and task-specific methods.
>     + ModernTCN shows great generality. In details, ModernTCN ranks top-1 in all five tasks while other competitors will fall short in some tasks ($\underline{\text{Figure 3 left}}$).
>
> 2. ModernTCN provides a better balance of efficiency and performance:
>     + Apart from achieving state-of-the-art performance, ModernTCN also maintains great efficiency. ($\underline{\text{Figure 3 right}}$).
>
> 3. Compared with popular Transformer-based models:
>     + ModernTCN can compete favorably with the best Transformer-based model (PatchTST) in forecasting tasks and surpass it in imputation, classification and anomaly detection tasks. In details, ModernTCN gains 40.9% relative improvement in imputation tasks. And ModernTCN also improves the accuracy in classification from 72.5% to 74.2% and improves the F1-score in anomaly detection from 84.82% to 86.62%.
>     + When compared with other Transformer-based models like FEDformer, the improvement is more significant. ModernTCN can outperform it even in forecasting tasks (with 32.4% relative improvement in long-term forecasting and 8.9% relative improvement in short-term forecasting).
>     + Meanwhile, ModernTCN has more advantage in efficiency than Transformer-based models.
>
> 4. Compared with popular MLP-based models:
>     + ModernTCN outperforms all MLP-based baselines (including newly added RMLP and RLinear) in all five tasks.
>
> 5. Compared with recent convolution-based models:
>     + ModernTCN outperforms SCINet and MICN in all five tasks. For example, ModernTCN gains 27.4% relative improvement in long-term forecasting and 32.8% relative improvement in imputation. And ModernTCN also improves the accuracy in classification from 71.7% to 74.2%.
>     + TimesNet also demonstrates excellent generality in five mainstream tasks. But ModernTCN still outperforms TimesNet in all five tasks with better efficiency, especially gaining great improvements in long-term forecasting and imputation tasks.
>
> > **Q3 b:** RLinear and RMLP should be compared with.
>
> * Thanks for your valuable suggestion. We add RLinear and RMLP as our baselines. The new results are updated in $\underline{\text{Figure 3,4}}$ and $\underline{\text{Table 1,2,3,27,28,29,30,31}}$.
> + ModernTCN achieves better performance than these two MLP-based models in all five tasks, especially achieving 45.1% relative improvement in imputation and improving the accuracy in classification from 70.6% to 74.2%.
>
> > **Q4 a:** CARD should be compared with in short term forecasting.
>
> + Thanks for your valuable suggestion. We re-implement CARD based on the pseudo-code in the original paper and add it as a short-term forecasting baseline.
> + In original paper, CARD is trained with cosine learning rate decay and linear warm-up. For fair comparison, we remove this additional training scheme.
> + The new results are updated in $\underline{\text{Table 2, 28}}$. ModernTCN can outperform CARD in short-term forecasting.

---

> ### Author Response · Authors · 2023-11-18
> **Response to Reviewer exNa (Part 2/2)**
>
> > **Q4 b:** DCdetector should be compared with in anomaly detection.
>
> + Thanks for your valuable suggestion. We provide following table to compare DCdetector with ModernTCN in anomaly detection tasks. We calculate F1-score as metric.
> + We align the experiment settings as follows:
>     + We set the series length as 105.
>     + We set the anomaly ratio as 0.5 in SMD dataset and 1 in others.
>
> |Dataset|ModernTCN|DCdetector|
> |:---:|:---:|:---:|
> |MSL|84.92|95.21|
> |PSM|97.23|97.16|
> |SMAP|71.26|96.20|
> |SMD|85.81|83.15|
> |SWAT|93.86|96.41|
>
> + As shown in above Table, DCdetector outperforms ModernTCN in 3 of 5 cases and gains great improvement in SMAP and MSL datasets. Some analysis are provided as follows:
>     + The anomaly detection tasks in our paper are based on the **classical reconstruction loss**, while DCdetector proposes a novel detection method based on **contrastive loss** and without reconstruction loss.
>     + As mentioned in the origin paper of DCdetector, **reconstruction-based methods will encounter great challenge in time series anomaly detection and may not be the best solution for time series anomaly detection.**
>     + In contrast, DCdetector provides a better framework for time series anomaly detection which can alleviate the problem faced by reconstruction-based methods.
>  In our paper, **our purpose is to compare the generality of time series backbones and thus only choose the classical reconstruction loss for a fair comparision.**
>  Given the great improvement of contrastive learning in time series anomaly tasks, **it is our future work to study how to combine ModernTCN with this excellent contrastive-based detection framework.**
>
>
> > **Q5 :** How about the short-term forecasting results on the multivariate time series datasets (e.g., the datasets used for long-term forecasting in the paper)?
>
> + Thanks for your valuable suggestion. We conduct the experiments and results are reporeted in $\underline{\text{Appendix K in revised paper}}$.
> + Experiment details are as follows:
>     + We choose the prediction lengths as {$6, 12, 18$}, which meets the prediction length in M4 datasets. And following M4 short-term forecasting tasks, we set input length to be 2 times of prediction length.
>     + We choose models that perform well in multivariate datasets (DLinear, RLinear, RMLP and PatchTST) or perform well in short-term forecasting tasks (PatchTST, CARD and TimesNet) as strong baselines.
>     + All models follow their official configurations on above datasets, we only change the input lengths and prediction lengths.
>     + We calculate MSE and MAE of the multivariate prediction results as metric.
> + As shown in $\underline{\text{Appendix K}}$, ModernTCN can outperform above competitive baselines in most cases, ranking top-1 in 49 of 54 cases.
>
> > **Q6 :** The code is not available for reproducibility.
>
> + We have already provided experimental details and model settings in $\underline{\text{Appendix A,B,C in original paper}}$. And details about tensor shape and model structure are also included in $\underline{\text{main text in our paper}}$. We also add a **Reproducibility Statement** in $\underline{\text{revised paper}}$. **We guarantee to make the code public upon paper acceptance.**

---

> ### Comment · Reviewer_exNa · 2023-11-20
>
> The authors have addressed my concerns, so I have increased my rating.

---

> > ### Author Response · Authors · 2023-11-21
> > **Thanks for the Response of Reviewer exNa**
> >
> > We'd like to thank Reviewer exNa again for providing the valuable review and insightful suggestions. Your constructive suggestions are very helpful for us to improve the paper in a better shape.
> >
> > And we'd also thank you for increasing the rating and recommending our paper!

---

### Official Review · Reviewer_kELp · 2023-10-31

**Soundness:** 3 good
**Presentation:** 4 excellent
**Contribution:** 4 excellent
**Rating:** 8
**Confidence:** 4

**Summary:**

The paper presents a pure convolution architecture for time series analysis. The proposed ModernTCN block is partially inspired by modern convolution blocks as in ConvNeXt. ModernTCN mainly utilizes Depthwise Convolution and Grouped Pointwise convolution to learn cross-variable and cross-feature information in a decoupled way. With the efficiency of convolution operations, ModernTCN achieves impressive performance on various time series analysis tasks. The authors also made a great effort to conduct comprehensive ablation studies to understand the different components of ModernTCN.

**Strengths:**

- The writing is pretty clear and easy to follow.
- State-of-the-art performance on various time series tasks and the convolution-based architecture enjoys good training efficiency.
- Extensive and comprehensive ablation studies to understand different components of the proposed model
- A valuable contribution to picking up pure-convolution based approach for time series analysis

**Weaknesses:**

- It remains unclear how the proposed ModernTCN block would connect to existing literature. While pure convolution architecture is a sweet spot of efficiency and performance, based on what is claimed in the paper, it would be expected that the way to incorporate cross-variable information with ConvFFN would be useful on top of other variable-independent approaches such as PatchTST and DLinear. But this is missing in the current manuscript.

**Questions:**

- What does *Discard Variable Dimension* refer to in Table 4? In Table 4, it is shown that discarding variable dimensions leads to a significant performance degradation. However, in Appendix E, it is shown that univariate forecasting performance is still quite competitive. So what is the difference between the two?
- From Table 4, one important piece of ModernTCN is to exploit cross-variable information. However, notice that some of the other models, e.g. DLinear are variable independent. So is the proposed modern TCN block orthogonal to other approaches and is it possible to apply it on top of variable-independent models like DLinear?
- Based on the appendix, it seems that very different hyperparameters are used for different tasks, e.g. channel number, two kernel sizes, patching size and strides, number of TCN blocks, etc. What is your hyperparameter search space?
- How is it different to use a projection layer (as in PatchTST) vs convolution as the stem layer?

---

> ### Author Response · Authors · 2023-11-18
> **Response to Reviewer kELp （Part 1/2）**
>
> We would like to sincerely thank Reviewer kELp for providing valuable comments and recognizing the value of our work.
>
> > **Q1a :** What does Discard Variable Dimension refer to in Table 4?
>
> + In $\underline{\text{section 3.2}}$, we design the variable-independent embedding to maintain the variable dimension, replacing the common variable-mixing embedding which will lead to the **discard of variable dimension** in the embedded series.
>
> + The essential difference behind these two embedding methods is the opinion of whether we should maintain the variable dimension when analysing multivariate time series.
>
> + **Since applying the variable-mixing embedding to the multivariate time series will discard its variable dimension, we denote it as the setting 'Discard variable dimension'.**
>
> + To verifies our opinion to maintain the variable dimension, we use the setting 'Discard variable dimension' for comparision.
>  The setting 'Discard variable dimension' leads to significant performance degradation, which verifies the necessity and effectiveness of our design to maintain the variable dimension.
>
> + We provide $\underline{\text{Appendix H in revised paper}}\$ to introduce the details of all settings in the ablation study. Please refer to it for more details of setting 'Discard variable dimension'.
>
> > **Q1b:** What is the difference between Discard Variable Dimension in Table 4 and univariate forecasting in Appendix E?
>
> + The former is a multivariate forecasting task. The input is multivariate time series which owes to have a variable dimension to store the information of each variable. And we also need to capture cross-variable information in this multivariate task.
>  But in the setting of 'Discard variable dimension', we apply variable-mixing embedding to the multivariate time series and discard its variable dimension, **leading to the loss of variable information and making it unable to further capture the cross-variable dependency**. As a result, the setting 'Discard variable dimension' leads to a significant performance degradation.
>
> + The latter is a univariate forecasting task. The input is univariate time series which doesn't need a variable dimension.
>  Since it's a univariate tasks, we mainly focus on capturing cross-time information and **don't need to capture the cross-variable information**.
>  Thanks to the larger ERF and better temporal modeling ability in DWConv, our ModernTCN can achieve competitive performance in univariate forecasting tasks.
>
> + Thanks for pointing out this issue, we rephrase the sentence in $\underline{\text{Appendix E in revised paper}}\$ to avoid confusion.
>
> > **Weaknesses1 & Q2 :** Apply the cross-variable component to variable-independent models like DLinear and PatchTST.
>
> + Thanks for your valuable suggestion. Using cross-variable components to exploit cross-variable information is orthogonal to the designs that focus on capturing temporal information.
> So we can apply our cross-variable component to DLinear and PatchTST.
>
> + We conduct the experiments in imputation tasks and provide results in $\underline{\text{Appendix J in revised paper}}$.
> In conclusion, applying the cross-variable component can bring averaged 28.7\% promotion on PatchTST and 19.7\% promotion on DLinear in imputation tasks,
> which validates that our cross-variable component can be used on top of other variable-independent models and improve their performance.

---

> ### Author Response · Authors · 2023-11-18
> **Response to Reviewer kELp （Part 2/2）**
>
> > **Q3 a:**  What is the hyperparameter search space?
>
> The search space is:
> + FFN ratio from {$1,2,4,8$}, number of layers from {$1,2,3$}, large kernel size from {$13,31,51,71$} (which are classic large kernel sizes in CV), and small kernel size is fixed as 5.
> + For patch size and stride, we search patch stride from {$4,8$} under two patching modes ($P=S$ and $P=2 \times S$). But in short-term forecasting tasks, some input lengths are very short, so we reduce the patch size and stride. And in imputation tasks, we don't use patching so as to avoid mixing the masked and un-maskded tokens.
> + For channel number, we search it based on different datasets:
>     + For long-term forecasting and imputation datasets, the search space is {$32,64,128,256,512$}, which are the common values in these datasets.
>     + For short-term M4 datasets, the search space is {$128,256,512,1024,2048$}.
>     + For classification UEA datasets and anomaly detection datasets, we decide the channel number following the criterion used in TimesNet [1].
>
> > **Q3 b:**  Different hyperparameters are used for different tasks.
>
> We use different hyperparameters for different tasks for following reasons:
> 1. The datasets in different tasks may have different property. Thus the suitable hyperparameters like channel number and layer number may vary from different tasks. One evidence is that TimesNet also have different hyperparameters in different tasks according to the original paper [1].
> 2. The patching operation should be different in different tasks. For example, in short-term forecasting tasks, some input lengths are very short, so we need to reduce the patch size and stride. And in imputation tasks, we don't use patching so as to avoid mixing the masked and un-maskded tokens.
>
>
> > **Q4 :** How is it different to use a projection layer (as in PatchTST) vs convolution as the stem layer?
>
> + These two methods are equivalent. Thus, we merge the segmenting and a Linear projection into a single convolution, which can make the code more concise. And the idea of using convolution to conduct the patching and embedding process is also adopted by some CV backbones like ViT [2] and MLP-mixer [3].
>
> + We conduct an ablation study on both patch-style embedding methods with ModernTCN, PatchTST and Crossformer. As the results shown in following table, these two patch-style embedding methods achieve almost the same performance, which is more evidence that the two methods are equivalent.
>
> |ETTh1(mse/mae)|96|192|336|720|
> |:---:|:---:|:---:|:---:|:---:|
> |ModernTCN(conv)|0.368/0.394|0.405/0.413|0.391/0.412|0.450/0.461|
> |ModernTCN(proj)|0.368/0.394|0.405/0.413|0.391/0.412|0.450/0.461|
> |PatchTST(conv)|0.370/0.399|0.413/0.421|0.422/0.436|0.447/0.466|
> |PatchTST(proj)|0.370/0.399|0.413/0.421|0.422/0.436|0.447/0.466|
> |Crossformer(conv)|0.386/0.429|0.419/0.444|0.440/0.461|0.519/0.524|
> |Crossformer(proj)|0.386/0.429|0.419/0.444|0.440/0.461|0.519/0.524|
>
> |ETTm1(mse/mae)|96|192|336|720|
> |:---:|:---:|:---:|:---:|:---:|
> |ModernTCN(conv)|0.292/0.346|0.332/0.368|0.365/0.391|0.416/0.417|
> |ModernTCN(proj)|0.292/0.346|0.332/0.368|0.365/0.391|0.416/0.417|
> |PatchTST(conv)|0.292/0.343|0.332/0.369|0.364/0.392|0.418/0.421|
> |PatchTST(proj)|0.290/0.342|0.332/0.369|0.366/0.392|0.416/0.420|
> |Crossformer(conv)|0.316/0.373|0.375/0.410|0.431/0.442|0.598/0.545|
> |Crossformer(proj)|0.316/0.373|0.377/0.411|0.431/0.442|0.600/0.547|
>
> |Weather(mse/mae)|96|192|336|720|
> |:---:|:---:|:---:|:---:|:---:|
> |ModernTCN(conv)|0.149/0.200|0.196/0.245|0.238/0.277|0.314/0.334|
> |ModernTCN(proj)|0.148/0.200|0.196/0.245|0.238/0.277|0.314/0.334|
> |PatchTST(conv)|0.149/0.198|0.195/0.241|0.245/0.282|0.314/0.334|
> |PatchTST(proj)|0.149/0.198|0.194/0.241|0.245/0.282|0.314/0.334|
> |Crossformer(conv)|0.153/0.217| 0.197/0.269|0.252/0.311|0.318/0.363|
> |Crossformer(proj)|0.153/0.217| 0.197/0.269|0.252/0.311|0.318/0.363|
>
> > Reference
>
> [1] Haixu Wu, et al. "Timesnet: Temporal 2d-variation modeling for general time series analysis."
>
> [2] Alexey Dosovitskiy, et al. "An image is worth 16x16 words: Transformers for image recognition at scale."
>
> [3] Ilya Tolstikhin, et al. "MLP-Mixer: An all-MLP Architecture for Vision."

---

> ### Author Response · Authors · 2023-11-23
> **Thanks to Reviewer kELp**
>
> Dear Reviewer kELp,
>
> Thanks again for providing the valuable review and insightful suggestions. Your constructive suggestions help us a lot to improve the paper in a better shape.
>
> And we'd also thank you for recognizing and recommending our paper!
>
> Sincerely,
>
> Authors

---

### Author Response · Authors · 2023-11-18
**Summary of Revisions**

We sincerely thank all the reviewers for their insightful reviews and valuable comments, which are instructive for us to improve our paper further.

In this paper, we dive into the question of how to better use convolution in time series and propose a novel solution. Unlike previous works that mainly focus on designing extra sophisticated structures to work with the traditional convolution, we take a seldom explored way to modernize and modify the traditional TCN and make it more suitable for time series tasks. Based on this, we propose ModernTCN and successfully bring convolution back to the arena of time series analysis. Experimentally, our pure convolution-based model achieves the consistent state-of-the-art performance on five mainstream analysis tasks with better efficiency.

The reviewers generally hold positive opinions of our paper, in that our writing is **"pretty clear and easy to follow"**, our experiment is **"sufficient"**, our ablation study is **"extensive and comprehensive"**, our model **"identifies consistent performance improvement with good efficiency"** and our idea **"is seldom explored" and "is a valuable contribution"**.

The reviewers also raise insightful and constructive concerns. We make every effort to address all the concerns by providing more detailed descriptions and requested results.
Please refer to the $\underline{\text{revised paper}}\$ for details. All updates are highlighted in orange and here is the summary of the revisions:

+ **More detailed descriptions and clarifications:**
    + **(Reviewer kELp, Reviewer dxG8):** We add $\underline{\text{Appendix H}}\$ to provide more discussion about our ablation study in $\underline{\text{Table 4}}\$. And we also add $\underline{\text{Figure 10}}$ to introduce the detailed experimental settings in our ablation study.

    + **(Reviewer exNa):** We add $\underline{\text{Appendix G}}\$ to provide background knowledge (e.g., group convolution and depth-wise separable convolution) and more details of our ModernTCN block (e.g., design details and implementation details).

+ **More baselines and new experiments (Reviewer exNa):** We add more advanced and latest models as our baselines and update our results in $\underline{\text{Table 1-3, 27-31 and Figure 3-4}}$. We also conduct extra short-term forecasting tasks on multivariate datasets and provide the results in $\underline{\text{Appendix K}}$.
As a result, more than 20 baselines and more than 30 datasets in 5 tasks are included for comprehensive comparisons.
By doing great efforts to complete these comparisons, we verify that our method still achieves the consistent state-of-the-art performance on new baselines and new tasks.

+ **More ablation studies:**

    + **(Reviewer kELp):** We conduct ablation study about stem layer with 3 different models to prove that our embedding method is equivalent to the one used in PatchTST but simpler to implement.
    + **(Reviewer dxG8):** We conduct ablation study about RevIN on four regression tasks with 6 models to study the impact of RevIN. The results in $\underline{\text{Appendix L}}\$ show that our ModernTCN is robust to the usage of RevIN and our designs in ModernTCN make great contribution to the performance improvement.

+ **Connection of proposed ModernTCN block to existing literature (Reviewer kELp):** We apply our cross-variable component to some variable-independent models (e.g., DLinear and PatchTST) and conduct experiments in imputation tasks.
The results in $\underline{\text{Appendix J}}\$ validates that our cross-variable component can be used on top of other variable-independent models and improve their performance.

+ **Theoretical analysis and visualization (Reviewer dxG8):** We include an additional visualization of the ERF of TimesNet and provide a detailed comparision of the ERFs between TimesNet and ModernTCN in $\underline{\text{Appendix I}}$. The result can help to better substantiate the claim of enlarging the effective receptive field (ERF) as the key to using convolution in time series analysis.

The reviewers also raise some specific concerns and point out some writing issues.
We fix the writing issues in our $\underline{\text{revised paper}}\$ and respond to the specific concerns in response to each reviewer. Hope that our response can address reviewers' concerns.

The valuable suggestions from reviewers are very helpful for us to revise our paper to a better shape. We'd be very happy to answer any further questions.

---

### Public Comment · ~David_W._Romero1 · 2023-11-23
**Questions and Concerns Regarding Recent Long Convolution Models**

Dear authors,

Thank you very much for your interesting paper!

Despite their interesting results and findings, I am concerned a bit regarding some of the claims in your paper and its connection to several recent works that propose long convolutional models. In particular, you claim that: "convolution-based models have received less
attention for a long time.", "But previous convolution-based models still have limited ERFs, which prevents their further performance improvements", and "As a result, convolution-based models are losing steam nowadays due to their limited ERFs". However, I would like to call to you attention that there has been several works that in fact provide convolutional models with global receptive fields [1, 2, 3, 4, 5, 6,7,8] --extending even to 4M tokens [9,10]-- (to name only a few). In addition, it was been shown that these works are able to surpass self-attention across countless applications, even to the point of matching them in LLM tasks [11].

I would sincerely appreciate if the authors could comment on this, and add these works to their discussion? To the best of my understanding, the parameterizations provided in CKConv [1], S4 [2], etc, achieve in fact much longer receptive fields that MondernTCN.

Thank you very much for your time and attention.

Best,

David W. Romero

---

### **References**

[1] Romero, D. W., Kuzina, A., Bekkers, E. J., Tomczak, J. M., & Hoogendoorn, M. (2021). Ckconv: Continuous kernel convolution for sequential data. arXiv preprint arXiv:2102.02611.

[2] Gu, A., Goel, K., & Ré, C. (2021). Efficiently modeling long sequences with structured state spaces. arXiv preprint arXiv:2111.00396.

[3] Smith, J. T., Warrington, A., & Linderman, S. W. (2022). Simplified state space layers for sequence modeling. arXiv preprint arXiv:2208.04933.

[4] Poli, M., Massaroli, S., Nguyen, E., Fu, D. Y., Dao, T., Baccus, S., ... & Ré, C. (2023). Hyena hierarchy: Towards larger convolutional language models. arXiv preprint arXiv:2302.10866.

[5] Li, Y., Cai, T., Zhang, Y., Chen, D., & Dey, D. (2022). What Makes Convolutional Models Great on Long Sequence Modeling?. arXiv preprint arXiv:2210.09298.

[6] Fu, D. Y., Arora, S., Grogan, J., Johnson, I., Eyuboglu, S., Thomas, A. W., ... & Ré, C. (2023). Monarch Mixer: A simple sub-quadratic GEMM-based architecture. arXiv preprint arXiv:2310.12109.

[7] Romero, D. W., Bruintjes, R. J., Tomczak, J. M., Bekkers, E. J., Hoogendoorn, M., & van Gemert, J. C. (2021). Flexconv: Continuous kernel convolutions with differentiable kernel sizes. arXiv preprint arXiv:2110.08059.

[8] Knigge, D. M., Romero, D. W., Gu, A., Gavves, E., Bekkers, E. J., Tomczak, J. M., ... & Sonke, J. J. (2023). Modelling Long Range Dependencies in ND: From Task-Specific to a General Purpose CNN. arXiv preprint arXiv:2301.10540.

[9] Fu, D. Y., Kumbong, H., Nguyen, E., & Ré, C. (2023). FlashFFTConv: Efficient Convolutions for Long Sequences with Tensor Cores. arXiv preprint arXiv:2311.05908.

[10] Nguyen, E., Poli, M., Faizi, M., Thomas, A., Birch-Sykes, C., Wornow, M., ... & Ré, C. (2023). Hyenadna: Long-range genomic sequence modeling at single nucleotide resolution. arXiv preprint arXiv:2306.15794.

[11] Massaroli, S., Poli, M., Fu, D. Y., Kumbong, H., Parnichkun, R. N., Timalsina, A., ... & Bengio, Y. (2023). Laughing Hyena Distillery: Extracting Compact Recurrences From Convolutions. arXiv preprint arXiv:2310.18780.

---

### Meta-Review · Area_Chair_7Y1g · 2023-12-06

**Metareview:**

The authors propose a new convolution structure for time series models, showing that their network has much larger receptive fields than prior convolutions structures. The reviewers appreciated the originality of the idea, as well as the design and extensiveness of the experiments, which support the claims made in the paper. A weakness that was indicated is the lack of theoretical analysis - this remains an open question, however having accompanying theory is not necessary for a paper to be valuable. Reviewers also made some suggestions concerning clarity, which the authors addressed successfully. Overall, this is a strong paper which should be of interest to a wide segment of the ICLR community.

**Justification For Why Not Higher Score:**

While the paper is good, and deserving of a spotlight, for it to be an oral presentation it must be truly exceptional.

**Justification For Why Not Lower Score:**

Unanimous accept from the reviewers.

---

### Decision · Program_Chairs · 2024-01-16

Accept (spotlight)